# CHARTING THE DESIGN SPACE OF NEURAL GRAPH REPRESENTATIONS FOR SUBGRAPH MATCHING

**Vaibhav Raj**\*, **Indradyumna Roy**\*, **Ashwin Ramachandran, Soumen Chakrabarti, Abir De**
Department of Computer Science and Engineering, IIT Bombay
`{vaibhavraj, indraroy15, soumen, abir}@cse.iitb.ac.in`
`ashwinramg @ucsd.edu`

## ABSTRACT

Subgraph matching is vital in knowledge graph (KG) question answering, molecule design, scene graph, code and circuit search, etc. Neural methods have shown promising results for subgraph matching. Our study of recent systems suggests refactoring them into a unified design space for graph matching networks. Existing methods occupy only a few isolated patches in this space, which remains largely uncharted. We undertake the first comprehensive exploration of this space, featuring such axes as attention-based vs. soft permutation-based interaction between query and corpus graphs, aligning nodes vs. edges, and the form of the final scoring network that integrates neural representations of the graphs. Our extensive experiments reveal that judicious and hitherto-unexplored combinations of choices in this space lead to large performance benefits. Beyond better performance, our study uncovers valuable insights and establishes general design principles for neural graph representation and interaction, which may be of wider interest. Our code and datasets are publicly available at https://github.com/structlearning/neural-subm-design-space.

## 1 INTRODUCTION

Subgraph matching-based retrieval is essential in tasks like querying knowledge graphs (Liang et al., 2024), biological graphs (Tian et al., 2007), chemical substructure search (Willett et al., 1998), *etc*. In all these applications, given a query graph, the goal is to score—and thereby rank—corpus graphs by how close they come to containing the query graph as a subgraph. Since exact subgraph matching is NP-Complete (Conte et al., 2004), there has been a growing focus on tractable neural methods, with the added benefit of learning the relevance scoring function in a differentiable framework.

**Prior work and complexity of their design choices** Neural architectures for estimating distance between graphs have been extensively studied in recent years (Bai et al., 2019; 2020; Doan et al., 2021; Li et al., 2019; Lou et al., 2020; Roy et al., 2022; Qin et al., 2021; Zhuo and Tan, 2022; Ranjan et al., 2022). These methods use graph neural networks (GNNs) to embed each graph, and then compute a distance between query and corpus graphs' representations. Among them, IsoNet (Roy et al., 2022), IsoNet++ (Ramachandran et al., 2025) and NeuroMatch (Lou et al., 2020) focus specifically on subgraph matching based graph retrieval, by introducing an asymmetric order embeddings (Vendrov et al., 2015) to characterize subgraph containment. In addition, IsoNet has two key design features: it trains an *injective* alignment map between query and corpus graphs; and uses this map to compute a relevance distance based on *set alignment*, in contrast to a distance between whole graph representations. The design space of such retrieval models is cluttered with various choices whose subtle interactions with each other are scarcely studied in prior work. We present two examples.

**(1)** GMN (Li et al., 2019) uses an early interaction GNN, integrating cross-graph signals during message passing. In contrast, IsoNet and NeuroMatch are late interaction models, which do not perform any cross-graph interaction while computing their representations. Challenging the widely held expectation that early interaction is more powerful, IsoNet's late interaction approach outperforms GMN. This raises important questions: Is IsoNet's advantage attributable to its use of set alignment vs. GMN's whole-graph representations, or its injective alignment over GMN's non-injective cross-attention? Can GMN improve with similar set alignment or injective mapping?
**(2)** Although IsoNet and NeuroMatch both use asymmetric hinge distances, NeuroMatch relies on whole-graph representations, while IsoNet uses set alignment at the node or edge level. Other

---

\*Vaibhav and Indradyumna contributed equally. Ashwin Ramachandran did this work while at IIT Bombay.

works (Bai et al., 2019; Zhuo and Tan, 2022; Qin et al., 2021) use neural distance layers that do not enforce either symmetry or asymmetry. This distinction raises an important question: Is IsoNet's set alignment-based distance more effective than other neural or non-neural methods in capturing the nuances of subgraph matching?

## 1.1 OUR CONTRIBUTIONS

Recent system presents aggregate performance comparisons against prior systems. However, such aggregate views miss the opportunity to factor out key design choices involving the representation of and interaction between graph representations, and then systematically explore subtle interplay between these design choices.

**Relevance distance: Set alignment vs. aggregated embedding**  In knowledge graph (KG) alignment, Euclidean distance or cosine similarity between single-vector graph representations (often via aggregating node embeddings) is often less effective than earth mover distance (EMD) between sets of neighbor embeddings (Tang et al., 2020). IsoNet adopts a similar set alignment approach to compare node or edge embeddings across query and corpus graphs. In contrast, GMN collapses node embeddings into a single graph-level representation and uses Euclidean distance between these aggregates. Neural scoring layers (MLP or neural tensor network), where aggregated embeddings are fed into a trainable layer, also remain popular (Qin et al., 2021; Bai et al., 2019).

**Interaction stage: Early vs. late**  Interaction across graphs may be arranged early or late in the comparison network. Early interaction models like GMN (Li et al., 2019) and ERIC (Zhuo and Tan, 2022) use information from the query graph to compute the representation of the corpus graph, and vice-versa, which renders their embeddings strongly dependent on each other. In contrast, a late interaction model like IsoNet (Roy et al., 2022) computes the representation of one graph independent of any other graph, and combine these embeddings at the very last stage during score computation. Early interaction models are expected to be able to compare query and corpus graph neighborhoods directly, while late interaction models have the potential for fast approximate nearest neighbor (ANN) retrieval (Simhadri et al., 2023).

**Interaction structure: Non-injective vs. injective**  GMN models cross-graph interactions using attention which induces a non-injective mapping—multiple nodes in the corpus graph may map to one node in the query graph (or vice versa). In contrast, IsoNet uses a soft permutation (doubly stochastic) matrix to approximate injective alignments. We investigate whether replacing attention-based interaction with a doubly stochastic alignment (even in systems other than IsoNet) leads to improved performance.

**Interaction non-linearity: Trainable vs. fixed**  GMN and IsoNet take divergent approaches to modeling graph-pair interactions. GMN relies on dot-product similarity, while IsoNet applies feedforward layers to both query and corpus embeddings and feed them into an interaction module. The relative merits of these approaches are not fully understood. Additionally, we introduce a natural alternative: incorporating an asymmetric ordering between graph components into the cost matrix.

**Interaction granularity: Nodes vs. edges**  IsoNet introduces the idea of aligning edges instead of nodes, based on the intuition that larger units might provide a more reliable signal for detecting isomorphism. This idea parallels the principle that joint distributions are better approximated when accounting for larger cliques (Koller and Friedman, 2009). It remains unclear how well edge-based alignment can integrate with different cross-graph interaction methods and scoring layers, warranting further investigation.

💡 **Key takeaways and design tips**  Our systematic navigation of the design space resolves hitherto-unexplained observations and provides reliable guidelines for future methods. **(1)** We conclusively explain (late-interaction) IsoNet's earlier-observed superiority over (early-interaction) GMN. If GMN's underlined early interaction is supplemented with any of set alignment, injective structure, hinge nonlinearity, or edge-based interaction, it can readily outperform IsoNet. **(2)** These five design principles are vital, and their combination unveils a novel graph retrieval model that surpasses all existing methods. **(3)** Shifting from late to early interaction may increase computational cost, but compensates for the limitations of relevance distance defined using aggregated single-vector embeddings.

## 2 PRELIMINARIES

**Notation**  We denote the set of query graphs $Q = \{G_q\}$ and the set of corpus graphs $C = \{G_c\}$, where $G_q = (V_q, E_q)$ and $G_c = (V_c, E_c)$ denote a query and corpus graph. Having padded with

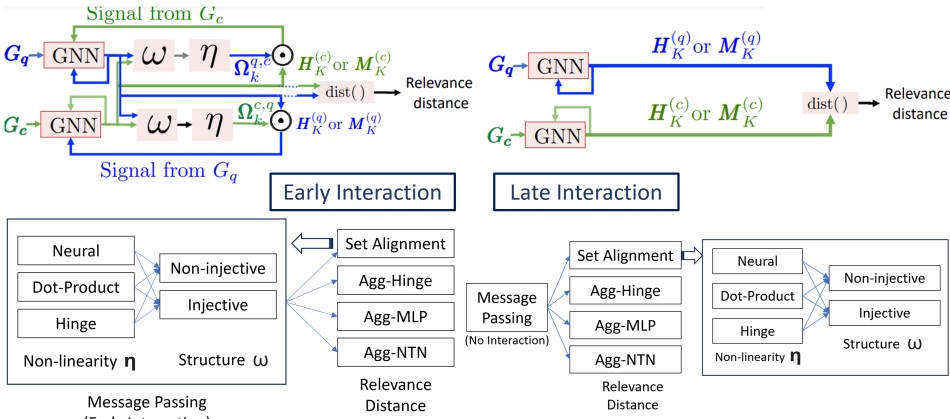

Figure 1: **Early interaction:** Cross graph interactions occur *during* embedding computation layers, with both Green signals from $G_c$ and blue signals from $G_q$ feeding into both graphs. **Late interaction:** No cross-graph signal transfer occurs. Green signals from $G_c$ and blue signals from $G_q$ are restricted to their respective graphs. Final layer embeddings, $\boldsymbol{H}_K^{(q)}$, $\boldsymbol{H}_K^{(c)}$ (node) or $\boldsymbol{M}_K^{(q)}$, $\boldsymbol{M}_K^{(c)}$ (edge), are used to compute relevance distance. **Axes of the design space:** Subgraph matching models work in two stages: message passing and relevance distance. Relevance distance $\text{dist}(\cdot, \cdot)$ can be set alignment, aggregated-hinge, aggregated-MLP or aggregated-NTN. $\omega$ represents the interaction structure (injective vs. non-injective), and $\eta$ defines the interaction non-linearity (Neural, dot product, or hinge). For early interaction (bottom left panel), message passing involves obtaining embeddings via cross-graph alignment, which is used for $\text{dist}(\cdot, \cdot)$ if set alignment is used for relevance distance (shown by thick arrow). In late interaction (bottom right panel), $\eta$ and $\omega$ are absent during message passing, but become active if we use set alignment to approximate $\text{dist}(G_q, G_c)$.

suitable number of nodes to obtain the same number of nodes across $G_q$ and $G_c$, we obtain $\boldsymbol{A}_q$ and $\boldsymbol{A}_c$ as the 0/1 adjacency matrices of size $N \times N$. We write the neighbors of $u$ as $\text{nbr}(u)$. $\boldsymbol{\Omega}$ denotes both node and edge alignment matrices, which are used during cross-graph interaction or relevance computation using set alignment. $\boldsymbol{a}$ denotes an attention matrix, where $\boldsymbol{a}[u, u']$ is the attention weight from $u$ to $u'$, with $\sum_{u'} \boldsymbol{a}[u, u'] = 1$. Similarly, we use $\boldsymbol{b}$ for an edge attention matrix with $\sum_{e'} \boldsymbol{b}[e, e'] = 1$. Given the nodes $u \in V_q$ and $u' \in V_c$, we write $\boldsymbol{h}_k^{(q)}(u) \in \mathbb{R}^{\dim_h}$ and $\boldsymbol{h}_k^{(c)}(u') \in \mathbb{R}^{\dim_h}$ to denote their embeddings, obtained after $k$-layer message passing using a GNN. The last layer is numbered $K$. We collect these embeddings into matrices $\boldsymbol{H}_k^{(q)}, \boldsymbol{H}_k^{(c)} \in \mathbb{R}^{N \times \dim_h}$.

$[\cdot]_+ = \max\{0, \cdot\}$ is the ReLU or hinge function, $[N]$ denotes the set $\{1, ..., N\}$, and $\|\boldsymbol{X}\|_{1,1}$ denotes $\sum_{i,j} |\boldsymbol{X}[i, j]|$. $\mathcal{P}_N$ and $\mathcal{B}_N$ indicate the set of $N \times N$ permutation matrices and doubly stochastic matrices respectively. We generally use 'permutation' and 'alignment' interchangeably.

**Subgraph search and (asymmetric) relevance score** In the context of subgraph matching, we define the relevance label $\text{rel}(G_q, G_c) = 1$, when $G_q \subseteq G_c$ ($G_q$ is a subgraph of $G_c$) and 0 otherwise. Given $G_q$, we define $C_{q\checkmark} = \{G_c \mid \text{rel}(G_q, G_c) = 1\}$ as the set of relevant corpus graphs and $C_{q\boldsymbol{x}} = C \setminus C_{q\checkmark}$ as the irrelevant corpus subset. $\text{dist}(G_q, G_c)$ denotes a relevance distance, which is inversely related to $\text{rel}(G_q, g_c)$. In general, $G_q \subseteq G_c \not\Rightarrow G_c \subseteq G_q$, so relevance labels are inherently asymmetric: $\text{rel}(G_q, G_c) \neq \text{rel}(G_c, G_q)$. This necessitates an asymmetric distance $\text{dist}(\cdot, \cdot)$ with $\text{dist}(G_q, G_c) \neq \text{dist}(G_c, G_q)$ in most cases (Roy et al., 2022; Ranjan et al., 2022; Lou et al., 2020). The goal is to rank $C_{q\checkmark}$ above $C_{q\boldsymbol{x}}$, given query $G_q$.

## 3 DESIGN SPACE OF GRAPH REPRESENTATIONS FOR SUBGRAPH MATCHING

From existing methods (Bai et al., 2019; Zhuo and Tan, 2022; Qin et al., 2021; Zhang et al., 2021; Roy et al., 2022; Li et al., 2019; Ranjan et al., 2022), we catalogue the salient design axes: **(1)** relevance distance, **(2)** interaction stage (late vs. early), **(3)** interaction structure **(4)** interaction non-linearity and **(5)** interaction granularity (node vs. edge). We first present a unified view of these design axes. Then, to control the complexity of exposition, we divide our analysis by interaction granularity, first analyzing the various options for node interactions, followed by a discussion for edge interactions. Figure 1 illustrates our framework.

## 3.1 UNIFIED FRAMEWORK (INTERACTION GRANULARITY: NODE)

Given a query graph $G_q$ and a corpus graph $G_c$, existing work computes the relevance distance in two steps. The first step applies GNN to perform iterative message passing across edges for $K$ propagation layers. At the end of each layer, we obtain the embeddings $\boldsymbol{H}_k^{(q)}$ and $\boldsymbol{H}_k^{(c)}$ with $1 \le k \le K$, for $G_q$ and $G_c$, respectively. The second step compares $\boldsymbol{H}_K^{(q)}$ and $\boldsymbol{H}_K^{(c)}$ to approximate $\mathrm{dist}(\boldsymbol{H}_K^{(q)}, \boldsymbol{H}_K^{(c)}) \approx \mathrm{dist}(G_q, G_c)$. We now present a generalized view of these two steps.

**Generalized message passing** Given a task on a single graph, *e.g.*, link prediction or node classification, GNN updates the node embeddings by collecting messages only from its neighbors, within that graph. If the task involves two graphs, we can adopt either *late* interaction after all $K$ layers of the GNN are computed, or *early* interaction between corresponding layers of the GNNs. E.g., Li et al. (2019) modified the GNN on $G_q$, so that it can incorporate signals from $G_c$, and vice-versa. To capture this interaction, we compute two node alignment matrices $\boldsymbol{\Omega}_k^{q,c} \in \mathbb{R}^{N \times N}$ and $\boldsymbol{\Omega}_k^{c,q} \in \mathbb{R}^{N \times N}$ after each propagation layer $k \in \{0, .., K-1\}$, based on the embeddings $\boldsymbol{H}_k^{(q)}$ and $\boldsymbol{H}_k^{(c)}$. Each of these matrices is computed in two layers: in the first layer, we apply a non-linear map $\eta$ to compute the similarity between $\boldsymbol{H}_k^{(q)}$ and $\boldsymbol{H}_k^{(c)}$ and then use another function $\omega$ to obtain the alignment scores across different node pairs.

$$\boldsymbol{\Omega}_k^{q,c} = \omega\big(\eta\big(\boldsymbol{H}_k^{(q)}, \boldsymbol{H}_k^{(c)}\big)\big), \quad \boldsymbol{\Omega}_k^{c,q} = \omega\big(\eta\big(\boldsymbol{H}_k^{(c)}, \boldsymbol{H}_k^{(q)}\big)\big) \tag{1}$$

As we shall discuss later, $\eta$ and $\omega$ represent the **interaction structure** and **interaction non-linearity**, which are described in Items (3) and (4) in the design space. Here, $\boldsymbol{\Omega}_k^{q,c}[u, u'], \boldsymbol{\Omega}_k^{c,q}[u', u] \in [0, 1]$. Both of them estimate the alignment score that $u' \in V_c$ match with $u \in V_q$. However, they are not necessarily equal except for certain design choices. Next, we use $\boldsymbol{\Omega}_k^{q,c}$ to update the query embeddings $\boldsymbol{h}_k^{(q)}(u) \to \boldsymbol{h}_{k+1}^{(q)}(u)$. To make this update for $u \in V_q$, we multiply $\boldsymbol{\Omega}_k^{q,c} \boldsymbol{H}_k^{(c)}$ to extract signals from the other graph $G_c$; aggregate messages from the neighbors $\mathrm{nbr}(u)$ in its own graph $G_q$, obtained through a neural network $\mathrm{msg}_\theta$; and combine these signals as follows:

$$\boldsymbol{h}_{k+1}^{(q)}(u) = \mathrm{update}_\theta \left( \boldsymbol{h}_k^{(q)}(u); \sum_{v \in \mathrm{nbr}(u)} \underbrace{\mathrm{msg}_\theta(\boldsymbol{h}_k^{(q)}(u), \boldsymbol{h}_k^{(q)}(v))}_{\text{Neighbors in own graph}}; \sum_{u' \in [N]} \underbrace{\boldsymbol{\Omega}_k^{q,c}[u, u']\boldsymbol{h}_k^{(c)}(u')}_{\text{Nodes in other graph}} \right) \tag{2}$$

Likewise, we update $\boldsymbol{h}_k^{(c)}(u') \to \boldsymbol{h}_{k+1}^{(c)}(u')$. For $k = 0$, the embeddings $\{\boldsymbol{h}_0^\bullet(u)\}$ are computed using node features. Having updated the embedding at layer $k + 1$, we assemble them into the the matrices $\boldsymbol{H}_{k+1}^{(q)}$ and $\boldsymbol{H}_{k+1}^{(c)}$ and then use them to compute fresh alignment matrices $\boldsymbol{\Omega}_{k+1}^\bullet$.

**Computation of the relevance distance** $\mathrm{dist}(\cdot, \cdot)$ As discussed in Section 2, the relevance distance is asymmetric, because the relevance label $\mathrm{rel}$ is asymmetric, owing to the asymmetric nature of the subgraph matching task (Roy et al., 2022; Ranjan et al., 2022; Lou et al., 2020). We briefly justify such asymmetric constructions. If $G_q \subseteq G_c$, then there exist some row-column permutation of $\boldsymbol{A}_c$, which will cover all ones in $\boldsymbol{A}_q$. I.e., there will exist a permutation matrix $\boldsymbol{P}$ of size $N \times N$ such that, whenever $\boldsymbol{A}_q[u, u'] = 1$, we have $(\boldsymbol{P}\boldsymbol{A}_c\boldsymbol{P}^\top)[u, u'] = 1$, which indicates any edge $(u, u') \in E_q$ is also present in $E_c$, leading to $G_q \subseteq G_c$. This implies that, for subgraph isomorphism, we will have $\boldsymbol{A}_q \le \boldsymbol{P}\boldsymbol{A}_c\boldsymbol{P}^\top$, elementwise. This suggests the natural definition

$$\mathrm{dist}(G_q, G_c) = \min_{\boldsymbol{P} \in \mathcal{P}_N} \; \left\| [\boldsymbol{A}_q - \boldsymbol{P}\boldsymbol{A}_c\boldsymbol{P}^\top]_+ \right\|_{1,1}. \tag{3}$$

Existing works (Roy et al., 2022; Ranjan et al., 2022; Lou et al., 2020) use a surrogate of $\mathrm{dist}(G_q, G_c)$, by applying a hinge distance across elements of the set of node embeddings $\boldsymbol{H}_K^{(q)}$ and $\boldsymbol{H}_K^{(c)}$ or the whole graph representations $\boldsymbol{g}^{(q)}$ and $\boldsymbol{g}^{(c)}$. Note that if $\mathrm{rel}(G_q, G_c) = 1$ (*i.e.*, $G_q \subseteq G_c$), we have $\mathrm{dist}(G_q, G_c) = 0$. The above distance also draws a parallel from the problem of graph isomorphism, where we have a symmetric distance $\min_{\boldsymbol{P} \in \mathcal{P}_N} \left\| [\boldsymbol{A}_q - \boldsymbol{P}\boldsymbol{A}_c\boldsymbol{P}^\top] \right\|_{1,1}$, instead of asymmetric distance (3). Because of the $\boldsymbol{P}\boldsymbol{A}_c\boldsymbol{P}^\top$ term, Eq. (3) aims to solve a quadratic assignment problem (QAP) which is NP-Hard (Conte et al., 2004). Hence, existing works approximate $\mathrm{dist}(G_q, G_c)$ using an asymmetric distance or trainable distance on the node embeddings $\boldsymbol{H}_K^{(q)}$ and $\boldsymbol{H}_K^{(c)}$. By overloading the notation, we write such a distance as $\mathrm{dist}(\boldsymbol{H}_K^{(q)}, \boldsymbol{H}_K^{(c)})$. In the following, we describe each axis of the design space, beginning with the relevance distance.

### 3.2 RELEVANCE DISTANCE: SET ALIGNMENT VS. AGGREGATED-HINGE VS. AGGREGATED-MLP VS. AGGREGATED-NTN

The relevance distance $\text{dist}(\boldsymbol{H}_K^{(q)}, \boldsymbol{H}_K^{(c)})$ can take four forms, based on (1) Set alignment, (2) Aggregated-hinge: Comparing aggregated graph representations $\boldsymbol{g}^{(q)}, \boldsymbol{g}^{(c)}$ derived from $\boldsymbol{H}_K^{(q)}, \boldsymbol{H}_K^{(c)}$, and subsequently using hinge distance, (3) Aggregated-MLP: a multi-layer perceptron (MLP) to compare $\boldsymbol{g}^{(q)}$ and $\boldsymbol{g}^{(c)}$, and (4) Aggregated-NTN: a neural tensor network (NTN) comparing $\boldsymbol{g}^{(q)}$ and $\boldsymbol{g}^{(c)}$.

**Set alignment** Here, we relax the quadratic assignment problem (QAP) (3) to a linear assignment problem (LAP), where we first represent the graphs $G_q$ and $G_c$ as *sets* of node embeddings $\boldsymbol{H}_K^{(q)}$ and $\boldsymbol{H}_K^{(c)}$, and then quantify the alignment between these sets using the earth mover distance (EMD):

$$\text{dist}(\boldsymbol{H}_K^{(q)}, \boldsymbol{H}_K^{(c)}) = \min_{\boldsymbol{P} \in \mathcal{P}_N} \left\| [\boldsymbol{H}_K^{(q)} - \boldsymbol{P} \boldsymbol{H}_K^{(c)}]_+ \right\|_{1,1} \tag{4}$$

Note that the EMD is induced by hinge distance rather than symmetric $L_p$ distances; therefore, it is not a metric, unlike symmetric EMD. In the context of graph retrieval, solving such an LAP for a large number of graph pairs is daunting, even though for one graph pair, LAP is more tractable than QAP. To get past the blocker of this explicit optimization for each graph pair, we replace $\boldsymbol{P}$ with the alignment matrix $\boldsymbol{\Omega}_K^{q;c}$, which is end-to-end trained under the distant supervision of $\text{rel}(G_q, G_c)$ (Section 2). Therefore, we compute $\text{dist}(\boldsymbol{H}^{(q)}, \boldsymbol{H}^{(c)})$ as:

$$\text{dist}(\boldsymbol{H}_K^{(q)}, \boldsymbol{H}_K^{(c)}) = \left\| \left[ \boldsymbol{H}_K^{(q)} - \boldsymbol{\Omega}_K^{q,c} \boldsymbol{H}_K^{(c)} \right]_+ \right\|_{1,1} \tag{5}$$

The above distance provides a natural alternative to Eq. (4). Here, we fix the ordering of the query nodes and align the corpus nodes using $\boldsymbol{\Omega}_K^{q,c}$. Similarly, one can also fix the order of the corpus nodes and apply $\boldsymbol{\Omega}_K^{c,q}$ on $\boldsymbol{H}_K^{(q)}$. They do not provide a significant difference in retrieval accuracy.

**Aggregated-hinge** Here, instead of representing a graph as a variable-sized set of node embeddings, we represent whole graphs $G_q$ and $G_c$ as single fixed-length vectors $\boldsymbol{g}^{(q)} \in \mathbb{R}^{\dim_g}$ and $\boldsymbol{g}^{(c)} \in \mathbb{R}^{\dim_g}$, using a neural network $\text{comb}_\theta$. Then, in Eq. (3), we replace $\boldsymbol{A}_q$ and $\boldsymbol{P} \boldsymbol{A}_c \boldsymbol{P}^\top$ with the corresponding whole graph representations $\boldsymbol{g}^{(q)}$ and $\boldsymbol{g}^{(c)}$ and compute $\text{dist}(\boldsymbol{H}^{(q)}, \boldsymbol{H}^{(c)})$ as follows:

$$\text{dist}(\boldsymbol{H}_K^{(q)}, \boldsymbol{H}_K^{(c)}) = \|[\boldsymbol{g}^{(q)} - \boldsymbol{g}^{(c)}]_+\|_1, \text{ where } \boldsymbol{g}^{(q)} = \text{comb}_\theta\left(\boldsymbol{H}_K^{(q)}\right), \boldsymbol{g}^{(c)} = \text{comb}_\theta\left(\boldsymbol{H}_K^{(c)}\right). \tag{6}$$

**Aggregated-MLP** Here we feed $\boldsymbol{g}^{(q)}$ and $\boldsymbol{g}^{(c)}$ defined in Eq. (6) into a trainable neural network $\gamma_\theta$ to compute the distance, where $\gamma_\theta$ is an MLP, which is free to implement an asymmetric function:

$$\text{dist}(\boldsymbol{H}_K^{(q)}, \boldsymbol{H}_K^{(c)}) = \gamma_\theta(\boldsymbol{g}^{(q)}, \boldsymbol{g}^{(c)}). \tag{7}$$

**Aggregated-NTN** SimGNN (Bai et al., 2019) proposed the usage of Neural Tensor Networks (Socher et al., 2013) to combine graph-level embeddings, as shown in Eq. (8) below. Here, $L$ represents the latent dimension of the score, $\boldsymbol{W}_{\text{NTN}}^{[1:L]} \in \mathbb{R}^{\dim_g \times \dim_g \times L}$ is a weight tensor, $\boldsymbol{V}_{\text{NTN}} \in \mathbb{R}^{L \times 2\dim_g}$ is a weight matrix applied on the concatenation of the two embedding sets, $b_{\text{NTN}} \in \mathbb{R}^L$ is a bias vector, and $\gamma_\theta$ is an MLP that outputs the scalar distance.

$$\text{dist}(\boldsymbol{H}_K^{(q)}, \boldsymbol{H}_K^{(c)}) = \gamma_\theta\left(\boldsymbol{g}^{(q)\top} \boldsymbol{W}_{\text{NTN}}^{[1:L]} \boldsymbol{g}^{(c)} + \boldsymbol{V}_{\text{NTN}}[\boldsymbol{g}^{(q)}; \boldsymbol{g}^{(c)}] + b_{\text{NTN}}\right) \tag{8}$$

**Time complexity** In set alignment (4), computation of $\boldsymbol{\Omega}_K^{q,c} \boldsymbol{H}_K^{(c)}$ takes $O(N^2)$ time, where $N$ is the number of nodes and $\boldsymbol{H}^{(c)} \in \mathbb{R}^{N \times \dim_h}$. For all other choices, the complexity is $O(N)$.

**Variants in existing works** Among existing works, GotSim (Doan et al., 2021) uses set alignment based distance of the form in Eq. (4) and optimizes for $\boldsymbol{P}$ using the Hungarian algorithm (Edmonds and Karp, 1972). In contrast, IsoNet (Roy et al., 2022) uses set alignment distance of the form in Eq. (5). GMN (Li et al., 2019), GREED (Ranjan et al., 2022), Neuromatch (Lou et al., 2020) use aggregated embedding based distance. SimGNN (Bai et al., 2019), ERIC (Zhuo and Tan, 2022), EGSC (Qin et al., 2021), and GraphSim (Bai et al., 2020) use neural distances. SimGNN (Bai et al., 2019) and ERIC (Zhuo and Tan, 2022) leverage distances in the form of Eq. (8), while EGSC (Qin et al., 2021) and GraphSim (Bai et al., 2020) use Eq. (7)

💡 **Takeaways and design tips** Compressing the entire graph into a low dimensional vector can result in information loss. Therefore, comparing the node embeddings in the set-level granularity results in better performance than their single vector representations. Similar experience has been reported from other domains. In knowledge graph alignment, encoding the neighborhood of an entity as a set, and then aligning two such sets, has been found better than comparing compressed single-vector representations of each entity (Tang et al., 2020). In textual entailment (Lai and Hockenmaier,

2017; Chen et al., 2020; Bevan et al., 2023), allowing cross-interaction between all tokens of the two sentences is generally better than compressing each sentence to a single vector and comparing them. Our work reconfirms this intuition also for subgraph retrieval.

### 3.3 Interaction stage: Early vs. late

**Early interaction**  If $\boldsymbol{\Omega}_k^{\bullet}$ used during computation of the node embedding in Eq. (2) is non-zero, then $\boldsymbol{H}_k^{(c)}$ and $\boldsymbol{H}_k^{(q)}$ become dependent on each other through the third term in the RHS of (2). This leads to early interaction GNN architecture.

**Late interaction**  In late interaction, no cross-graph interaction occurs during the message-passing steps, as shown in Figure 1. Such models are obtained by setting $\boldsymbol{\Omega}_k^{\bullet} = \boldsymbol{0}$ in Eq. (2) (and similarly in $\boldsymbol{h}_{k+1}^{(c)}$). Here, the embedding update step becomes the same as a standard GNN. E.g., for $G_q$, we have $\boldsymbol{h}_k^{(q)}(u) = \text{update}_\theta(\boldsymbol{h}_k^{(q)}(u); \sum_{v \in \text{nbr}(u)} \text{msg}_\theta(\boldsymbol{h}_k^{(q)}(u), \boldsymbol{h}_k^{(q)}(v)))$, separating the computation of query and corpus graph embeddings. However, even if $\boldsymbol{\Omega}_k^{q,c}$ is not used in the message passing interaction, it can still be used during relevance distance computation based on set alignment (5).

**Time complexity**  For early interaction, we have the following computation costs: (1) Computation of $\boldsymbol{h}_u^{\bullet}$ takes $O(N)$ time. (2) For each layer $k \in [K-1]$, computation of $\boldsymbol{\Omega}_k^{\bullet,\bullet}$ takes $O(N^2)$ time. (3) For each layer $k$, message passing in embedding update step takes $O(|E_q|)$ and $O(|E_c|)$ time for $G_q$ and $G_c$; computation of $\sum_{u' \in [N]} \boldsymbol{\Omega}_k^{\bullet,\bullet}[u,u']\boldsymbol{h}_k^{\bullet}(u')$ takes $O(N^2)$ time. Hence, the overall time complexity is $O(KN^2)$. For late interaction model, the overall complexity is $O(K|E_q| + K|E_c|)$.

**Variants in existing works**  GMN (Li et al., 2019) and H2MN (Zhang et al., 2021) use early interaction models, whereas SimGNN (Bai et al., 2019), IsoNet (Roy et al., 2022), ERIC (Zhuo and Tan, 2022), GREED (Ranjan et al., 2022), GOTSim (Doan et al., 2021), EGSC (Qin et al., 2021) and Neuromatch (Lou et al., 2020) use late interaction models.

💡**Takeaways and design tips**  Although late interaction potentially enables fast nearest neighbor search, early interaction is generally known to be superior in text retrieval (Khattab and Zaharia, 2020, Figure 1). The comparison between IsoNet (Roy et al., 2022) vs GMN (Li et al., 2019) apparently contradicts this general trend. Therefore, it is important to resolve this issue using carefully controlled experiments.

### 3.4 Interaction structure: Non-injective vs. Injective

The interaction structure $\omega$ computes the alignment scores $\boldsymbol{\Omega}_k^{\bullet,\bullet}$ in Eq. (1), by normalizing across different dimensions of $\eta(\boldsymbol{H}_k^{(q)}, \boldsymbol{H}_k^{(c)})$. Depending on the nature of normalization, we obtain approximate non-injective or injective mappings for node to node alignment.

**Non-injective**  Here, the function $\omega$ performs softmax-style normalization across different columns for each row, which makes the entries of $\boldsymbol{\Omega}_k^{\bullet,\bullet}$ similar to "attention weights" or "probabilty parameters in a multinomial distribution". Given a temperature $\tau$, we write the values of the alignment matrices:

$$\boldsymbol{\Omega}_k^{q,c}[u,u'] = \frac{e^{\eta(\boldsymbol{h}_k^{(q)}(u),\boldsymbol{h}_k^{(c)}(u'))/\tau}}{\sum_{u^\dagger \in [N]} e^{\eta(\boldsymbol{h}_k^{(q)}(u),\boldsymbol{h}_k^{(c)}(u^\dagger))/\tau}}, \quad \boldsymbol{\Omega}_k^{c,q}[u',u] = \frac{e^{\eta(\boldsymbol{h}_k^{(q)}(u),\boldsymbol{h}_k^{(c)}(u'))/\tau}}{\sum_{u^\dagger \in [N]} e^{\eta(\boldsymbol{h}_k^{(q)}(u^\dagger),\boldsymbol{h}_k^{(c)}(u'))/\tau}} \quad (9)$$

The above form of $\boldsymbol{\Omega}_k^{q,c}$ enables a unique mapping from $G_q$ to $G_c$ but not vice-versa. I.e., the mapping, even if 0-1, could be many-to-one. We note that $\sum_{u'} \boldsymbol{\Omega}_k^{q,c}[u,u'] = 1$. Hence, for any fixed node $u \in G_q$, the matrix $\boldsymbol{\Omega}^{q,c}$ distributes a unit score across all nodes $u' \in G_c$. If $\tau \to 0$, $u$ will be matched to only one node $u'$. However, such allocation is performed independently for different nodes in the query graph. Therefore, given $u' \in G_c$, both $\boldsymbol{\Omega}^{q,c}[u,u']$ and $\boldsymbol{\Omega}^{q,c}[v,u']$ can be high for $u \neq v \in G_q$. Therefore, as $\tau \to 0$, we have two different matched nodes in $G_q$, resulting in an approximate non-injective interaction structure.

**Injective**  Injectivity seeks to mitigate the above limitation. Specifically, in addition to $G_q \to G_c$, an approximate unique mapping is also enforced from $G_c \to G_q$. I.e., for each node $u' \in G_c$, as $\tau \to 0$, there will exist exactly one $u$ for which $\boldsymbol{\Omega}_k^{q,c}[u,u'] = 1$ and vice-versa. These constraints naturally ensure that, as $\tau \to 0$, $\boldsymbol{\Omega}_k^{q,c}$ approaches a permutation matrix. Similar to the relaxation of argmax operation using softmax function, the permutation matrix is approximated with a doubly stochastic matrix using Sinkhorn iterations (Sinkhorn and Knopp, 1967; Cuturi, 2013; Mena et al., 2018). Instead of only row normalization, it excutes iterative row-column normalization for $T$ steps,

starting with initialization: $\boldsymbol{Z}_0 = \exp(\eta(\boldsymbol{H}_k^{(q)}, \boldsymbol{H}_k^{(c)})/\tau)$ at $t = 0$.

$$\boldsymbol{Z}_{t+1}[u, u'] = \frac{\boldsymbol{Z}_t'[u, u']}{\sum_{v' \in [N]} \boldsymbol{Z}_t'[u, v']}, \text{ where } \boldsymbol{Z}_t'[u, u'] = \frac{\boldsymbol{Z}_t[u, u']}{\sum_{v \in [N]} \boldsymbol{Z}_t[v, u']}, \text{ for all } (u, u') \quad (10)$$

As $T$ grows, $\boldsymbol{Z}_T$ approaches a doubly stochastic matrix, satisfying $\sum_{u' \in [N]} \boldsymbol{Z}_T[u, u'] = 1$ and $\sum_{u \in [N]} \boldsymbol{Z}_T[u, u'] = 1$. After $T$ iterations, we set $\boldsymbol{\Omega}_k^{q,c} = \boldsymbol{Z}_T$ and $\boldsymbol{\Omega}_k^{c,q} = \boldsymbol{Z}_T^\top$.

**Time complexity**  Computation of $\boldsymbol{\Omega}_k^{c,q}$ and $\boldsymbol{\Omega}_k^{q,c}$ requires $O(N^2)$ time for a non-injective mapping, where $N$ is the number of nodes. For an injective mapping, $T$ iterations of row-column normalization raise the complexity to $O(TN^2)$.

**Variants in existing work**  GMN (Li et al., 2019) and H2MN (Zhang et al., 2021) use non-injective mapping during early interaction, whereas GotSim (Doan et al., 2021) uses injective mapping. However, as mentioned earlier, GotSim (Doan et al., 2021) optimizes the alignment using the Hungarian method. IsoNet (Roy et al., 2022) uses trainable Sinkhorn iterations (10).

💡 **Takeaways and design tips**  The combinatorial definition of graph matching includes finding an injective mapping between pairs of nodes from the two graphs. The mapping is also an interpretable artifact. Attention from one node to all nodes in the other graph, even if maintained from each graph separately, cannot achieve a consistent 1-1 mapping. Our experiments suggest that an injective mapping (or its continuous relaxation — doubly stochastic matrices) performs better.

### 3.5  INTERACTION NON-LINEARITY: NEURAL VS. DOT PRODUCT VS. HINGE

Alignment computation involves three possible choices of $\eta$:

***Neural:*** $\eta(\boldsymbol{H}_k^{(q)}, \boldsymbol{H}_k^{(c)})[u, u'] = \mathrm{LRL}_\theta(\boldsymbol{h}_k^{(q)}(u))^\top \mathrm{LRL}_\theta(\boldsymbol{h}_k^{(c)}(u'))$, where $\mathrm{LRL}_\theta$ is a Linear-ReLU-Linear network with parameter $\theta$.

***Dot product:*** Here, $\eta(\boldsymbol{H}_k^{(q)}, \boldsymbol{H}_k^{(c)})[u, u'] = (\boldsymbol{h}_k^{(q)}(u))^\top \boldsymbol{h}_k^{(c)}(u')$.

***Hinge:*** Here, $\eta(\boldsymbol{H}_k^{(q)}, \boldsymbol{H}_k^{(c)})[u, u'] = -\left\|[\boldsymbol{h}_k^{(q)}(u) - \boldsymbol{h}_k^{(c)}(u')]_+\right\|_1$.

**Time complexity**  All methods require $O(N^2)$ complexity, since we compute one scalar for each node pair and computations along the embedding dimension are considered $O(1)$.

**Variants in existing works**  GMN (Li et al., 2019), H2MN (Zhang et al., 2021) perform dot-product operations, and IsoNet (Roy et al., 2022) and GOTSim (Doan et al., 2021) use a neural model. In this paper, we also investigate hinge-based similarity as an alternative, with justification provided in Appendix D.1.

**Comparison between different design choices**  Prior experimental work lacks systematic analysis of this issue, which we address here.

### 3.6  GENERALIZING ALIGNMENTS FROM NODES TO EDGES

We complete our design space by altering the interaction and alignment granularity from nodes to edges, which is largely unexplored. While IsoNet (Roy et al., 2022) proposed edge based interaction, this was explored with specific choices of other design axes: set alignment, late interaction, injective interaction structure and neural interaction non-linearity. More details are in Appendix D.

**Unified framework**  Instead of node alignment (1), here we compute edge alignment matrices $\boldsymbol{\Omega}_k^{q,c}$ and $\boldsymbol{\Omega}_k^{c,q}$. These matrices are computed using edge embeddings $\boldsymbol{M}_k^{(q)} = [\boldsymbol{m}_k^{(q)}(e) \,|\, e \in [N']] \in \mathbb{R}^{N' \times \dim_m}$ and $\boldsymbol{M}_k^{(c)} = [\boldsymbol{m}_k^{(c)}(e') \,|\, e' \in [N']] \in \mathbb{R}^{N' \times \dim_m}$, where $N'$ is the total number of edges after padding. After suitable padding with edges, we have: $\boldsymbol{\Omega}_k^{q,c} = \omega(\eta(\boldsymbol{M}_k^{(q)}, \boldsymbol{M}_k^{(c)}))$, $\boldsymbol{\Omega}_k^{c,q} = \omega(\eta(\boldsymbol{M}_k^{(c)}, \boldsymbol{M}_k^{(q)}))$ Edge embeddings are obtained as $\boldsymbol{m}_k^{(q)}(u, v) = \mu_\theta(\boldsymbol{h}_k^{(q)}(u), \boldsymbol{h}_k^{(q)}(v))$ and $\boldsymbol{m}_k^{(c)}(u', v') = \mu_\theta(\boldsymbol{h}_k^{(c)}(u'), \boldsymbol{h}_k^{(c)}(v'))$, where $\mu_\theta$ is a neural network. To compute $\boldsymbol{m}_\bullet^{(q)}$, we use the alignment matrix $\boldsymbol{\Omega}_k^{q,c}$ applied on the edge embeddings in $G_c$ for cross graph signals ($\boldsymbol{m}_{k+1}^{(c)}(u)$ is computed likewise):

$$\boldsymbol{h}_{k+1}^{(q)}(u) = \mathrm{update}_\theta\left(\boldsymbol{h}_k^{(q)}(u); \sum_{v \in \mathrm{nbr}(u)} \mathrm{msg}_\theta\left(\boldsymbol{h}_k^{(q)}(u), \boldsymbol{h}_k^{(q)}(v), \boldsymbol{m}_k^{(q)}(e)\right)\right) \quad (11)$$

$$\boldsymbol{m}_{k+1}^{(q)}(e) = \mathrm{join}_\theta\left(\underbrace{\mathrm{msg}_\theta\left(\boldsymbol{h}_{k+1}^{(q)}(u), \boldsymbol{h}_{k+1}^{(q)}(v), \boldsymbol{m}_k^{(q)}(e)\right)}_{\text{Query graph}}, \underbrace{\sum_{e' \in E_c} \boldsymbol{\Omega}_k^{q,c}[e, e'] \boldsymbol{m}_k^{(c)}(e')}_{\text{Corpus graph}}\right) \quad (12)$$

***Relevance distance:*** We compute alignment between edge embedding sets as $\big\|[M_k^{(q)} - \Omega^{q,c} M_k^{(c)}]_+\big\|_{1,1}$ similar to Eq. (5) used in node interactions. Set aggregate based distances or neural distances are computed similar to Eqs. (6) and (7).

***Interaction stage:*** Eq. (12) results in early interaction for a non-zero $\Omega_k^{q,c}$, whereas setting $\Omega_k^{\bullet,\bullet}$ to zero leads to a late interaction model (with $\text{join}_\theta$ becoming dormant).

***Interaction structure:*** Similar to section (3.4) $\omega$ can be injective or non-injective. For injective $\omega$, we perform iterative Sinkhorn iterations on $\eta(m^{(q)}(e), m^{(c)}(e'))$ for different edge pairs $e \in G_q, e' \in G_c$, whereas for non-injective $\omega$, we use attention.

***Interaction non-linearity:*** Similar to section (3.5), we use neural, dot product and hinge.

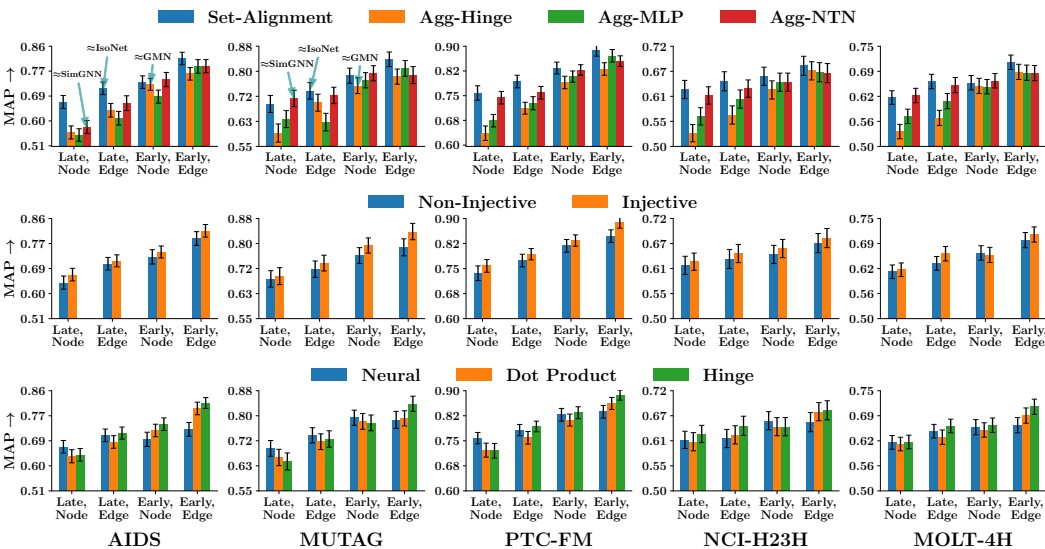

Figure 2: MAP for various choices of design axes. Each column corresponds to a data set. Each chart has four bar groups, corresponding to interaction stage (late, early) × granularity (node, edge). In the top row, each color represents a relevance distance (set alignment vs. aggregated hinge, MLP, and NTN). In the middle row, colors correspond to non-injective and injective interactions. In the bottom row, each color represents a different form of interaction non-linearity (neural, dot product, and hinge). Each bar shows the test MAP after choosing all other axes, policies or hyperparameters to maximize validation MAP. Individually, set alignment (first row), early interaction (third and fourth groups of bars in each row), injective mapping (second row), hinge nonlinearity (third row) and edge based interaction (second and fourth groups of bars) are the best choices in their corresponding axes.

## 4 EXPERIMENTS

In this section, we systematically evaluate different configurations across the five salient design axes on ten real datasets with diverse graph sizes. Appendix F contains additional experiments.

### 4.1 EXPERIMENTAL SETUP

**Datasets**  We select ten real-world datasets from the TUDatasets repository (Morris et al., 2020), *viz.*, AIDS, Mutag, PTC-FM (FM), NCI, MOLT, PTC-FR (FR), PTC-MM (MM), PTC-MR (MR), MCF and MSRC. Detailed descriptions of these datasets are provided in Appendix E. Here, we present results on the first five datasets, relegating others to Appendix F.

**Training and Evaluation**  We split query graphs $Q = \{G_q\}$ into training, validation and test folds in the ratio 60:15:25. Given $C_{q\checkmark}$, *i.e.*, the set of all corpus graphs that are supergraphs of a query graph $G_q$, we estimate model parameters by minimizing a ranking loss $\sum_{q \in \text{Train-set}} \sum_{c_+ \in C_{q\checkmark}, c_- \notin C_{q\checkmark}} [\delta + \text{dist}(H_K^{(q)}, H_K^{(c+)}) - \text{dist}(H_K^{(q)}, H_K^{(c-)})]_+$, where $\delta$ is the margin hyperparameter, following the approach in (Roy et al., 2022; Li et al., 2019). For each test query $q$, we rank corpus graphs by ascending order of $\text{dist}(H_K^{(q)}, H_K^{(c)})$. We evaluate performance using mean average precision (MAP). (Additional settings, hyperparameters are reported in Appendix E).

## 4.2 RESULTS

**Optimal choice for each design axis**   The problem of selecting the best design choices entails exploring five design axes with 66 configurations. To keep the exploration manageable in Figure 2, each column corresponds to one data set. Within each chart, we vary (what we found as) the two most influential axes: interaction stage (late, early) × granularity (node, edge). The top row explores the effect of the choice of relevance distance. (Therefore, each color corresponds to one choice of relevance distance.) The middle row explores the effect of non-injective vs. injective interactions. The bottom row explores the effect of the choice of interaction non-linearity. This covers the five major design axes. Each chart shows the best MAP over all choices of axes not involved.

| (A) | AIDS | | NCI | | | (B) | AIDS | | NCI | |
|---|---|---|---|---|---|---|---|---|---|---|
| | Agg (Late → Early) | SA (Late) | Agg (Late → Early) | SA (Late) | | | Agg (Node → Edge) | SA. (Node) | Agg (Node → Edge) | SA. (Node) |
| Node | 0.557 → 0.726 | 0.664 | 0.529 → 0.624 | 0.625 | | Late | 0.557 → 0.635 | 0.664 | 0.529 → 0.569 | 0.625 |
| Edge | 0.635 → 0.763 | 0.712 | 0.569 → 0.666 | 0.643 | | Early | 0.726 → 0.763 | 0.734 | 0.624 → 0.666 | 0.654 |

| (C) | AIDS | | NCI | | | (D) | AIDS | | NCI | |
|---|---|---|---|---|---|---|---|---|---|---|
| | Node (Agg → SA) | Edge (Agg) | Node (Agg → SA) | Edge (Agg) | | | Late (Agg → SA) | Early (Agg) | Late (Agg → SA) | Early (Agg) |
| Late | 0.557 → 0.664 | 0.635 | 0.529 → 0.625 | 0.569 | | Node | 0.557 → 0.664 | 0.726 | 0.529 → 0.625 | 0.624 |
| Early | 0.726 → 0.734 | 0.763 | 0.624 → 0.654 | 0.666 | | Edge | 0.635 → 0.712 | 0.763 | 0.569 → 0.643 | 0.666 |

Table 1: Does transition from worse to better choice in one design axis improve the performance of the worse choice in another axis? Highlighted numbers show the best performer.

Figure 2 shows the results, which reveals the following observations. **(1)** *Early interaction performs better than late interaction* (groups 1 vs. 3, 2 vs. 4 in each chart) across almost all datasets. While this is consistent with the observations from other domains (Lai and Hockenmaier, 2017; Chen et al., 2020; Wang et al., 2021; Khattab and Zaharia, 2020), this trend is opposite to what IsoNet (Roy et al., 2022) reports — we dig deeper into this by comparing IsoNet against careful modifications to GMN, later in this section. For AIDS and Mutag, we observe that the closest cousin of GMN outperforms the closest cousin of IsoNet. We attribute this to upgrading GMN's default non-injective mapping to IsoNet's injective mapping. **(2)** In terms of alignment granularity, *edge is better than node* (groups 1 vs. 2, 3 vs. 4). **(3)** *Set alignment is the best relevance distance* (top row). **(4)** *Injective mapping is better than non-injective* (middle row). **(5)** *Hinge is the best nonlinearity* (third row).

**Performance sensitivity to design choices**   To enhance clarity, we present numerical summaries in Table 1 for the AIDS and NCI datasets, providing a quick reference to support our key arguments alongside the detailed bar plots in Figure 2. In panel (A) of Table 1, replacing late interaction with early, while keeping either edge or node granularity fixed, allows aggregated-hinge to outperform set alignment in three out of four cases, with a near tie in the fourth. This demonstrates that shifting from late to early interaction can offset some limitations of relevance distance. In contrast, Panel (B) shows that transitioning from node to edge granularity with aggregated-hinge and set alignment has less predictable effect, with set alignment retaining its lead in late-stage interaction. Panel (C) shows that the performance gain from switching to set alignment is more pronounced in late interaction than early interaction. Panel (D) reveals that while early interaction is generally superior, switching to set alignment in the NCI dataset boosts performance, consistent with observations in IsoNet.

💡 *Design guideline to maximize accuracy through change of only one axis:*  From these insights, we propose the following design guidelines. **(1)** Shifting from late to early interaction can overcome scoring layer limitations, **(2)** In late interaction, switching from aggregated-hinge to set alignment is highly effective, accounting for IsoNet's superiority over GMN; and, **(3)** In the absence of any other design constraints, early edge interaction with set alignment with injective structure and hinge nonlinearity is the optimal configuration.

**Best design choices for accuracy-time trade off**   We investigate how to optimize design choices while considering time constraints, by analyzing the trade-off between MAP and inference time. Figure 3 presents the scatter plot for AIDS dataset, where each point represents a unique design combination. Each subplot replicates the same set of points but colors them according to different choices along the design axes. Here are the key takeaways.

💡 *Design guideline to optimize accuracy-time trade off:* **(1)** Set alignment consistently performs best across both reduced-time late interaction models and high-MAP, slower early interaction setups. In the mid-range accuracy-time trade-off, NTN and MLP occasionally perform better. **(2)** Early interaction models generally achieve higher MAP, though even the fastest early variant is slower than the slowest late interaction one. Some late interaction designs provide better MAP, making them an effective choice if time is scarce. **(3)** Injective mappings consistently outperform non-injective ones

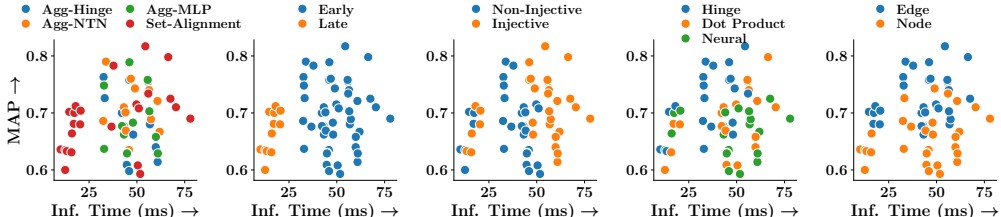

Figure 3: Scatter plot of MAP versus inference time for different design choices on the AIDS dataset. Each point represents a unique combination of design axes, with colors indicating variations in relevance distance, interaction structure, stage, non-linearity, and granularity.

in both high-MAP, high-inference-time scenarios and low-MAP, low-inference-time setups. Their better performance comes with only a slight time increase in low-time settings. But, in high-MAP, in high-inference-time regimes, injective mappings result in significant increases in computational cost, due to iterative Sinkhorn normalization, which makes makes non-injective mappings a viable alternative in that regime. **(4)** Hinge and dot product perform best in high-MAP, slower setups, while hinge and neural options excel in faster, lower-MAP configurations. Hinge offers balanced performance and inference time. **(5)** Edge-level interaction consistently yields higher MAP than node-level interaction, but mostly at the cost of slower query execution.

**Comparison against baselines**   We now highlight the optimal combinations of the design space, selected through the best validation MAP across all possible design choices, against existing state-of-the-art (SOTA) methods. Table 2 shows the results, along with the underlying design choices. As evident from the table, recent models occupy only a few isolated areas within the broader design space, leaving many potential configurations unexplored. We make the following observations: **(1)** Our best combination from early interaction models (**Our-Early-Best**) is the winner across all datasets, whereas the our best possible late interaction model (**Our-Late-Best**) shows the second best performance across four out of five datasets. **(2)** Models using set alignment, *viz.*, IsoNet, **Our-Late-Best**, **Our-Early-Best**, consistently outperform those employing aggregated heuristics. GOTSim finds alignment by using a Hungarian solver, which prevents end-to-end differentiability. resulting in poor performance. Other methods which use aggregated embeddings perform poorly. **(3)** While **Our-Early-Best** leverages set alignment and injective structure, GMN uses aggregated embeddings and non-injective interaction structure, showing lower performance. H2MN, despite using edge granularity, under-performs significantly, suggesting that its other design choices made in early interactions might not effectively leverage the structural advantages.

| | | Rel. Dist. | Structure | Non-lin. | Granularity | AIDS | Mutag | FM | NCI | MOLT |
|---|---|---|---|---|---|---|---|---|---|---|
| Late | SimGNN (Bai et al., 2019) | Agg-NTN | NA | NA | Node | 0.326 | 0.303 | 0.416 | 0.39 | 0.421 |
| | GraphSim (Bai et al., 2020) | Agg-MLP | NA | NA | Node | 0.173 | 0.182 | 0.231 | 0.468 | 0.499 |
| | GOTSim (Doan et al., 2021) | Set align. | Injective | Neural | Node | 0.336 | 0.387 | 0.459 | 0.382 | 0.462 |
| | ERIC (Zhuo and Tan, 2022) | Agg-NTN | NA | NA | Node | 0.512 | 0.558 | 0.624 | 0.556 | 0.549 |
| | EGSC (Qin et al., 2021) | Agg-MLP | NA | NA | Node | 0.5 | 0.446 | 0.643 | 0.528 | 0.555 |
| | GREED (Ranjan et al., 2022) | Agg-hinge | NA | NA | Node | 0.502 | 0.551 | 0.545 | 0.478 | 0.507 |
| | GEN (Li et al., 2019) | Agg-hinge | Non-inj | Dot-Prod | Node | 0.557 | 0.594 | 0.636 | 0.529 | 0.537 |
| | Neuromatch (Lou et al., 2020) | Agg-hinge | NA | NA | Node | 0.454 | 0.583 | 0.622 | 0.513 | 0.552 |
| | IsoNet (Roy et al., 2022) | Set align. | Inj | Neural | Edge | 0.704 | 0.733 | 0.782 | 0.615 | 0.649 |
| | **Our-Late-Best** | Set align. | Inj | Hinge | Edge | 0.712 | 0.721 | 0.793 | 0.643 | 0.662 |
| Early | H2MN (Zhang et al., 2021) | Agg-MLP | Non-inj | Dot-Prod | Edge | 0.267 | 0.282 | 0.364 | 0.381 | 0.405 |
| | GMN-Match (Li et al., 2019) | Agg-hinge | Non-inj | Dot-Prod | Node | 0.609 | 0.693 | 0.686 | 0.588 | 0.603 |
| | **Our-Early-Best** | Set align. | Inj | Hinge | Edge | 0.817 | 0.837 | 0.887 | 0.677 | 0.71 |

Table 2: Comparison of MAP between our best combination of choices across different design axes, for both early (**Our-Early-Best**) and late (**Our-Late-Best**) interaction and the state-of-the-art subgraph matching methods. Green, Blue and Yellow cells indicate best early, best late and second-best late interaction methods.

## 5   CONCLUSION

In this work, we have methodically charted the design space of neural subgraph matching models, which has been sparsely explored by previous models that proposed variants using a limited subset of design choices. Our investigation into hitherto uncharted design regions gives new insights and uncovers design combinations that significantly boost performance. We proposed a set of guidelines to inform future work, helping researchers make optimal design choices based on accuracy requirements and query time constraints.

ACKNOWLEDGEMENT

Indradyumna acknowledges Google PhD fellowship; Abir acknowledges SERB CRG grant; and, Soumen acknowledges SERB and IBM grants.

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

# Charting the Design Space of Neural Graph Representations for Subgraph Matching
# (Appendix)

## A  LIMITATIONS

We present a thorough investigation into the design space of interaction models, with our best model significantly outperforming existing variants, and provide valuable insights into accuracy and time trade-offs. However, there could be potential limitations as follows:

**(1)** Many real-world applications, such as ligand-based screening in chemical compound libraries, rely on 3D graph topology to orient subgraphs for maximizing binding affinity with target graphs. Our approach does not account for such characteristics, which could potentially be encoded as features. However, experimental verification of this encoding is necessary to establish its effectiveness.

**(2)** The robustness of the models within our outlined design space against noise remains uncertain. There may be instances of missing or noisy ground truth information, as well as inaccuracies in node or edge information caused by adversarial attacks or perturbations. Further investigation is required to assess the impact of these factors on model performance.

**(3)** In scenarios where out-of-distribution (OOD) performance is essential, it is unclear which design choices exhibit greater robustness. Some models may effectively leverage distributional characteristics for subgraph matching in in-distribution datasets but may not perform as reliably when faced with OOD data. This aspect warrants further exploration to determine the models' adaptability to varying data distributions.

## B  BROADER IMPACT

In recent years, there has been a surge of interest in this area, leading to numerous publications that explore different facets of subgraph matching. However, the multitude of design choices available has not been thoroughly investigated, often steering research efforts in suboptimal directions. By carefully navigating this complex design space, we have provided a benchmark that outlines promising axes for future exploration. Our findings offer a foundational framework upon which subsequent research can build, guiding efforts toward more effective and efficient models. Additionally, our insights into the intricacies of subgraph matching and scoring mechanisms could inspire further analyses of failure cases and potential avenues for improvement, fostering advancements that could benefit a wide array of applications.

In terms of practical applications, the broader impact of this work is significant across various real-world graph similarity scoring and retrieval tasks.

**(1) In-silico drug discovery** (Rahman et al., 2009; Wang et al., 2023; Willett et al., 1998). Subgraph matching based retrieval is a critical component of ligand-based virtual screening for in-silico drug discovery, enabling the identification of potential drug candidates by comparing molecular structures to known ligands. This is crucial for quickly searching for potential drug candidates from large chemical compound libraries.

**(2) Querying pathway fragments in biological graph databases** (Tian et al., 2007). In bioinformatics, subgraph matching assists in querying pathway fragments, allowing researchers to retrieve and analyze biological pathways efficiently, thereby facilitating understanding of complex biological interactions and mechanisms.

**(3) Image localization** (Shankar et al., 2016). Subgraph matching techniques are useful in improving the accuracy of computer vision systems in real-world navigation scenarios, by identifying and matching features between images and reference maps.

**(4) Feature correspondence in computer vision** (Saxena et al., 1998; Suh et al., 2015). Furthermore, in computer vision tasks, subgraph matching enables effective feature correspondence, improving object recognition and scene understanding by identifying similar structures across different images.

**(5) Hardware Trojan detection** (Piccolboni et al., 2017). Subgraph matching is useful in hardware security for detecting hardware Trojans in integrated circuits, by identifying malicious modifications.

While our work presents numerous practical applications across various domains, we do not anticipate any ethical concerns inherent in our investigations as part of this paper. Our focus has been on

advancing the understanding and effectiveness of subgraph matching techniques, and we believe that our findings will contribute positively to the relevant fields without raising ethical issues.

## C    RELATED WORK

Parts of the design space we explore have been latent in various applied set and graph analysis communities in recent years.

**Interaction modeling in graphs across different domains**    Knowledge graphs (KGs) provide well-motivated graph search and alignment applications. E.g., one may want partial alignments between KGs with nodes (entities) and edges (relations) labeled in multiple languages. Representing nodes $u, v$ as bags of embeddings of their respective neighborhoods, and featurizing the Cartesian space of their member embeddings, has been found superior to comparing single vectors representing $u$ and $v$ (Tang et al., 2020; Chakrabarti et al., 2022). Others exploit duality between edge and node embeddings (Wu et al., 2019; Sun et al., 2020; Zhu et al., 2020; 2021).

**Set based relevance distance models**    In dense text retrieval (Mitra and Craswell, 2018), a query and a corpus document are each represented by a single vector, obtained by encoding each of them separately through a suitable contextual embedding transformer network, and these vectors are compared with a similarity function (commonly cosine). The late interaction enables use of approximate nearest neighbor indexes (Simhadri et al., 2023). Today's large language models (LLM) beat such "single-vector" probes. Typically, the query and shortlisted passages, together with a tuned instruction or 'prompt' are all concatenated into the input context of an LLM, which is then asked to output passage IDs in decreasing order of relevance (as it deems) (Reddy et al., 2024). However, this early interaction comes with a steep price of slower execution. ColBERT (Khattab and Zaharia, 2020) strikes an effective compromise related to set alignment: compute contextual embeddings of all query and corpus words, and then, for each query word embedding, find the best-matching 'partner' from a candidate passage to score it.

**Subgraph matching in other domains**    Searching knowledge graphs like Wikidata or YAGO present strong motivation for graph matching (Bhalotia et al., 2002; Suchanek et al., 2007; Kasneci et al., 2009; Bast et al., 2016). In such applications, the query is usually a connected small graph with wildcard features on nodes or edges, e.g., (?m, mother-of, Barack Obama), (?m, attended-school, ?s), or a natural language question ("where did Barack Obama's mother go to school?") that is 'compiled' to such a small query graph $G_q$. This graph must then be 'overlaid' on to a large corpus graph (the knowledge graph) with low distortion with regard to node and edge features. XML search (Deutsch et al., 1999; Liu and Chen, 2008) and code search (Ling et al., 2020) have many of the same characteristics.

**Automated network structure search**    Within specific families of graph encoder networks, such as GNNs/GATs (Zhang et al., 2023b) or graph transformers (Zhang et al., 2023a), researchers have proposed super-networks to explore the parametric space of encoder networks, using a bi-level optimization framework. We do not yet see NAS as automating the combination of design spaces described in our paper. But it would be of future interest to investigate if NAS methods can be extended to subgraph search and other combinatorial graph problems, to automatically explore network design spaces.

# D  ADDITIONAL DETAILS ABOUT THE UNIFIED FRAMEWORKS AND DIFFERENT DESIGN COMPONENTS

## D.1  JUSTIFICATION BEHIND THE HINGE NON-LINEARITY

The neural surrogate of the optimization problem for the subgraph isomorphism problem can be written as follows:

$$\min_{\boldsymbol{\Omega}^{q,c} \in \mathcal{P}_N} \left\| [\boldsymbol{H}^{(q)} - \boldsymbol{\Omega}^{q,c} \boldsymbol{H}^{(c)}]_+ \right\|_{1,1} \tag{13}$$

We manipulate the optimization target as follows:

$$\left\| [\boldsymbol{H}^{(q)} - \boldsymbol{\Omega}^{q,c} \boldsymbol{H}^{(c)}]_+ \right\|_{1,1} = \sum_{u,v} \underbrace{\sum_i [\boldsymbol{H}^{(q)}[u,i] - \boldsymbol{H}^{(c)}[v,i]]_+}_{C} \boldsymbol{\Omega}^{q,c}[u,v]$$

$$= \text{Trace}(\boldsymbol{C}^\top \boldsymbol{\Omega}^{q,c})$$

Differentiable optimization of $\boldsymbol{\Omega}_K^{q;c}$ is carried out by adding a entropic regularizer (Cuturi, 2013; Sinkhorn and Knopp, 1967; Mena et al., 2018).

$$\min_{\boldsymbol{\Omega}^{q,c} \in \mathcal{B}_N} \text{Trace}(\boldsymbol{C}^\top \boldsymbol{\Omega}^{q,c}) + \epsilon \sum_{u,v} \boldsymbol{\Omega}^{q,c}[u,v] \log \boldsymbol{\Omega}^{q,c}[u,v] \tag{14}$$

Hence, the iterative row-column normalization described in Eq. (10) is reduced to

$$\boldsymbol{Z}_0 = \exp(\boldsymbol{C}/\tau) \tag{15}$$

$$\boldsymbol{Z}_{t+1}[u,u'] = \frac{\boldsymbol{Z}_t'[u,u']}{\sum_{v' \in [N]} \boldsymbol{Z}_t'[u,v']}, \text{ where } \boldsymbol{Z}_t'[u,u'] = \frac{\boldsymbol{Z}_t[u,u']}{\sum_{v \in [N]} \boldsymbol{Z}_t[v,u']}, \text{ for all } (u,u') \tag{16}$$

Clearly, here $\boldsymbol{C} = \eta(\boldsymbol{H}_k^{(q)}, \boldsymbol{H}_k^{(c)}))$. Hence, such an initialization leads to solve optimization of the problem described in Eq. (13), leading to a high inductive bias.

## D.2  UNIFIED FRAMEWORK FOR EDGE-GRANULARITY MODELS

The message passing framework requires modification to conform to edge-level granularity. The query graph equations are shown in Eqs. (11), (12) in the main text. The equations for corpus graphs are shown below, with edge $e' = (u', v')$.

$$\boldsymbol{h}_{k+1}^{(c)}(u') = \text{update}_\theta \left( \boldsymbol{h}_k^{(c)}(u'); \sum_{v' \in \text{nbr}(u')} \text{msg}_\theta \left( \boldsymbol{h}_k^{(c)}(u'), \boldsymbol{h}_k^{(c)}(v'), \boldsymbol{m}_k^{(c)}(e') \right) \right) \tag{17}$$

$$\boldsymbol{m}_{k+1}^{(c)}(e') = \text{join}_\theta \left( \underbrace{\text{msg}_\theta \left( \boldsymbol{h}_{k+1}^{(c)}(u'), \boldsymbol{h}_{k+1}^{(c)}(v'), \boldsymbol{m}_k^{(c)}(e') \right)}_{\text{Corpus graph}}, \sum_{e \in E_q} \underbrace{\boldsymbol{\Omega}_k^{c,q}[e', e] \, \boldsymbol{m}_k^{(q)}(e)}_{\text{Query graph}} \right) \tag{18}$$

For a late interaction model, Eqs. (12) and (18) respectively transform into the following:

$$\boldsymbol{m}_{k+1}^{(q)}(e) = \text{msg}_\theta \left( \boldsymbol{h}_{k+1}^{(q)}(u), \boldsymbol{h}_{k+1}^{(q)}(v), \boldsymbol{m}_k^{(c)}(e) \right) \tag{19}$$

$$\boldsymbol{m}_{k+1}^{(c)}(e') = \text{msg}_\theta \left( \boldsymbol{h}_{k+1}^{(c)}(u'), \boldsymbol{h}_{k+1}^{(c)}(v'), \boldsymbol{m}_k^{(c)}(e') \right) \tag{20}$$

## D.3  END-TO-END FRAMEWORK FOR OUR BEST MODEL

In this section, we describe the components of our best-performing model **Our-Early-Best**, which ranks first on nine out of ten datasets with respect to MAP.

***Relevance distance:*** Set Alignment
***Interaction stage:*** Early
***Interaction structure:*** Injective
***Interaction non-linearity:*** Hinge
***Interaction granularity:*** Edge

The following equations represent message passing within the GNN which emits $\boldsymbol{M}_K^{(q)}$ and $\boldsymbol{M}_K^{(c)}$.

$$\boldsymbol{h}_{k+1}^{(q)}(u) = \text{update}_\theta \left( \boldsymbol{h}_k^{(q)}(u); \sum_{v \in \text{nbr}(u)} \text{msg}_\theta \left( \boldsymbol{h}_k^{(q)}(u), \boldsymbol{h}_k^{(q)}(v), \boldsymbol{m}_k^{(q)}(e) \right) \right) \tag{21}$$

$$\boldsymbol{m}_{k+1}^{(q)}(e) = \text{join}_\theta \left( \underbrace{\text{msg}_\theta \left( \boldsymbol{h}_{k+1}^{(q)}(u), \boldsymbol{h}_{k+1}^{(q)}(v), \boldsymbol{m}_k^{(q)}(e) \right)}_{\text{Query graph}}, \sum_{e' \in E_c} \underbrace{\boldsymbol{\Omega}_k^{q,c}[e,e']\, \boldsymbol{m}_k^{(c)}(e')}_{\text{Corpus graph}} \right) \tag{22}$$

At each layer of message passing, the edge representations are used to compute the **injective** alignment matrices $\boldsymbol{\Omega}_\bullet^{q,c}$ and $\boldsymbol{\Omega}_\bullet^{c,q}$ using the following equations. $E$ represents the number of edges.

$$\boldsymbol{Z}_0 = \exp(\eta(\boldsymbol{M}_k^{(q)}, \boldsymbol{M}_k^{(c)})/\tau) \tag{23}$$

$$\boldsymbol{Z}_{t+1}[e,e'] = \frac{\boldsymbol{Z}_t'[e,e']}{\sum_{f' \in [E]} \boldsymbol{Z}_t'[e,f']}, \text{ where } \boldsymbol{Z}_t'[e,e'] = \frac{\boldsymbol{Z}_t[e,e']}{\sum_{f \in [E]} \boldsymbol{Z}_t[f,e']}, \text{ for all } (e,e') \tag{24}$$

$$\boldsymbol{\Omega}_k^{q,c} = \boldsymbol{Z}_T \qquad \boldsymbol{\Omega}_k^{c,q} = \boldsymbol{Z}_T^\top \tag{25}$$

The Hinge non-linearity is used to compute the initial alignment estimate $\boldsymbol{Z}_0$ in Eq. (23) as such:

$$\eta(\boldsymbol{M}_k^{(q)}, \boldsymbol{M}_k^{(c)})[e,e'] = - \left\| [\boldsymbol{m}_k^{(q)}(e) - \boldsymbol{m}_k^{(c)}(e')]_+ \right\|_1 \tag{26}$$

Finally, the relevance distance Set align. is used to compute the score between the query and corpus graphs. It utilizes the computation of $\boldsymbol{\Omega}^{q,c}$ and $\boldsymbol{\Omega}^{c,q}$ shown above.

$$\text{dist}(G_q, G_c) = \left\| \left[ \boldsymbol{M}_K^{(q)} - \boldsymbol{\Omega}_K^{q,c} \boldsymbol{M}_K^{(c)} \right]_+ \right\|_{1,1} \tag{27}$$

## D.4 DIFFERENCE BETWEEN GMN AND GMN$^\star$

Li et al. (2019) proposed GMN for the graph isomorphism task, which matches the query and corpus nodes as is. This leads to an **unbalanced** Optimal Transport (OT) problem if the query and corpus graphs differ in size, which is the case for subgraph isomorphism. Roy et al. (2022) introduced padding nodes to have equi-sized graphs, thus converting it into a **balanced** OT problem.

Suppose we have two graphs $G_q = (V_q, E_q)$ and $G_c = (V_c, E_c)$ and we pad the query graph with $|V_c| - |V_q|$ disconnected nodes to obtain $G_q' = (V_q \cup V_{\text{pad}}, E_q)$. Two different non-injective alternatives are described below. As a shorthand, we represent $\text{sim}(a,b) = \eta(\boldsymbol{h}_k^{(q)}(a), \boldsymbol{h}_k^{(c)}(b))$ here.

- Including the padding nodes $V_{\text{pad}}$ to compute the alignment. The following equations hold for all $u \in V_q \cup V_{\text{pad}}$ and $u' \in V_c$.

$$\boldsymbol{\Omega}_k^{q,c}[u,u'] = \frac{e^{\text{sim}(u,u')/\tau}}{\sum_{u^\dagger \in V_c} e^{\text{sim}(u,u^\dagger)/\tau}}, \ \boldsymbol{\Omega}_k^{c,q}[u',u] = \frac{e^{\text{sim}(u,u')/\tau}}{\sum_{u^\dagger \in \mathbf{V_q} \cup \mathbf{V}_{\text{pad}}} e^{\text{sim}(u^\dagger,u')/\tau}} \tag{28}$$

- Mask out the padding nodes from the attention computation. For $u \in V_q, u' \in V_c$, the following equations hold. Note the difference in the denominator for computation of $\boldsymbol{\Omega}^{c,q}$.

$$\boldsymbol{\Omega}_k^{q,c}[u,u'] = \frac{e^{\text{sim}(u,u')/\tau}}{\sum_{u^\dagger \in V_c} e^{\text{sim}(u,u^\dagger)/\tau}}, \ \boldsymbol{\Omega}_k^{c,q}[u',u] = \frac{e^{\text{sim}(u,u')/\tau}}{\sum_{u^\dagger \in \mathbf{V_q}} e^{\text{sim}(u^\dagger,u')/\tau}} \tag{29}$$

For $u \in V_{\text{pad}}, u' \in V_c$, we simply set $\boldsymbol{\Omega}_k^{q,c}[u,u'] = \boldsymbol{\Omega}_k^{c,q}[u',u] = 0$.

The first alternative induces the total mass on the query nodes and that on the corpus nodes to be identical, as should be the case with a perfect alignment and hence, we choose it to improve the inductive bias.

# E ADDITIONAL DETAILS ABOUT EXPERIMENTS

## E.1 DATASETS

Ten datasets are chosen from the TUDatasets repository (Morris et al., 2020), six of which were earlier used by (Roy et al., 2022). Key statistics about these datasets are shown in Table 3. $\text{count}(y)$ represents the number of pairs $(G_q, G_c)$ such that $\text{rel}(G_q, G_c) = y$.

Observe that the final four datasets - NCI, MOLT, MSRC and MCF have significantly higher query and corpus graph sizes than the remaining six. These datasets were added to the analysis to study the generalizability of our methods to large input graph sizes.

Since these are real-world datasets not readily available in a format suitable for the subgraph matching task, we adopt the technique proposed by Lou et al. (2020) to sample query and corpus graphs from these datasets. A graph size $|V|$ is randomly sampled from a range (maximum specified in Table 3) and a Breadth First Search (BFS) is commenced from an arbitrary node till $|V|$ number of nodes are traversed. We perform this operation independently to obtain 300 query graphs and 800 query graphs, and their pairwise subgraph isomorphism relationship (rel, the ground truth) is computed using the VF2 algorithm (Cordella et al., 2004) from the NetworkX library (Hagberg et al., 2008).

| | Avg. $|V_q|$ | Max $|V_q|$ | Avg. $|E_q|$ | Avg. $|V_c|$ | Max $|V_c|$ | Avg. $|E_c|$ | $\text{count}(1)$ | $\text{count}(0)$ | $\frac{\text{count}(1)}{\text{count}(0)}$ |
|---|---|---|---|---|---|---|---|---|---|
| AIDS | 11.61 | 15 | 11.25 | 18.50 | 20 | 18.87 | 41001 | 198999 | 0.2118 |
| Mutag | 12.91 | 15 | 13.27 | 18.41 | 20 | 19.89 | 42495 | 197505 | 0.2209 |
| FM | 11.73 | 15 | 11.35 | 18.30 | 20 | 18.81 | 40516 | 199484 | 0.2085 |
| FR | 11.81 | 15 | 11.39 | 18.32 | 20 | 18.79 | 39829 | 200171 | 0.2043 |
| MM | 11.80 | 15 | 11.37 | 18.36 | 20 | 18.79 | 40069 | 199931 | 0.2056 |
| MR | 11.87 | 15 | 11.49 | 18.32 | 20 | 18.78 | 40982 | 199018 | 0.2119 |
| MSRC | 14.01 | 20 | 20.28 | 45.41 | 50 | 93.58 | 41374 | 198626 | 0.2140 |
| MCF | 22.03 | 26 | 21.60 | 44.79 | 50 | 46.50 | 35951 | 204049 | 0.1805 |
| NCI | 19.02 | 25 | 18.70 | 44.89 | 50 | 46.55 | 40548 | 199452 | 0.2094 |
| MOLT | 19.04 | 25 | 18.69 | 44.94 | 50 | 46.56 | 40177 | 199823 | 0.2058 |

Table 3: Statistics for the 10 datasets selected from the TUDatasets collection (Morris et al., 2020)

## E.2 METRICS

For a query graph $G_q$, we sort the set of corpus graphs $C = \{G_c\}$ in increasing order of $\text{dist}(G_q, G_c)$ to obtain the ranked list $C_{q,\text{sort}}$. As noted in the main text, $C_{q\checkmark}$ refers to the set of supergraphs of $G_q$, and is a subset of $C$.

Average Precision (AP) is defined as the ratio of the rank (according to $C_{q,\text{sort}}$) of a relevant item in $C_{q\checkmark}$ to its rank in $C$, averaged over all relevant corpus items. Note that $\text{rel}(G_q, G_c) = \mathbb{1}[G_c \in C_{q\checkmark}]$.

$$\mathsf{AP}(q) = \frac{1}{|C_{q\checkmark}|} \sum_{\text{pos}=1}^{|C|} \frac{\text{rel}(G_q, C_{q,\text{sort}}[\text{pos}]) \times \sum_{i=1}^{\text{pos}} \text{rel}(G_q, C_{q,\text{sort}}[i])}{\text{pos}}$$

Mean Average Precision is the mean of $\mathsf{AP}(q)$ over all query items $q \in Q$.

$$\mathsf{MAP} = \frac{\sum_{q \in Q} \mathsf{AP}(q)}{|Q|}$$

### E.3 IMPLEMENTATION DETAILS ABOUT GNN, MLPS

We list all the hyperparameters and details for different network components here. we use $\mathrm{Linear}(a, b)$ to denote a linear layer with input dimension $a$ and output dimension $b$. Composition of $\mathrm{Linear}(a, b)$, ReLU and $\mathrm{Linear}(b, c)$ networks is denoted by $\mathrm{MLP}(a, b, c)$ (similarly for $> 3$ arguments).

- Node features are set to $[1]$ following (Roy et al., 2022), since the goal is to ensure feature agnostic subgraph matching. Such identical featurization ensures that two structurally isomorphic graphs have exact same embeddings.
- The node and edge embedding sizes are respectively set to $\dim_h = 10$ and $\dim_m = 20$. Below, we use $\dim$ to refer to one of these based on the granularity of the network involved.
- $\mathrm{msg}_\theta$ computes messages by operating on the concatenation of the embeddings of the nodes that form the edge and optionally the edge's embeddings, as shown in Eq. (2). To ensure that the undirected nature of graphs is respected, $\mathrm{msg}_\theta$ is applied to both ordered pairs $(u, v)$ and $(v, u)$ and their sum is returned. For node-granularity models, the edge embedding is a fixed one-sized vector and thus, the $\mathrm{msg}_\theta$ is a $\mathrm{Linear}(21, 20)$ network. For edge-granularity models, the edge embedding is of size 20, resulting in a $\mathrm{Linear}(40, 20)$ network.
- $\mathrm{update}_\theta$ is modeled as a GRU where the initial hidden state is the node representation (the first argument) of size 10, as shown in Eqs. (2) and (11). Exactly one GRU update is performed on top of this base vector. However, the update vector varies based on the stage and granularity of the network. For early interaction networks with edge granularity and late interaction networks, the update vector is simply an aggregation (sum) of all message vectors corresponding to this node, of dimension 20. For node-granularity early interaction networks, the update vector is the concatenation of the aggregated message vector and the difference between the node representation and the signal from the other graph:

$$\left[ \sum_{v \in \mathrm{nbr}(u)} \mathrm{msg}_\theta(\boldsymbol{h}_k^{(q)}(u), \boldsymbol{h}_k^{(q)}(v)); \left( \boldsymbol{h}_k^{(q)}(u) - \sum_{u' \in [N]} \boldsymbol{\Omega}_k^{q,c}[u, u'] \boldsymbol{h}_k^{(c)}(u') \right) \right]$$

The dimension of this vector is 30 (20 from the first term, 10 from the second term).
- $\mathrm{join}_\theta$ operates on the concatenation of messages from the same graph and alignment-based signal from the other graph. It is modeled as an $\mathrm{MLP}(40, 40, 20)$ network.
- $\mathrm{comb}_\theta$ inputs the node/edge embeddings and outputs whole graph embeddings, as shown in Eq. (6). For each individual node/edge, We first apply a $\mathrm{Linear}(\dim, 2 \cdot \dim)$ network on its embedding, split into two $\dim$-sized vectors and use the first as a gate (by applying sigmoid) for the second vector, $\mathrm{sigmoid}(v_1) \times v_2$. Individual vectors are added and passed through a $\mathrm{Linear}(\dim, \dim)$ layer to obtain the $\dim$-sized whole graph vector.
- The MLP in Eq. (7) operates on the concatenation of whole graph embeddings, and is modeled as an $\mathrm{MLP}(2 \cdot \dim, \dim, 1)$ network.
- Neural Tensor Network - We use a latent dimension of $L = 16$. The MLP in Eq. (8), $\gamma_\theta$ is an $\mathrm{MLP}(16, 8, 4, 1)$ network, resulting in a scalar score for every graph pair.
- $\mathrm{LRL}_\theta$, used for Neural non-linearity in Section 3.5, is an $\mathrm{MLP}(\dim, \dim, 16)$ network.

### E.4 TRAINING DETAILS

We adopt a common training procedure for all models. The Adam optimizer is used to perform gradient descent on the ranking loss with a learning rate of $10^{-3}$ and a weight decay parameter of $5 \times 10^{-4}$. We use a batch size of 128, a margin of 0.5 for the ranking loss, and cap the number of training epochs to 1000. To prevent overfitting on the training dataset, we adopt early stopping with respect to the MAP score on the validation split, with a patience of 50 epochs & a tolerance of $10^{-4}$.

**Seed Selection** We select three integers uniformly at random from the range $[0, 10^4]$ to obtain our seed set $(1704, 4929, 7762)$. We train each model on all three seeds for 10 epochs, choose the best seed per model per dataset based on the MAP score for the validation split and train this selected seed till completion.

**Software & Hardware Details** All models were implemented with PyTorch 2.1.2 in Python 3.10.13. Experiments were run on Nvidia RTX A6000 (48 GB) GPUs.

# F ADDITIONAL EXPERIMENTS

## F.1 SELECTION OF THE BEST CHOICE ON OTHER DATASETS

Figure 4 is an extension of Figure 2 to all ten datasets. We observe similar trends overall across all ten datasets. **(1)** Edge-granularity models dominate those with node granularity and early interaction usually outperforms late interaction. **(2)** Injective mapping allows for significantly greater performance, compared to non-injective. **(3)** Hinge non-linearity remains the best among the three options.

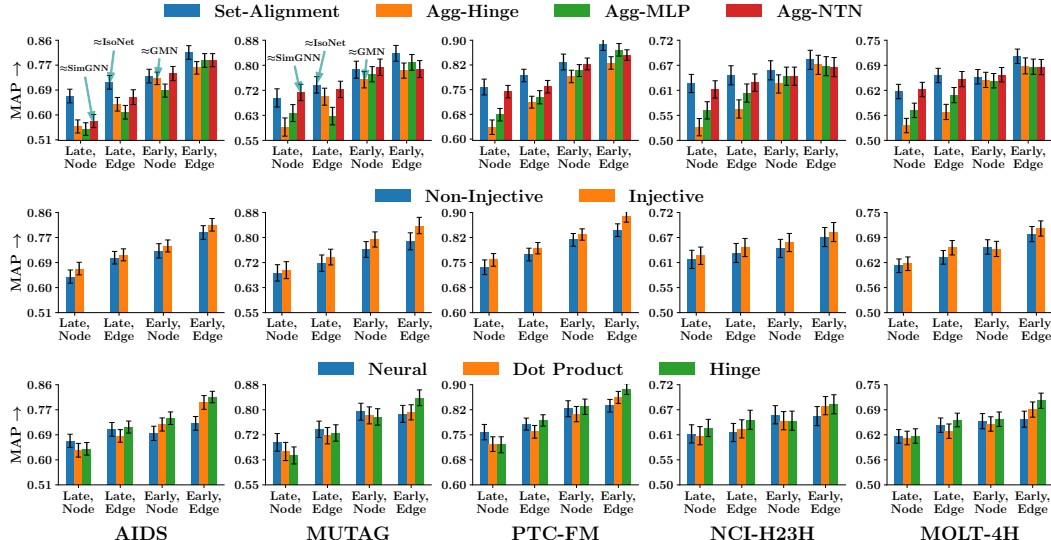

(a) MAP for various choices of design axes for AIDS, Mutag, FM, NCI and MOLT datasets.

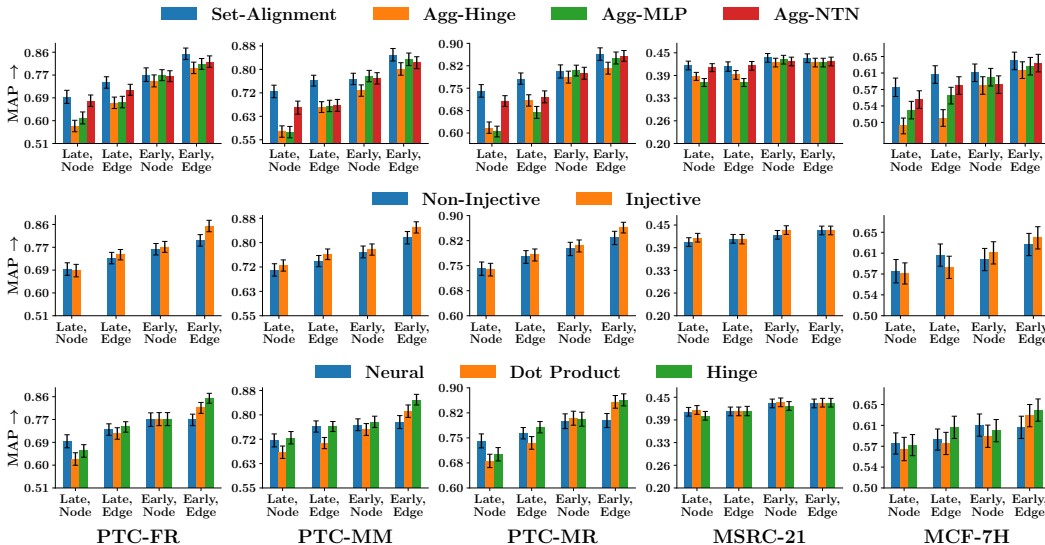

(b) MAP for various choices of design axes for FR, MM, MR, MSRC and MCF datasets.

Figure 4: Each column corresponds to a dataset. Each chart has four bar groups, corresponding to interaction stage (late, early) × granularity (node, edge). In the top row, each color represents a relevance distance (set alignment vs. aggregated hinge, MLP, and NTN). In the middle row, colors correspond to non-injective and injective interactions. In the bottom row, each color represents a different form of interaction non-linearity (neural, dot product, and hinge). Each bar shows the test MAP after choosing all other axes, policies or hyperparameters to maximize validation MAP. Individually, set alignment (first row), early interaction (third and fourth groups of bars in each row), injective mapping (second row), hinge nonlinearity (third row) and edge based interaction (second and fourth groups of bars) are the best choices in their corresponding design axes.

## F.2 COMPARISON BETWEEN GMN* ON OTHER DATASETS

Here, we expand Table **??** to include all datasets. We observe that the trends noted in the main text remain largely consistent:

**(1)** Modifying the non-linearity $\eta$ in Eq. (1) from dot product to hinge significantly improves GMN*, outperforming IsoNet in seven out of ten datasets.

**(2)** Changing interaction granularity from Node to Edge, or interaction structure from non-injective to injective offers moderate benefits, improving GMN* over IsoNet in five and four datasets respectively.

**(3)** Altering the relevance distance dist from aggregated-hinge to set alignment does not help, as IsoNet continues to outperform GMN* in eight out of ten datasets. This is likely due to the sensitivity of relevance distance in late interaction models, whereas GMN* uses early interaction.

**(4)** Combining set alignment and injective mapping, allows GMN* to surpass IsoNet's performance.

**(5)** While Hinge non-linearity alone helps GMN* outperform IsoNet, combining it with set alignment reduces performance. This is perhaps due to the incompatibility of GMN*'s non-injective mapping with hinge non-linearity.

**(6)** Combining hinge non-linearity with injective structure affords moderate benefits, allowing GMN* to outperformIsoNet in three out of ten cases.

**(7)** Transitioning to edge-level models and adjusting any single design axis consistently improves GMN* over IsoNet across all datasets, underscoring the critical role of edge granularity in enhancing subgraph matching performance.

| | Prior art | | Change one design axis in GMN* | | | | Change two design axes in GMN* | | | | | |
|---|---|---|---|---|---|---|---|---|---|---|---|---|
| Dataset | IsoNet | GMN | Agg-Hinge → Set-Align. | Non-Inj → Inj | DP → Hinge | Node → Edge | Set align., Inj. | Set align., Hinge | Inj., Hinge | Set align., Edge | Inj., Edge | Hinge, Edge |
| AIDS | 0.704 | 0.609 | 0.608 | 0.64 | 0.726 | 0.7 | 0.71 | 0.676 | 0.662 | 0.715 | 0.755 | 0.763 |
| Mutag | 0.733 | 0.693 | 0.754 | 0.75 | 0.723 | 0.734 | 0.779 | 0.669 | 0.734 | 0.779 | 0.763 | 0.753 |
| FM | 0.782 | 0.686 | 0.759 | 0.79 | 0.79 | 0.78 | 0.8 | 0.772 | 0.785 | 0.812 | 0.826 | 0.831 |
| NCI | 0.615 | 0.588 | 0.603 | 0.619 | 0.618 | 0.625 | 0.632 | 0.599 | 0.624 | 0.635 | 0.656 | 0.666 |
| MOLT | 0.649 | 0.603 | 0.62 | 0.63 | 0.651 | 0.653 | 0.652 | 0.637 | 0.635 | 0.659 | 0.67 | 0.686 |
| FR | 0.734 | 0.667 | 0.706 | 0.73 | 0.75 | 0.727 | 0.774 | 0.678 | 0.733 | 0.799 | 0.8 | 0.772 |
| MM | 0.758 | 0.627 | 0.691 | 0.699 | 0.723 | 0.706 | 0.736 | 0.711 | 0.713 | 0.789 | 0.777 | 0.799 |
| MR | 0.764 | 0.683 | 0.715 | 0.736 | 0.787 | 0.746 | 0.803 | 0.768 | 0.752 | 0.81 | 0.803 | 0.815 |
| MSRC | 0.411 | 0.416 | 0.423 | 0.423 | 0.414 | 0.421 | 0.436 | 0.419 | 0.403 | 0.421 | 0.415 | 0.423 |
| MCF | 0.587 | 0.549 | 0.553 | 0.574 | 0.584 | 0.601 | 0.593 | 0.601 | 0.575 | 0.609 | 0.616 | 0.612 |

Table 4: Changing one or two design axes in GMN outperforms IsoNet's default design.

### F.3 MAP-TIME TRADE OFF ON OTHER DATASETS (CORRESPONDING FIGURES ARE IN NEXT TWO PAGES)

In the main paper, we presented MAP vs. inference time scatter plots for the AIDS dataset. Here, we extend this analysis to all ten datasets to validate whether the initial conclusions still hold. Our key observations are:

**(1)** The trade-off between early and late-stage interaction remains consistent across all datasets. While early interaction models yield significantly higher MAP scores, they come with a notable increase in inference time. In contrast, late interaction models are much faster, and although they lag behind the top-performing early models, they still outperform a significant portion of the early variants. This makes late interaction a viable compromise when inference time is a critical factor.

**(2)** Set alignment continues to be the best choice in both early and late interaction setups, offering the highest MAP scores without a significant increase in inference time compared to other relevance distance measures.

**(3)** Injective interaction structures consistently outperform non-injective variants across most datasets. However, there are exceptions, such as in MSRC, where some non-injective variants come close to matching the best-performing injective models.

**(4)** Interestingly, in several datasets, including MM, FR, MOLT, and Mutag, hinge non-linearity significantly boosts MAP scores compared to other variants. This reinforces our recommendation that hinge non-linearity is an optimal choice for subgraph matching tasks.

**(5)** Edge-level interaction continues to outperform node-level granularity. This supports our guideline that, in graphs with a similar number of nodes and edges, edge granularity is the superior design choice.

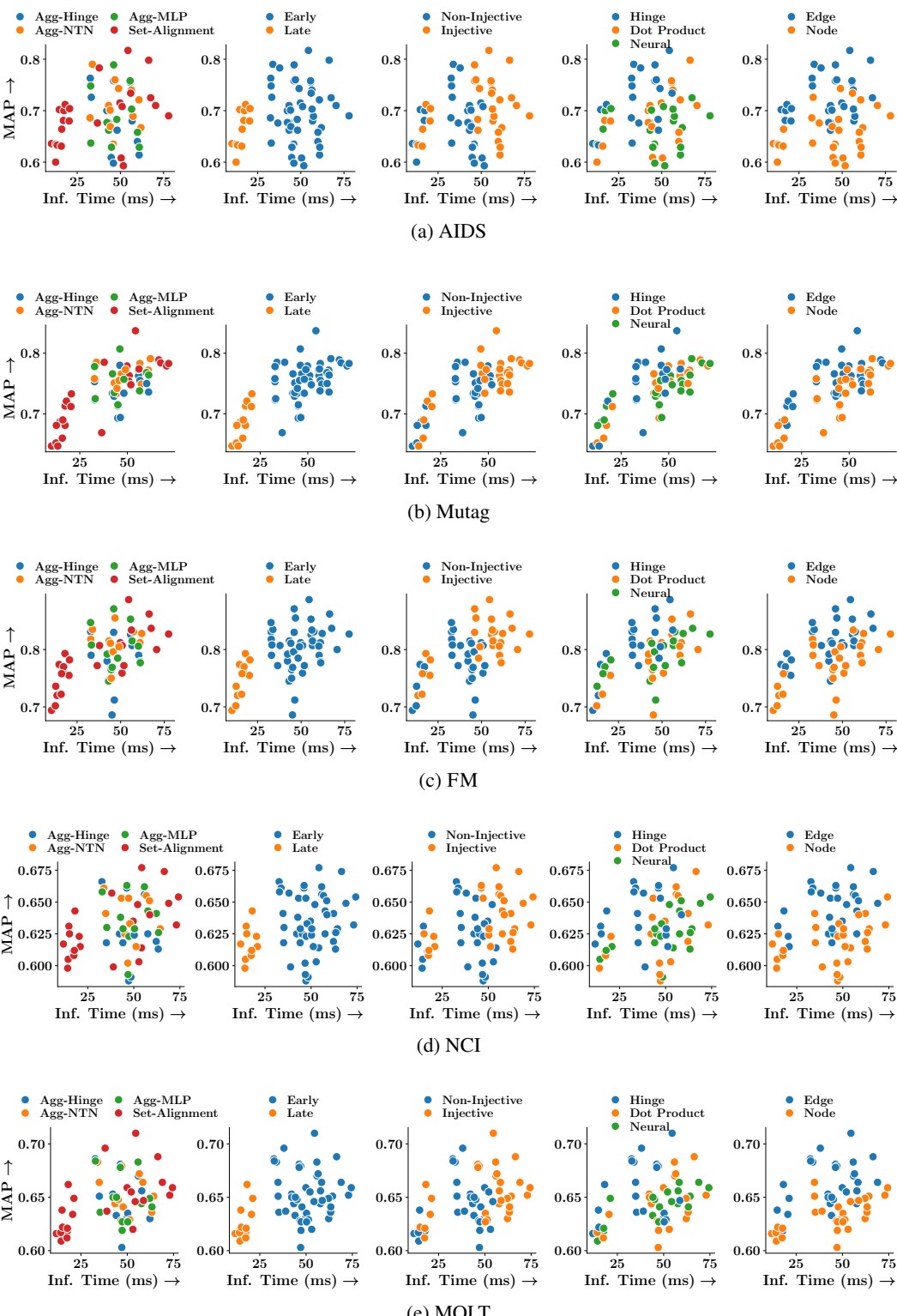

Figure 5: Scatter plot of MAP versus inference time for different design choices. Each point represents a unique combination of design axes, with colors indicating variations in relevance distance, interaction structure, stage, non-linearity, and granularity.

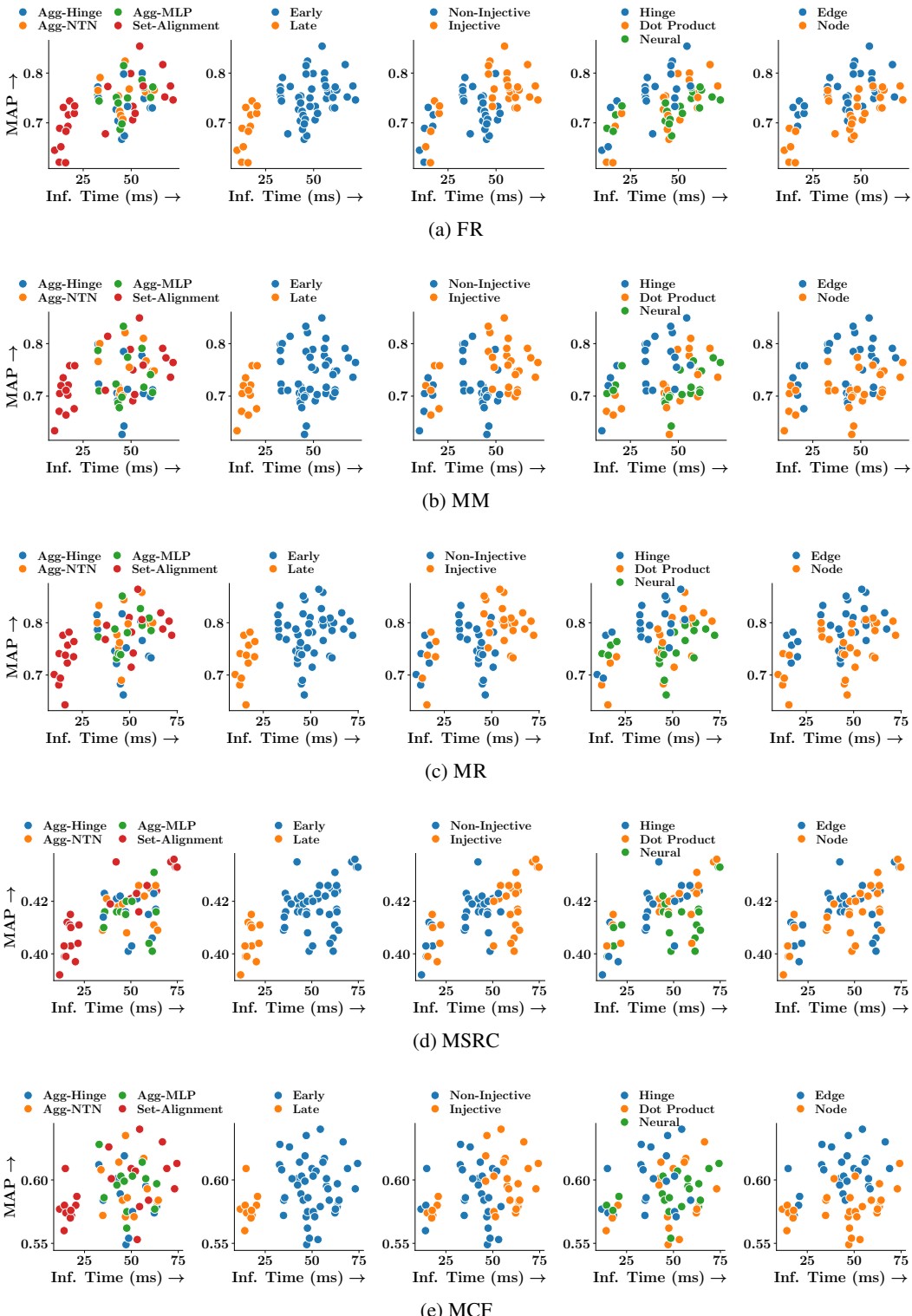

Figure 6: Scatter plot of MAP versus inference time for different design choices. Each point represents a unique combination of design axes, with colors indicating variations in relevance distance, interaction structure, stage, non-linearity, and granularity.

### F.4 COMPARISON WITH BASELINES ON OTHER DATASETS

In Table 2 of the main paper, we compared the optimal configuration from our design space exploration against state-of-the-art baselines across five datasets. Here, we extend these comparisons to all 10 datasets. Table 5 details the design choices made by each baseline according to our specified design axes, highlighting that they occupy only sparse regions of the design space. Table 6 presents the full performance results, reaffirming our previous observations. Specifically, (**1**) our best early interaction model (**Our-Early-Best**) consistently outperforms all other methods across datasets, and (**2**) models utilizing set alignment consistently surpass those relying on aggregated heuristics. (**3**) Despite using an Injective structure with Set align. relevance distance, GOTSim performs poorly because it uses a Hungarian solver (instead of Sinkhorn iterations) which doesn't allow end-to-end differentiability.

| | Model | Rel. Dist. | Structure | Non-linearity | Granularity |
|---|---|---|---|---|---|
| Late | SimGNN (Bai et al., 2019) | Agg-NTN | NA | NA | Node |
| | GraphSim (Bai et al., 2020) | Agg-MLP | NA | NA | Node |
| | GOTSim (Doan et al., 2021) | Set align. | Injective | Neural | Node |
| | ERIC (Zhuo and Tan, 2022) | Agg-NTN | NA | NA | Node |
| | EGSC (Qin et al., 2021) | Agg-MLP | NA | NA | Node |
| | GREED (Ranjan et al., 2022) | Agg-hinge | NA | NA | Node |
| | GEN (Li et al., 2019) | Agg-hinge | Non-injective | Dot Product | Node |
| | NeuroMatch (Lou et al., 2020) | Agg-hinge | NA | NA | Node |
| | IsoNet (Roy et al., 2022) | Set align. | Injective | Neural | Edge |
| | **Our-Late-Best** | Set align. | Injective | Hinge | Edge |
| Early | H2MN (Zhang et al., 2021) | Agg-MLP | Non-injective | Dot Product | Edge |
| | GMN-Match (Li et al., 2019) | Agg-hinge | Non-injective | Dot Product | Node |
| | **Our-Early-Best** | Set align. | Injective | Hinge | Edge |

Table 5: Choices corresponding to different design axes for state-of-the-art subgraph matching methods.

| | Model | AIDS | Mutag | FM | NCI | MOLT |
|---|---|---|---|---|---|---|
| Late | SimGNN | $0.326 \pm 0.019$ | $0.303 \pm 0.012$ | $0.416 \pm 0.015$ | $0.39 \pm 0.018$ | $0.421 \pm 0.015$ |
| | GraphSim | $0.173 \pm 0.007$ | $0.182 \pm 0.008$ | $0.231 \pm 0.011$ | $0.468 \pm 0.016$ | $0.499 \pm 0.016$ |
| | GOTSim | $0.336 \pm 0.017$ | $0.387 \pm 0.018$ | $0.459 \pm 0.017$ | $0.382 \pm 0.016$ | $0.462 \pm 0.015$ |
| | ERIC | $0.512 \pm 0.022$ | $0.558 \pm 0.027$ | $0.624 \pm 0.019$ | $0.556 \pm 0.018$ | $0.549 \pm 0.016$ |
| | EGSC | $0.5 \pm 0.021$ | $0.446 \pm 0.021$ | $0.643 \pm 0.018$ | $0.528 \pm 0.019$ | $0.555 \pm 0.016$ |
| | GREED | $0.502 \pm 0.022$ | $0.551 \pm 0.026$ | $0.545 \pm 0.019$ | $0.478 \pm 0.019$ | $0.507 \pm 0.017$ |
| | GEN | $0.557 \pm 0.023$ | $0.594 \pm 0.03$ | $0.636 \pm 0.022$ | $0.529 \pm 0.019$ | $0.537 \pm 0.018$ |
| | NeuroMatch | $0.454 \pm 0.025$ | $0.583 \pm 0.027$ | $0.622 \pm 0.019$ | $0.513 \pm 0.018$ | $0.552 \pm 0.017$ |
| | IsoNet | $0.704 \pm 0.021$ | $0.733 \pm 0.023$ | $0.782 \pm 0.017$ | $0.615 \pm 0.019$ | $0.649 \pm 0.016$ |
| | Late-Best | $0.712 \pm 0.018$ | $0.721 \pm 0.025$ | $0.793 \pm 0.016$ | $0.643 \pm 0.02$ | $0.662 \pm 0.016$ |
| Early | H2MN | $0.267 \pm 0.014$ | $0.282 \pm 0.012$ | $0.364 \pm 0.016$ | $0.381 \pm 0.015$ | $0.405 \pm 0.013$ |
| | GMN-Match | $0.609 \pm 0.02$ | $0.693 \pm 0.026$ | $0.686 \pm 0.018$ | $0.588 \pm 0.019$ | $0.603 \pm 0.018$ |
| | Early-Best | $0.817 \pm 0.017$ | $0.837 \pm 0.02$ | $0.887 \pm 0.012$ | $0.677 \pm 0.02$ | $0.71 \pm 0.018$ |

| | Model | FR | MM | MR | MSRC | MCF |
|---|---|---|---|---|---|---|
| Late | SimGNN | $0.355 \pm 0.015$ | $0.358 \pm 0.015$ | $0.308 \pm 0.017$ | $0.237 \pm 0.008$ | $0.141 \pm 0.006$ |
| | GraphSim | $0.165 \pm 0.007$ | $0.2 \pm 0.009$ | $0.216 \pm 0.013$ | $0.348 \pm 0.011$ | $0.43 \pm 0.016$ |
| | GOTSim | $0.361 \pm 0.013$ | $0.417 \pm 0.017$ | $0.430 \pm 0.017$ | $0.240 \pm 0.009$ | $0.352 \pm 0.016$ |
| | ERIC | $0.572 \pm 0.021$ | $0.573 \pm 0.02$ | $0.639 \pm 0.018$ | $0.364 \pm 0.011$ | $0.502 \pm 0.017$ |
| | EGSC | $0.602 \pm 0.02$ | $0.583 \pm 0.019$ | $0.576 \pm 0.02$ | $0.358 \pm 0.011$ | $0.47 \pm 0.019$ |
| | GREED | $0.549 \pm 0.022$ | $0.492 \pm 0.019$ | $0.561 \pm 0.02$ | $0.354 \pm 0.01$ | $0.33 \pm 0.019$ |
| | GEN | $0.577 \pm 0.022$ | $0.579 \pm 0.02$ | $0.618 \pm 0.019$ | $0.385 \pm 0.011$ | $0.492 \pm 0.018$ |
| | NeuroMatch | $0.572 \pm 0.023$ | $0.522 \pm 0.019$ | $0.565 \pm 0.02$ | $0.359 \pm 0.011$ | $0.463 \pm 0.019$ |
| | IsoNet | $0.734 \pm 0.02$ | $0.758 \pm 0.016$ | $0.764 \pm 0.015$ | $0.411 \pm 0.012$ | $0.587 \pm 0.019$ |
| | **Our-Late-Best** | $0.744 \pm 0.019$ | $0.758 \pm 0.015$ | $0.782 \pm 0.014$ | $0.397 \pm 0.013$ | $0.572 \pm 0.02$ |
| Early | H2MN | $0.285 \pm 0.011$ | $0.308 \pm 0.015$ | $0.371 \pm 0.013$ | $0.265 \pm 0.009$ | $0.273 \pm 0.016$ |
| | GMN-Match | $0.667 \pm 0.021$ | $0.627 \pm 0.02$ | $0.683 \pm 0.017$ | $0.416 \pm 0.012$ | $0.549 \pm 0.018$ |
| | **Our-Early-Best** | $0.854 \pm 0.013$ | $0.849 \pm 0.012$ | $0.864 \pm 0.011$ | $0.424 \pm 0.012$ | $0.64 \pm 0.018$ |

Table 6: Comparison of MAP between our best combination of choices across different design axes, for both early (**Our-Early-Best**) and late (**Our-Late-Best**) interaction and state-of-the-art subgraph matching methods. Green, Blue and Yellow cells indicate best early, best late and second-best late interaction methods.

## F.5 EXPLORATION THROUGH ALL POSSIBLE COMBINATIONS IN DESIGN SPACE

In this section, we present the detailed numerical data used to generate the bar plots in Figure 2. Following the analytical framework established in Figure 2, we categorize the runs into four components: Late Node, Late Edge, Early Node, and Early Edge. Within each component, we provide separate tables that group the runs along the three design axes: relevance distance, interaction structure, and interaction nonlinearity. We will analyze each of these tables in turn.

**Analysis of Late Node Interaction along Relevance Distance Design Choices** In Table 28, we observe: **(1)** For eight datasets, one of the networks using the Set align. distance emerges as the best performer indicating its compatibility with late interaction. **(2)** Agg-NTN distance is the second best alternative, winning on two datasets and demonstrating competitive performance across others. **(3)** Agg-hinge & Agg-MLP distances consistently fail to match the performance of the previously mentioned relevance distances in any of the datasets.

| Rel. Dist. | Structure | Non-linearity | AIDS | Mutag | FM | NCI | MOLT |
|---|---|---|---|---|---|---|---|
| Agg-hinge | NA | NA | $0.557 \pm 0.023$ | $0.594 \pm 0.03$ | $0.636 \pm 0.022$ | $0.529 \pm 0.019$ | $0.537 \pm 0.018$ |
| Agg-MLP | NA | NA | $0.548 \pm 0.022$ | $0.64 \pm 0.028$ | $0.566 \pm 0.019$ | $0.575 \pm 0.018$ |
| Agg-NTN | NA | NA | $0.576 \pm 0.023$ | $0.708 \pm 0.027$ | $0.744 \pm 0.019$ | $0.612 \pm 0.019$ | $0.627 \pm 0.018$ |
| Set align. | Non-injective | Dot Product | $0.6 \pm 0.023$ | $0.652 \pm 0.027$ | $0.702 \pm 0.021$ | $0.598 \pm 0.02$ | $0.617 \pm 0.017$ |
| Set align. | Non-injective | Hinge | $0.636 \pm 0.022$ | $0.647 \pm 0.027$ | $0.694 \pm 0.022$ | $0.617 \pm 0.019$ | $0.616 \pm 0.017$ |
| Set align. | Non-injective | Neural | $0.635 \pm 0.022$ | $0.681 \pm 0.026$ | $0.736 \pm 0.017$ | $0.605 \pm 0.019$ | $0.609 \pm 0.017$ |
| Set align. | Injective | Dot Product | $0.631 \pm 0.022$ | $0.66 \pm 0.026$ | $0.722 \pm 0.019$ | $0.608 \pm 0.019$ | $0.612 \pm 0.016$ |
| Set align. | Injective | Hinge | $0.633 \pm 0.021$ | $0.647 \pm 0.028$ | $0.72 \pm 0.019$ | $0.625 \pm 0.019$ | $0.622 \pm 0.017$ |
| Set align. | Injective | Neural | $0.664 \pm 0.021$ | $0.69 \pm 0.026$ | $0.758 \pm 0.017$ | $0.612 \pm 0.019$ | $0.621 \pm 0.017$ |

| Rel. Dist. | Structure | Non-linearity | FR | MM | MR | MSRC | MCF |
|---|---|---|---|---|---|---|---|
| Agg-hinge | NA | NA | $0.577 \pm 0.022$ | $0.579 \pm 0.02$ | $0.618 \pm 0.019$ | $0.385 \pm 0.011$ | $0.492 \pm 0.018$ |
| Agg-MLP | NA | NA | $0.608 \pm 0.023$ | $0.577 \pm 0.02$ | $0.605 \pm 0.018$ | $0.368 \pm 0.011$ | $0.528 \pm 0.02$ |
| Agg-NTN | NA | NA | $0.674 \pm 0.022$ | $0.663 \pm 0.023$ | $0.707 \pm 0.018$ | $0.409 \pm 0.011$ | $0.552 \pm 0.02$ |
| Set align. | Non-injective | Dot Product | $0.621 \pm 0.024$ | $0.671 \pm 0.019$ | $0.701 \pm 0.02$ | $0.403 \pm 0.012$ | $0.56 \pm 0.021$ |
| Set align. | Non-injective | Hinge | $0.645 \pm 0.024$ | $0.634 \pm 0.021$ | $0.701 \pm 0.02$ | $0.392 \pm 0.012$ | $0.577 \pm 0.019$ |
| Set align. | Non-injective | Neural | $0.689 \pm 0.021$ | $0.705 \pm 0.019$ | $0.741 \pm 0.017$ | $0.399 \pm 0.012$ | $0.58 \pm 0.019$ |
| Set align. | Injective | Dot Product | $0.62 \pm 0.023$ | $0.664 \pm 0.019$ | $0.643 \pm 0.019$ | $0.415 \pm 0.012$ | $0.57 \pm 0.019$ |
| Set align. | Injective | Hinge | $0.652 \pm 0.024$ | $0.72 \pm 0.017$ | $0.694 \pm 0.018$ | $0.399 \pm 0.012$ | $0.574 \pm 0.019$ |
| Set align. | Injective | Neural | $0.683 \pm 0.022$ | $0.711 \pm 0.017$ | $0.738 \pm 0.017$ | $0.41 \pm 0.012$ | $0.576 \pm 0.019$ |

Table 7: Comparison of different **relevance distances** for **late interaction** models with **node-level granularity** across ten datasets, using mean average precision (MAP). Green and yellow cells indicate the best and second best methods respectively for the corresponding dataset.

**Analysis of Late Edge Interaction along Relevance Distance Design Choices** In Table 8, we observe: **(1)** Similar to node-granularity networks in Table 28, Set align. distance is the best performing alternative, followed by Agg-NTN. **(2)** Agg-hinge & Agg-MLP distances again demonstrate significantly weaker performance compared to the other alternatives.

| Rel. Dist. | Structure | Non-linearity | AIDS | Mutag | FM | NCI | MOLT |
|---|---|---|---|---|---|---|---|
| Agg-hinge | NA | NA | $0.635 \pm 0.024$ | $0.694 \pm 0.028$ | $0.712 \pm 0.018$ | $0.569 \pm 0.02$ | $0.571 \pm 0.019$ |
| Agg-MLP | NA | NA | $0.607 \pm 0.024$ | $0.63 \pm 0.029$ | $0.727 \pm 0.02$ | $0.604 \pm 0.02$ | $0.613 \pm 0.019$ |
| Agg-NTN | NA | NA | $0.66 \pm 0.026$ | $0.718 \pm 0.027$ | $0.759 \pm 0.019$ | $0.627 \pm 0.019$ | $0.653 \pm 0.019$ |
| Set align. | Non-injective | Dot Product | $0.681 \pm 0.022$ | $0.681 \pm 0.026$ | $0.759 \pm 0.019$ | $0.611 \pm 0.019$ | $0.621 \pm 0.017$ |
| Set align. | Non-injective | Hinge | $0.702 \pm 0.021$ | $0.687 \pm 0.027$ | $0.774 \pm 0.016$ | $0.631 \pm 0.021$ | $0.638 \pm 0.017$ |
| Set align. | Non-injective | Neural | $0.7 \pm 0.022$ | $0.713 \pm 0.025$ | $0.77 \pm 0.017$ | $0.609 \pm 0.02$ | $0.618 \pm 0.017$ |
| Set align. | Injective | Dot Product | $0.68 \pm 0.021$ | $0.712 \pm 0.026$ | $0.755 \pm 0.02$ | $0.623 \pm 0.02$ | $0.634 \pm 0.018$ |
| Set align. | Injective | Hinge | $0.712 \pm 0.018$ | $0.721 \pm 0.025$ | $0.793 \pm 0.016$ | $0.643 \pm 0.02$ | $0.662 \pm 0.016$ |
| Set align. | Injective | Neural | $0.704 \pm 0.021$ | $0.733 \pm 0.023$ | $0.782 \pm 0.017$ | $0.615 \pm 0.019$ | $0.649 \pm 0.016$ |

| Rel. Dist. | Structure | Non-linearity | FR | MM | MR | MSRC | MCF |
|---|---|---|---|---|---|---|---|
| Agg-hinge | NA | NA | $0.666 \pm 0.022$ | $0.665 \pm 0.019$ | $0.709 \pm 0.019$ | $0.389 \pm 0.012$ | $0.51 \pm 0.019$ |
| Agg-MLP | NA | NA | $0.669 \pm 0.022$ | $0.669 \pm 0.02$ | $0.669 \pm 0.02$ | $0.368 \pm 0.011$ | $0.561 \pm 0.019$ |
| Agg-NTN | NA | NA | $0.716 \pm 0.021$ | $0.671 \pm 0.021$ | $0.721 \pm 0.02$ | $0.414 \pm 0.012$ | $0.584 \pm 0.02$ |
| Set align. | Non-injective | Dot Product | $0.693 \pm 0.022$ | $0.702 \pm 0.019$ | $0.723 \pm 0.019$ | $0.411 \pm 0.012$ | $0.571 \pm 0.02$ |
| Set align. | Non-injective | Hinge | $0.731 \pm 0.018$ | $0.735 \pm 0.017$ | $0.776 \pm 0.017$ | $0.412 \pm 0.012$ | $0.609 \pm 0.018$ |
| Set align. | Non-injective | Neural | $0.716 \pm 0.02$ | $0.721 \pm 0.016$ | $0.757 \pm 0.017$ | $0.403 \pm 0.011$ | $0.572 \pm 0.019$ |
| Set align. | Injective | Dot Product | $0.719 \pm 0.02$ | $0.676 \pm 0.018$ | $0.735 \pm 0.018$ | $0.404 \pm 0.012$ | $0.58 \pm 0.019$ |
| Set align. | Injective | Hinge | $0.744 \pm 0.019$ | $0.758 \pm 0.015$ | $0.782 \pm 0.014$ | $0.397 \pm 0.013$ | $0.572 \pm 0.02$ |
| Set align. | Injective | Neural | $0.734 \pm 0.02$ | $0.758 \pm 0.016$ | $0.764 \pm 0.015$ | $0.411 \pm 0.012$ | $0.587 \pm 0.019$ |

Table 8: Comparison of different **relevance distances** for **late interaction** models with **edge-level granularity** across ten datasets, using mean average precision (MAP). Green and yellow cells indicate the best and second best methods respectively for the corresponding dataset.

**Analysis of Early Node Interaction along Relevance Distance Design Choices** In Table 9 and Table 10, we observe: **(1)** The Set align. and Agg-NTN distances dominate in the presence of injective and non-injective networks, respectively. **(2)** The Agg-hinge distance generally performs the worst among the alternatives. **(3)** The Agg-MLP method demonstrates mediocre performance, falling between the former two and the latter across datasets and base networks.

| Relevance Distance ↓ | AIDS | Mutag | FM | NCI | MOLT |
|---|---|---|---|---|---|
| Fixed axes: Stage - Early; Structure - Injective; Non-linearity - Dot Product; Granularity - Node | | | | | |
| Agg-hinge | $0.64 \pm 0.019$ | $0.75 \pm 0.023$ | $0.79 \pm 0.016$ | $0.619 \pm 0.019$ | $0.63 \pm 0.018$ |
| Agg-MLP | $0.658 \pm 0.019$ | $0.768 \pm 0.023$ | $0.806 \pm 0.016$ | $0.641 \pm 0.019$ | $0.649 \pm 0.017$ |
| Agg-NTN | $0.721 \pm 0.019$ | $0.772 \pm 0.022$ | $0.812 \pm 0.016$ | $0.626 \pm 0.019$ | $0.636 \pm 0.018$ |
| Set align. | $0.71 \pm 0.019$ | $0.779 \pm 0.022$ | $0.8 \pm 0.017$ | $0.632 \pm 0.02$ | $0.652 \pm 0.017$ |
| Fixed axes: Stage - Early; Structure - Injective; Non-linearity - Hinge; Granularity - Node | | | | | |
| Agg-hinge | $0.662 \pm 0.019$ | $0.734 \pm 0.025$ | $0.785 \pm 0.017$ | $0.624 \pm 0.02$ | $0.635 \pm 0.018$ |
| Agg-MLP | $0.683 \pm 0.019$ | $0.757 \pm 0.023$ | $0.785 \pm 0.016$ | $0.629 \pm 0.019$ | $0.627 \pm 0.018$ |
| Agg-NTN | $0.743 \pm 0.018$ | $0.773 \pm 0.024$ | $0.805 \pm 0.016$ | $0.615 \pm 0.019$ | $0.629 \pm 0.017$ |
| Set align. | $0.734 \pm 0.019$ | $0.774 \pm 0.023$ | $0.834 \pm 0.016$ | $0.64 \pm 0.019$ | $0.647 \pm 0.018$ |
| Fixed axes: Stage - Early; Structure - Injective; Non-linearity - Neural; Granularity - Node | | | | | |
| Agg-hinge | $0.614 \pm 0.021$ | $0.736 \pm 0.025$ | $0.779 \pm 0.017$ | $0.613 \pm 0.02$ | $0.637 \pm 0.019$ |
| Agg-MLP | $0.629 \pm 0.021$ | $0.764 \pm 0.024$ | $0.777 \pm 0.016$ | $0.626 \pm 0.02$ | $0.641 \pm 0.017$ |
| Agg-NTN | $0.667 \pm 0.02$ | $0.791 \pm 0.021$ | $0.828 \pm 0.015$ | $0.629 \pm 0.019$ | $0.651 \pm 0.018$ |
| Set align. | $0.69 \pm 0.02$ | $0.783 \pm 0.023$ | $0.827 \pm 0.015$ | $0.654 \pm 0.019$ | $0.659 \pm 0.018$ |
| Fixed axes: Stage - Early; Structure - Non-injective; Non-linearity - Dot Product; Granularity - Node | | | | | |
| Agg-hinge | $0.609 \pm 0.02$ | $0.693 \pm 0.026$ | $0.686 \pm 0.018$ | $0.588 \pm 0.019$ | $0.603 \pm 0.018$ |
| Agg-MLP | $0.63 \pm 0.022$ | $0.713 \pm 0.025$ | $0.765 \pm 0.016$ | $0.593 \pm 0.019$ | $0.619 \pm 0.017$ |
| Agg-NTN | $0.669 \pm 0.022$ | $0.742 \pm 0.025$ | $0.75 \pm 0.016$ | $0.602 \pm 0.019$ | $0.628 \pm 0.017$ |
| Set align. | $0.608 \pm 0.022$ | $0.754 \pm 0.024$ | $0.759 \pm 0.017$ | $0.603 \pm 0.019$ | $0.62 \pm 0.017$ |
| Fixed axes: Stage - Early; Structure - Non-injective; Non-linearity - Hinge; Granularity - Node | | | | | |
| Agg-hinge | $0.726 \pm 0.02$ | $0.723 \pm 0.026$ | $0.79 \pm 0.016$ | $0.618 \pm 0.019$ | $0.651 \pm 0.018$ |
| Agg-MLP | $0.637 \pm 0.021$ | $0.725 \pm 0.024$ | $0.808 \pm 0.015$ | $0.63 \pm 0.02$ | $0.636 \pm 0.017$ |
| Agg-NTN | $0.686 \pm 0.019$ | $0.758 \pm 0.022$ | $0.818 \pm 0.015$ | $0.641 \pm 0.02$ | $0.664 \pm 0.017$ |
| Set align. | $0.676 \pm 0.022$ | $0.669 \pm 0.024$ | $0.772 \pm 0.018$ | $0.599 \pm 0.02$ | $0.637 \pm 0.018$ |
| Fixed axes: Stage - Early; Structure - Non-injective; Non-linearity - Neural; Granularity - Node | | | | | |
| Agg-hinge | $0.598 \pm 0.021$ | $0.694 \pm 0.025$ | $0.712 \pm 0.019$ | $0.591 \pm 0.019$ | $0.629 \pm 0.018$ |
| Agg-MLP | $0.629 \pm 0.023$ | $0.715 \pm 0.025$ | $0.769 \pm 0.017$ | $0.623 \pm 0.018$ | $0.627 \pm 0.018$ |
| Agg-NTN | $0.635 \pm 0.025$ | $0.755 \pm 0.025$ | $0.796 \pm 0.016$ | $0.633 \pm 0.019$ | $0.641 \pm 0.017$ |
| Set align. | $0.593 \pm 0.021$ | $0.748 \pm 0.025$ | $0.772 \pm 0.016$ | $0.614 \pm 0.019$ | $0.646 \pm 0.017$ |

Table 9: Comparison of different **relevance distances** for **early interaction** models with **node-level granularity** across first five datasets, using mean average precision (MAP). Green and yellow cells indicate the best and second best methods respectively for the corresponding dataset..

| Relevance Distance ↓ | FR | MM | MR | MSRC | MCF |
|---|---|---|---|---|---|
| Fixed axes: Stage - Early; Structure - Injective; Non-linearity - Dot Product; Granularity - Node | | | | | |
| Agg-hinge | $0.73 \pm 0.019$ | $0.699 \pm 0.019$ | $0.736 \pm 0.016$ | $0.423 \pm 0.012$ | $0.574 \pm 0.018$ |
| Agg-MLP | $0.751 \pm 0.018$ | $0.741 \pm 0.017$ | $0.809 \pm 0.014$ | $0.431 \pm 0.012$ | $0.577 \pm 0.019$ |
| Agg-NTN | $0.764 \pm 0.019$ | $0.748 \pm 0.017$ | $0.796 \pm 0.015$ | $0.426 \pm 0.012$ | $0.583 \pm 0.019$ |
| Set align. | $0.774 \pm 0.017$ | $0.736 \pm 0.017$ | $0.803 \pm 0.014$ | $0.436 \pm 0.012$ | $0.593 \pm 0.019$ |
| Fixed axes: Stage - Early; Structure - Injective; Non-linearity - Hinge; Granularity - Node | | | | | |
| Agg-hinge | $0.733 \pm 0.021$ | $0.713 \pm 0.018$ | $0.752 \pm 0.018$ | $0.403 \pm 0.011$ | $0.575 \pm 0.02$ |
| Agg-MLP | $0.75 \pm 0.019$ | $0.774 \pm 0.017$ | $0.781 \pm 0.016$ | $0.42 \pm 0.011$ | $0.603 \pm 0.019$ |
| Agg-NTN | $0.768 \pm 0.019$ | $0.755 \pm 0.017$ | $0.798 \pm 0.015$ | $0.422 \pm 0.012$ | $0.571 \pm 0.019$ |
| Set align. | $0.774 \pm 0.017$ | $0.759 \pm 0.016$ | $0.806 \pm 0.013$ | $0.426 \pm 0.012$ | $0.584 \pm 0.019$ |
| Fixed axes: Stage - Early; Structure - Injective; Non-linearity - Neural; Granularity - Node | | | | | |
| Agg-hinge | $0.73 \pm 0.02$ | $0.712 \pm 0.018$ | $0.733 \pm 0.017$ | $0.417 \pm 0.011$ | $0.579 \pm 0.019$ |
| Agg-MLP | $0.772 \pm 0.017$ | $0.707 \pm 0.018$ | $0.785 \pm 0.015$ | $0.416 \pm 0.011$ | $0.597 \pm 0.019$ |
| Agg-NTN | $0.766 \pm 0.019$ | $0.748 \pm 0.018$ | $0.8 \pm 0.015$ | $0.409 \pm 0.011$ | $0.584 \pm 0.019$ |
| Set align. | $0.746 \pm 0.019$ | $0.764 \pm 0.016$ | $0.776 \pm 0.015$ | $0.433 \pm 0.012$ | $0.613 \pm 0.019$ |
| Fixed axes: Stage - Early; Structure - Non-injective; Non-linearity - Dot Product; Granularity - Node | | | | | |
| Agg-hinge | $0.667 \pm 0.021$ | $0.627 \pm 0.02$ | $0.683 \pm 0.017$ | $0.416 \pm 0.012$ | $0.549 \pm 0.018$ |
| Agg-MLP | $0.687 \pm 0.02$ | $0.678 \pm 0.018$ | $0.741 \pm 0.017$ | $0.42 \pm 0.011$ | $0.562 \pm 0.019$ |
| Agg-NTN | $0.715 \pm 0.021$ | $0.711 \pm 0.019$ | $0.762 \pm 0.017$ | $0.415 \pm 0.011$ | $0.571 \pm 0.019$ |
| Set align. | $0.706 \pm 0.02$ | $0.691 \pm 0.017$ | $0.715 \pm 0.018$ | $0.423 \pm 0.012$ | $0.553 \pm 0.019$ |
| Fixed axes: Stage - Early; Structure - Non-injective; Non-linearity - Hinge; Granularity - Node | | | | | |
| Agg-hinge | $0.75 \pm 0.018$ | $0.723 \pm 0.018$ | $0.787 \pm 0.015$ | $0.414 \pm 0.012$ | $0.584 \pm 0.02$ |
| Agg-MLP | $0.744 \pm 0.02$ | $0.71 \pm 0.018$ | $0.773 \pm 0.016$ | $0.41 \pm 0.011$ | $0.586 \pm 0.02$ |
| Agg-NTN | $0.765 \pm 0.019$ | $0.766 \pm 0.016$ | $0.8 \pm 0.014$ | $0.409 \pm 0.012$ | $0.572 \pm 0.02$ |
| Set align. | $0.678 \pm 0.022$ | $0.711 \pm 0.018$ | $0.768 \pm 0.017$ | $0.419 \pm 0.012$ | $0.601 \pm 0.02$ |
| Fixed axes: Stage - Early; Structure - Non-injective; Non-linearity - Neural; Granularity - Node | | | | | |
| Agg-hinge | $0.674 \pm 0.022$ | $0.643 \pm 0.02$ | $0.662 \pm 0.02$ | $0.401 \pm 0.011$ | $0.554 \pm 0.019$ |
| Agg-MLP | $0.698 \pm 0.021$ | $0.698 \pm 0.018$ | $0.739 \pm 0.018$ | $0.415 \pm 0.012$ | $0.575 \pm 0.019$ |
| Agg-NTN | $0.707 \pm 0.021$ | $0.701 \pm 0.019$ | $0.69 \pm 0.019$ | $0.408 \pm 0.011$ | $0.586 \pm 0.018$ |
| Set align. | $0.719 \pm 0.02$ | $0.703 \pm 0.019$ | $0.742 \pm 0.016$ | $0.416 \pm 0.012$ | $0.579 \pm 0.018$ |

Table 10: Comparison of different **relevance distances** for **early interaction** models with **node-level granularity** across last five datasets, using mean average precision (MAP). Green and yellow cells indicate the best and second best methods respectively for the corresponding dataset..

**Analysis of Early Edge Interaction along Relevance Distance Design Choices**  In Table 11 and Table 12, we observe: **(1)** Set align. consistently outperforms Agg-NTN, regardless of the interaction structure. The only exception is found in non-injective networks utilizing Hinge non-linearity, where their performances are comparable. **(2)** Notably, the top-performing early interaction network identified in our study, which surpasses all others across nine datasets, employs Set align. as its relevance distance in conjunction with Hinge non-linearity.

| Relevance Distance ↓ \| | AIDS | Mutag | FM | NCI | MOLT |
|---|---|---|---|---|---|
| Fixed axes: Stage - Early; Structure - Injective; Non-linearity - Dot Product; Granularity - Edge | | | | | |
| Agg-hinge | $0.755 \pm 0.019$ | $0.763 \pm 0.025$ | $0.826 \pm 0.014$ | $0.656 \pm 0.02$ | $0.67 \pm 0.019$ |
| Agg-MLP | $0.758 \pm 0.019$ | $0.773 \pm 0.023$ | $0.853 \pm 0.014$ | $0.662 \pm 0.02$ | $0.683 \pm 0.017$ |
| Agg-NTN | $0.74 \pm 0.02$ | $0.783 \pm 0.023$ | $0.833 \pm 0.015$ | $0.655 \pm 0.02$ | $0.672 \pm 0.018$ |
| Set align. | $0.798 \pm 0.019$ | $0.789 \pm 0.023$ | $0.862 \pm 0.015$ | $0.674 \pm 0.019$ | $0.688 \pm 0.018$ |
| Fixed axes: Stage - Early; Structure - Injective; Non-linearity - Hinge; Granularity - Edge | | | | | |
| Agg-hinge | $0.758 \pm 0.019$ | $0.78 \pm 0.023$ | $0.83 \pm 0.015$ | $0.661 \pm 0.02$ | $0.681 \pm 0.019$ |
| Agg-MLP | $0.789 \pm 0.017$ | $0.807 \pm 0.023$ | $0.871 \pm 0.013$ | $0.663 \pm 0.02$ | $0.678 \pm 0.018$ |
| Agg-NTN | $0.76 \pm 0.019$ | $0.766 \pm 0.025$ | $0.855 \pm 0.013$ | $0.653 \pm 0.021$ | $0.679 \pm 0.018$ |
| Set align. | $0.817 \pm 0.017$ | $0.837 \pm 0.02$ | $0.887 \pm 0.012$ | $0.677 \pm 0.02$ | $0.71 \pm 0.018$ |
| Fixed axes: Stage - Early; Structure - Injective; Non-linearity - Neural; Granularity - Edge | | | | | |
| Agg-hinge | $0.68 \pm 0.022$ | $0.755 \pm 0.024$ | $0.807 \pm 0.016$ | $0.625 \pm 0.02$ | $0.656 \pm 0.019$ |
| Agg-MLP | $0.703 \pm 0.021$ | $0.738 \pm 0.026$ | $0.815 \pm 0.015$ | $0.643 \pm 0.021$ | $0.644 \pm 0.019$ |
| Agg-NTN | $0.689 \pm 0.022$ | $0.749 \pm 0.027$ | $0.831 \pm 0.015$ | $0.651 \pm 0.021$ | $0.664 \pm 0.019$ |
| Set align. | $0.725 \pm 0.021$ | $0.784 \pm 0.023$ | $0.837 \pm 0.015$ | $0.649 \pm 0.019$ | $0.664 \pm 0.018$ |
| Fixed axes: Stage - Early; Structure - Non-injective; Non-linearity - Dot Product; Granularity - Edge | | | | | |
| Agg-hinge | $0.7 \pm 0.022$ | $0.734 \pm 0.025$ | $0.78 \pm 0.018$ | $0.625 \pm 0.02$ | $0.653 \pm 0.018$ |
| Agg-MLP | $0.677 \pm 0.022$ | $0.766 \pm 0.023$ | $0.792 \pm 0.016$ | $0.638 \pm 0.019$ | $0.65 \pm 0.018$ |
| Agg-NTN | $0.71 \pm 0.022$ | $0.751 \pm 0.024$ | $0.815 \pm 0.016$ | $0.653 \pm 0.02$ | $0.644 \pm 0.017$ |
| Set align. | $0.715 \pm 0.021$ | $0.779 \pm 0.023$ | $0.812 \pm 0.017$ | $0.635 \pm 0.019$ | $0.659 \pm 0.017$ |
| Fixed axes: Stage - Early; Structure - Non-injective; Non-linearity - Hinge; Granularity - Edge | | | | | |
| Agg-hinge | $0.763 \pm 0.018$ | $0.753 \pm 0.025$ | $0.831 \pm 0.015$ | $0.666 \pm 0.02$ | $0.686 \pm 0.017$ |
| Agg-MLP | $0.748 \pm 0.019$ | $0.778 \pm 0.023$ | $0.847 \pm 0.014$ | $0.658 \pm 0.021$ | $0.684 \pm 0.018$ |
| Agg-NTN | $0.79 \pm 0.017$ | $0.785 \pm 0.023$ | $0.835 \pm 0.014$ | $0.661 \pm 0.02$ | $0.683 \pm 0.018$ |
| Set align. | $0.783 \pm 0.017$ | $0.785 \pm 0.024$ | $0.807 \pm 0.015$ | $0.657 \pm 0.021$ | $0.696 \pm 0.018$ |
| Fixed axes: Stage - Early; Structure - Non-injective; Non-linearity - Neural; Granularity - Edge | | | | | |
| Agg-hinge | $0.677 \pm 0.022$ | $0.729 \pm 0.025$ | $0.747 \pm 0.019$ | $0.618 \pm 0.019$ | $0.633 \pm 0.018$ |
| Agg-MLP | $0.662 \pm 0.024$ | $0.735 \pm 0.024$ | $0.745 \pm 0.019$ | $0.629 \pm 0.02$ | $0.65 \pm 0.018$ |
| Agg-NTN | $0.701 \pm 0.023$ | $0.734 \pm 0.028$ | $0.811 \pm 0.016$ | $0.625 \pm 0.02$ | $0.648 \pm 0.019$ |
| Set align. | $0.708 \pm 0.022$ | $0.763 \pm 0.024$ | $0.807 \pm 0.016$ | $0.648 \pm 0.019$ | $0.655 \pm 0.018$ |

Table 11: Comparison of different **relevance distances** for **early interaction** models with **edge-level granularity** across **first five datasets**, using mean average precision (MAP). Green and yellow cells indicate the best and second best methods respectively for the corresponding dataset.

| Relevance Distance ↓ | FR | MM | MR | MSRC | MCF |
|---|---|---|---|---|---|
| Fixed axes: Stage - Early; Structure - Injective; Non-linearity - Dot Product; Granularity - Edge | | | | | |
| Agg-hinge | 0.8 ± 0.018 | 0.777 ± 0.018 | 0.803 ± 0.015 | 0.415 ± 0.011 | 0.616 ± 0.018 |
| Agg-MLP | 0.786 ± 0.017 | 0.791 ± 0.016 | 0.827 ± 0.014 | 0.404 ± 0.011 | 0.614 ± 0.018 |
| Agg-NTN | 0.771 ± 0.018 | 0.81 ± 0.014 | 0.858 ± 0.012 | 0.425 ± 0.012 | 0.617 ± 0.018 |
| Set align. | 0.817 ± 0.016 | 0.791 ± 0.016 | 0.819 ± 0.014 | 0.435 ± 0.012 | 0.63 ± 0.018 |
| Fixed axes: Stage - Early; Structure - Injective; Non-linearity - Hinge; Granularity - Edge | | | | | |
| Agg-hinge | 0.798 ± 0.019 | 0.785 ± 0.016 | 0.817 ± 0.014 | 0.421 ± 0.012 | 0.619 ± 0.019 |
| Agg-MLP | 0.815 ± 0.016 | 0.833 ± 0.014 | 0.851 ± 0.011 | 0.423 ± 0.012 | 0.599 ± 0.018 |
| Agg-NTN | 0.824 ± 0.016 | 0.821 ± 0.014 | 0.844 ± 0.013 | 0.426 ± 0.012 | 0.635 ± 0.019 |
| Set align. | 0.854 ± 0.013 | 0.849 ± 0.012 | 0.864 ± 0.011 | 0.424 ± 0.012 | 0.64 ± 0.018 |
| Fixed axes: Stage - Early; Structure - Injective; Non-linearity - Neural; Granularity - Edge | | | | | |
| Agg-hinge | 0.749 ± 0.022 | 0.705 ± 0.019 | 0.766 ± 0.016 | 0.406 ± 0.012 | 0.595 ± 0.018 |
| Agg-MLP | 0.771 ± 0.019 | 0.717 ± 0.018 | 0.795 ± 0.016 | 0.401 ± 0.011 | 0.601 ± 0.018 |
| Agg-NTN | 0.762 ± 0.02 | 0.767 ± 0.018 | 0.802 ± 0.016 | 0.411 ± 0.012 | 0.593 ± 0.018 |
| Set align. | 0.752 ± 0.02 | 0.773 ± 0.016 | 0.788 ± 0.015 | 0.433 ± 0.012 | 0.609 ± 0.02 |
| Fixed axes: Stage - Early; Structure - Non-injective; Non-linearity - Dot Product; Granularity - Edge | | | | | |
| Agg-hinge | 0.727 ± 0.019 | 0.706 ± 0.02 | 0.746 ± 0.019 | 0.421 ± 0.012 | 0.601 ± 0.018 |
| Agg-MLP | 0.751 ± 0.019 | 0.723 ± 0.019 | 0.788 ± 0.016 | 0.416 ± 0.011 | 0.596 ± 0.02 |
| Agg-NTN | 0.754 ± 0.019 | 0.719 ± 0.021 | 0.777 ± 0.016 | 0.418 ± 0.012 | 0.614 ± 0.018 |
| Set align. | 0.799 ± 0.017 | 0.789 ± 0.016 | 0.81 ± 0.016 | 0.421 ± 0.012 | 0.609 ± 0.02 |
| Fixed axes: Stage - Early; Structure - Non-injective; Non-linearity - Hinge; Granularity - Edge | | | | | |
| Agg-hinge | 0.772 ± 0.018 | 0.799 ± 0.015 | 0.815 ± 0.014 | 0.423 ± 0.012 | 0.612 ± 0.019 |
| Agg-MLP | 0.756 ± 0.019 | 0.787 ± 0.015 | 0.786 ± 0.015 | 0.416 ± 0.012 | 0.628 ± 0.019 |
| Agg-NTN | 0.791 ± 0.017 | 0.8 ± 0.016 | 0.833 ± 0.014 | 0.421 ± 0.011 | 0.608 ± 0.02 |
| Set align. | 0.773 ± 0.019 | 0.814 ± 0.015 | 0.795 ± 0.016 | 0.435 ± 0.012 | 0.626 ± 0.02 |
| Fixed axes: Stage - Early; Structure - Non-injective; Non-linearity - Neural; Granularity - Edge | | | | | |
| Agg-hinge | 0.704 ± 0.022 | 0.691 ± 0.02 | 0.722 ± 0.02 | 0.422 ± 0.011 | 0.589 ± 0.019 |
| Agg-MLP | 0.74 ± 0.02 | 0.687 ± 0.021 | 0.732 ± 0.02 | 0.416 ± 0.011 | 0.603 ± 0.018 |
| Agg-NTN | 0.723 ± 0.022 | 0.704 ± 0.02 | 0.753 ± 0.018 | 0.419 ± 0.012 | 0.584 ± 0.02 |
| Set align. | 0.733 ± 0.02 | 0.75 ± 0.018 | 0.782 ± 0.017 | 0.421 ± 0.011 | 0.607 ± 0.018 |

Table 12: Comparison of different **relevance distances** for **early interaction** models with **edge-level granularity** across **last five datasets**, using mean average precision (MAP). Green and yellow cells indicate the best and second best methods respectively for the corresponding dataset.

**Analysis of Late Node Interaction along Interaction Structure Design Choices** In Table 13, we observe: **(1)**The Injective structure is the preferred option across all non-linearities, achieving victories in more than five datasets in each instance. **(2)**The Non-injective structure is also highly competitive and, in some cases, significantly outperforms Injective, particularly evident in the MR under the Dot Product non-linearity.

| Interaction Structure ↓ | AIDS | Mutag | FM | NCI | MOLT |
|---|---|---|---|---|---|
| Fixed axes: Rel. Dist. - Set align.; Stage - Late; Non-linearity - Dot Product; Granularity - Node | | | | | |
| Non-injective | $0.6 \pm 0.023$ | $0.652 \pm 0.027$ | $0.702 \pm 0.021$ | $0.598 \pm 0.02$ | $0.617 \pm 0.017$ |
| Injective | $0.631 \pm 0.022$ | $0.66 \pm 0.026$ | $0.722 \pm 0.019$ | $0.608 \pm 0.019$ | $0.612 \pm 0.016$ |
| Fixed axes: Rel. Dist. - Set align.; Stage - Late; Non-linearity - Hinge; Granularity - Node | | | | | |
| Non-injective | $0.636 \pm 0.022$ | $0.647 \pm 0.027$ | $0.694 \pm 0.022$ | $0.617 \pm 0.019$ | $0.616 \pm 0.017$ |
| Injective | $0.633 \pm 0.021$ | $0.647 \pm 0.028$ | $0.72 \pm 0.019$ | $0.625 \pm 0.019$ | $0.622 \pm 0.017$ |
| Fixed axes: Rel. Dist. - Set align.; Stage - Late; Non-linearity - Neural; Granularity - Node | | | | | |
| Non-injective | $0.635 \pm 0.022$ | $0.681 \pm 0.026$ | $0.736 \pm 0.017$ | $0.605 \pm 0.019$ | $0.609 \pm 0.017$ |
| Injective | $0.664 \pm 0.021$ | $0.69 \pm 0.026$ | $0.758 \pm 0.017$ | $0.612 \pm 0.019$ | $0.621 \pm 0.017$ |

| Interaction Structure ↓ | FR | MM | MR | MSRC | MCF |
|---|---|---|---|---|---|
| Fixed axes: Rel. Dist. - Set align.; Stage - Late; Non-linearity - Dot Product; Granularity - Node | | | | | |
| Non-injective | $0.621 \pm 0.024$ | $0.671 \pm 0.019$ | $0.681 \pm 0.02$ | $0.403 \pm 0.012$ | $0.56 \pm 0.021$ |
| Injective | $0.62 \pm 0.023$ | $0.664 \pm 0.019$ | $0.643 \pm 0.019$ | $0.415 \pm 0.012$ | $0.57 \pm 0.019$ |
| Fixed axes: Rel. Dist. - Set align.; Stage - Late; Non-linearity - Hinge; Granularity - Node | | | | | |
| Non-injective | $0.645 \pm 0.024$ | $0.634 \pm 0.021$ | $0.701 \pm 0.02$ | $0.392 \pm 0.012$ | $0.577 \pm 0.019$ |
| Injective | $0.652 \pm 0.024$ | $0.72 \pm 0.017$ | $0.694 \pm 0.018$ | $0.399 \pm 0.012$ | $0.574 \pm 0.019$ |
| Fixed axes: Rel. Dist. - Set align.; Stage - Late; Non-linearity - Neural; Granularity - Node | | | | | |
| Non-injective | $0.689 \pm 0.021$ | $0.705 \pm 0.019$ | $0.741 \pm 0.017$ | $0.399 \pm 0.012$ | $0.58 \pm 0.019$ |
| Injective | $0.683 \pm 0.022$ | $0.711 \pm 0.017$ | $0.738 \pm 0.017$ | $0.41 \pm 0.012$ | $0.576 \pm 0.019$ |

Table 13: Comparison of different **interaction structures** for **late interaction** models with **node-level granularity** across ten datasets, using mean average precision (MAP). Green cells indicate the best methods respectively for the corresponding datasets.

**Analysis of Late Edge Interaction along Interaction Structure Design Choices** In Table 14, we observe: **(1)** The Injective structure continues to demonstrate superior performance, with one of its models utilizing Hinge non-linearity emerging as the best-performing late interaction network in our study. **(2)** While the Non-injective structure remains competitive, its effectiveness relative to Injective diminishes as we transition from Dot Product to Hinge and finally to Neural, ultimately resulting in no datasets won.

| Interaction Structure ↓ | AIDS | Mutag | FM | NCI | MOLT |
|---|---|---|---|---|---|
| Fixed axes: Rel. Dist. - Set align.; Stage - Late; Non-linearity - Dot Product; Granularity - Edge | | | | | |
| Non-injective | $0.681 \pm 0.022$ | $0.681 \pm 0.026$ | $0.759 \pm 0.019$ | $0.611 \pm 0.019$ | $0.621 \pm 0.017$ |
| Injective | $0.68 \pm 0.021$ | $0.712 \pm 0.026$ | $0.755 \pm 0.016$ | $0.623 \pm 0.02$ | $0.634 \pm 0.018$ |
| Fixed axes: Rel. Dist. - Set align.; Stage - Late; Non-linearity - Hinge; Granularity - Edge | | | | | |
| Non-injective | $0.702 \pm 0.021$ | $0.687 \pm 0.027$ | $0.774 \pm 0.016$ | $0.631 \pm 0.021$ | $0.638 \pm 0.017$ |
| Injective | $0.712 \pm 0.018$ | $0.721 \pm 0.025$ | $0.793 \pm 0.016$ | $0.643 \pm 0.02$ | $0.662 \pm 0.016$ |
| Fixed axes: Rel. Dist. - Set align.; Stage - Late; Non-linearity - Neural; Granularity - Edge | | | | | |
| Non-injective | $0.7 \pm 0.022$ | $0.713 \pm 0.025$ | $0.77 \pm 0.017$ | $0.609 \pm 0.02$ | $0.618 \pm 0.017$ |
| Injective | $0.704 \pm 0.021$ | $0.733 \pm 0.023$ | $0.782 \pm 0.017$ | $0.615 \pm 0.019$ | $0.649 \pm 0.016$ |

| Interaction Structure ↓ | FR | MM | MR | MSRC | MCF |
|---|---|---|---|---|---|
| Fixed axes: Rel. Dist. - Set align.; Stage - Late; Non-linearity - Dot Product; Granularity - Edge | | | | | |
| Non-injective | $0.693 \pm 0.022$ | $0.702 \pm 0.019$ | $0.723 \pm 0.019$ | $0.411 \pm 0.012$ | $0.571 \pm 0.02$ |
| Injective | $0.719 \pm 0.02$ | $0.676 \pm 0.018$ | $0.735 \pm 0.018$ | $0.404 \pm 0.012$ | $0.58 \pm 0.019$ |
| Fixed axes: Rel. Dist. - Set align.; Stage - Late; Non-linearity - Hinge; Granularity - Edge | | | | | |
| Non-injective | $0.731 \pm 0.018$ | $0.735 \pm 0.017$ | $0.776 \pm 0.017$ | $0.412 \pm 0.012$ | $0.609 \pm 0.018$ |
| Injective | $0.744 \pm 0.019$ | $0.758 \pm 0.015$ | $0.782 \pm 0.014$ | $0.397 \pm 0.013$ | $0.572 \pm 0.02$ |
| Fixed axes: Rel. Dist. - Set align.; Stage - Late; Non-linearity - Neural; Granularity - Edge | | | | | |
| Non-injective | $0.716 \pm 0.02$ | $0.721 \pm 0.016$ | $0.757 \pm 0.017$ | $0.403 \pm 0.011$ | $0.572 \pm 0.019$ |
| Injective | $0.734 \pm 0.02$ | $0.758 \pm 0.016$ | $0.764 \pm 0.015$ | $0.411 \pm 0.012$ | $0.587 \pm 0.019$ |

Table 14: Comparison of different **interaction structures** for **late interaction** models with **edge-level granularity** across ten datasets, using mean average precision (MAP). Green cells indicate the best methods respectively for the corresponding datasets.

**Analysis of Early Node Interaction along Interaction Structure Design Choices** In Tables 15 and 16, we observe: Networks with injective structures consistently outperform those with non-injective structures, with a few notable exceptions. **(1)** The relevance distance using Agg-hinge with non-linearity Hinge wins in eight datasets. **(2)** The relevance distance using Agg-NTN with non-linearity Hinge wins in six datasets. **(3)** The relevance distance using Agg-MLP with non-linearity Hinge wins in three datasets. These observations lead us to conclude that non-injective structures tend to perform well when paired with Hinge non-linearity.

| Interaction Structure ↓ | AIDS | Mutag | FM | NCI | MOLT |
|---|---|---|---|---|---|
| Fixed axes: Rel. Dist. - Agg-hinge; Stage - Early; Non-linearity - Dot Product; Granularity - Node | | | | | |
| Non-injective | $0.609 \pm 0.02$ | $0.693 \pm 0.026$ | $0.686 \pm 0.018$ | $0.588 \pm 0.019$ | $0.603 \pm 0.018$ |
| Injective | $0.64 \pm 0.019$ | $0.75 \pm 0.023$ | $0.79 \pm 0.016$ | $0.619 \pm 0.019$ | $0.63 \pm 0.018$ |
| Fixed axes: Rel. Dist. - Agg-hinge; Stage - Early; Non-linearity - Hinge; Granularity - Node | | | | | |
| Non-injective | $0.726 \pm 0.02$ | $0.723 \pm 0.026$ | $0.79 \pm 0.016$ | $0.618 \pm 0.019$ | $0.651 \pm 0.018$ |
| Injective | $0.662 \pm 0.019$ | $0.734 \pm 0.025$ | $0.785 \pm 0.017$ | $0.624 \pm 0.02$ | $0.635 \pm 0.018$ |
| Fixed axes: Rel. Dist. - Agg-hinge; Stage - Early; Non-linearity - Neural; Granularity - Node | | | | | |
| Non-injective | $0.598 \pm 0.021$ | $0.694 \pm 0.025$ | $0.712 \pm 0.019$ | $0.591 \pm 0.019$ | $0.629 \pm 0.018$ |
| Injective | $0.614 \pm 0.021$ | $0.736 \pm 0.025$ | $0.779 \pm 0.017$ | $0.613 \pm 0.02$ | $0.637 \pm 0.019$ |
| Fixed axes: Rel. Dist. - Agg-MLP; Stage - Early; Non-linearity - Dot Product; Granularity - Node | | | | | |
| Non-injective | $0.63 \pm 0.022$ | $0.713 \pm 0.025$ | $0.765 \pm 0.016$ | $0.593 \pm 0.019$ | $0.619 \pm 0.017$ |
| Injective | $0.658 \pm 0.019$ | $0.768 \pm 0.023$ | $0.806 \pm 0.016$ | $0.641 \pm 0.019$ | $0.649 \pm 0.017$ |
| Fixed axes: Rel. Dist. - Agg-MLP; Stage - Early; Non-linearity - Hinge; Granularity - Node | | | | | |
| Non-injective | $0.637 \pm 0.021$ | $0.725 \pm 0.024$ | $0.808 \pm 0.015$ | $0.63 \pm 0.02$ | $0.636 \pm 0.017$ |
| Injective | $0.683 \pm 0.019$ | $0.757 \pm 0.023$ | $0.785 \pm 0.016$ | $0.629 \pm 0.019$ | $0.627 \pm 0.018$ |
| Fixed axes: Rel. Dist. - Agg-MLP; Stage - Early; Non-linearity - Neural; Granularity - Node | | | | | |
| Non-injective | $0.629 \pm 0.023$ | $0.715 \pm 0.025$ | $0.769 \pm 0.017$ | $0.623 \pm 0.018$ | $0.627 \pm 0.018$ |
| Injective | $0.629 \pm 0.021$ | $0.764 \pm 0.024$ | $0.777 \pm 0.016$ | $0.626 \pm 0.02$ | $0.641 \pm 0.017$ |
| Fixed axes: Rel. Dist. - Set align.; Stage - Early; Non-linearity - Dot Product; Granularity - Node | | | | | |
| Non-injective | $0.608 \pm 0.022$ | $0.754 \pm 0.024$ | $0.759 \pm 0.016$ | $0.603 \pm 0.019$ | $0.62 \pm 0.017$ |
| Injective | $0.71 \pm 0.019$ | $0.779 \pm 0.022$ | $0.8 \pm 0.017$ | $0.632 \pm 0.02$ | $0.652 \pm 0.017$ |
| Fixed axes: Rel. Dist. - Set align.; Stage - Early; Non-linearity - Hinge; Granularity - Node | | | | | |
| Non-injective | $0.676 \pm 0.022$ | $0.669 \pm 0.024$ | $0.772 \pm 0.018$ | $0.599 \pm 0.02$ | $0.637 \pm 0.018$ |
| Injective | $0.734 \pm 0.019$ | $0.774 \pm 0.023$ | $0.834 \pm 0.016$ | $0.64 \pm 0.019$ | $0.647 \pm 0.018$ |
| Fixed axes: Rel. Dist. - Set align.; Stage - Early; Non-linearity - Neural; Granularity - Node | | | | | |
| Non-injective | $0.593 \pm 0.021$ | $0.748 \pm 0.025$ | $0.772 \pm 0.016$ | $0.614 \pm 0.019$ | $0.646 \pm 0.017$ |
| Injective | $0.69 \pm 0.02$ | $0.783 \pm 0.023$ | $0.827 \pm 0.015$ | $0.654 \pm 0.019$ | $0.659 \pm 0.018$ |
| Fixed axes: Rel. Dist. - Agg-NTN; Stage - Early; Non-linearity - Dot Product; Granularity - Node | | | | | |
| Non-injective | $0.669 \pm 0.022$ | $0.742 \pm 0.025$ | $0.75 \pm 0.016$ | $0.602 \pm 0.019$ | $0.628 \pm 0.017$ |
| Injective | $0.721 \pm 0.019$ | $0.772 \pm 0.022$ | $0.812 \pm 0.016$ | $0.626 \pm 0.019$ | $0.636 \pm 0.018$ |
| Fixed axes: Rel. Dist. - Agg-NTN; Stage - Early; Non-linearity - Hinge; Granularity - Node | | | | | |
| Non-injective | $0.686 \pm 0.019$ | $0.758 \pm 0.022$ | $0.818 \pm 0.015$ | $0.641 \pm 0.02$ | $0.664 \pm 0.017$ |
| Injective | $0.743 \pm 0.018$ | $0.773 \pm 0.024$ | $0.805 \pm 0.016$ | $0.615 \pm 0.019$ | $0.629 \pm 0.017$ |
| Fixed axes: Rel. Dist. - Agg-NTN; Stage - Early; Non-linearity - Neural; Granularity - Node | | | | | |
| Non-injective | $0.635 \pm 0.025$ | $0.755 \pm 0.025$ | $0.796 \pm 0.016$ | $0.633 \pm 0.019$ | $0.641 \pm 0.017$ |
| Injective | $0.667 \pm 0.02$ | $0.791 \pm 0.021$ | $0.828 \pm 0.015$ | $0.629 \pm 0.019$ | $0.651 \pm 0.018$ |

Table 15: Comparison of different **interaction structures** for **early interaction** models with **node-level granularity** across **first five datasets**, using mean average precision (MAP). Green cells indicate the best method for the corresponding datasets.

| Interaction Structure ↓ | FR | MM | MR | MSRC | MCF |
|---|---|---|---|---|---|
| Fixed axes: Rel. Dist. - Agg-hinge; Stage - Early; Non-linearity - Dot Product; Granularity - Node | | | | | |
| Non-injective | $0.667 \pm 0.021$ | $0.627 \pm 0.02$ | $0.683 \pm 0.017$ | $0.416 \pm 0.012$ | $0.549 \pm 0.018$ |
| Injective | $0.73 \pm 0.019$ | $0.699 \pm 0.019$ | $0.736 \pm 0.016$ | $0.423 \pm 0.012$ | $0.574 \pm 0.018$ |
| Fixed axes: Rel. Dist. - Agg-hinge; Stage - Early; Non-linearity - Hinge; Granularity - Node | | | | | |
| Non-injective | $0.75 \pm 0.018$ | $0.723 \pm 0.018$ | $0.787 \pm 0.015$ | $0.414 \pm 0.012$ | $0.584 \pm 0.02$ |
| Injective | $0.733 \pm 0.021$ | $0.713 \pm 0.018$ | $0.752 \pm 0.018$ | $0.403 \pm 0.011$ | $0.575 \pm 0.02$ |
| Fixed axes: Rel. Dist. - Agg-hinge; Stage - Early; Non-linearity - Neural; Granularity - Node | | | | | |
| Non-injective | $0.674 \pm 0.022$ | $0.643 \pm 0.02$ | $0.662 \pm 0.02$ | $0.401 \pm 0.011$ | $0.554 \pm 0.019$ |
| Injective | $0.73 \pm 0.02$ | $0.712 \pm 0.018$ | $0.733 \pm 0.017$ | $0.417 \pm 0.011$ | $0.579 \pm 0.019$ |
| Fixed axes: Rel. Dist. - Agg-MLP; Stage - Early; Non-linearity - Dot Product; Granularity - Node | | | | | |
| Non-injective | $0.687 \pm 0.02$ | $0.678 \pm 0.018$ | $0.741 \pm 0.017$ | $0.42 \pm 0.011$ | $0.562 \pm 0.019$ |
| Injective | $0.751 \pm 0.018$ | $0.741 \pm 0.017$ | $0.809 \pm 0.014$ | $0.431 \pm 0.012$ | $0.577 \pm 0.019$ |
| Fixed axes: Rel. Dist. - Agg-MLP; Stage - Early; Non-linearity - Hinge; Granularity - Node | | | | | |
| Non-injective | $0.744 \pm 0.02$ | $0.71 \pm 0.018$ | $0.773 \pm 0.016$ | $0.41 \pm 0.011$ | $0.586 \pm 0.02$ |
| Injective | $0.75 \pm 0.019$ | $0.774 \pm 0.017$ | $0.781 \pm 0.016$ | $0.42 \pm 0.011$ | $0.603 \pm 0.019$ |
| Fixed axes: Rel. Dist. - Agg-MLP; Stage - Early; Non-linearity - Neural; Granularity - Node | | | | | |
| Non-injective | $0.698 \pm 0.021$ | $0.698 \pm 0.018$ | $0.739 \pm 0.018$ | $0.415 \pm 0.012$ | $0.575 \pm 0.019$ |
| Injective | $0.772 \pm 0.017$ | $0.707 \pm 0.018$ | $0.785 \pm 0.015$ | $0.416 \pm 0.011$ | $0.597 \pm 0.019$ |
| Fixed axes: Rel. Dist. - Set align.; Stage - Early; Non-linearity - Dot Product; Granularity - Node | | | | | |
| Non-injective | $0.706 \pm 0.02$ | $0.691 \pm 0.017$ | $0.715 \pm 0.019$ | $0.423 \pm 0.012$ | $0.553 \pm 0.019$ |
| Injective | $0.774 \pm 0.017$ | $0.736 \pm 0.017$ | $0.803 \pm 0.014$ | $0.436 \pm 0.012$ | $0.593 \pm 0.019$ |
| Fixed axes: Rel. Dist. - Set align.; Stage - Early; Non-linearity - Hinge; Granularity - Node | | | | | |
| Non-injective | $0.678 \pm 0.022$ | $0.711 \pm 0.018$ | $0.768 \pm 0.017$ | $0.419 \pm 0.012$ | $0.601 \pm 0.02$ |
| Injective | $0.774 \pm 0.017$ | $0.759 \pm 0.016$ | $0.806 \pm 0.013$ | $0.426 \pm 0.012$ | $0.584 \pm 0.019$ |
| Fixed axes: Rel. Dist. - Set align.; Stage - Early; Non-linearity - Neural; Granularity - Node | | | | | |
| Non-injective | $0.719 \pm 0.02$ | $0.703 \pm 0.019$ | $0.742 \pm 0.016$ | $0.416 \pm 0.012$ | $0.579 \pm 0.018$ |
| Injective | $0.746 \pm 0.019$ | $0.764 \pm 0.016$ | $0.776 \pm 0.015$ | $0.433 \pm 0.012$ | $0.613 \pm 0.019$ |
| Fixed axes: Rel. Dist. - Agg-NTN; Stage - Early; Non-linearity - Dot Product; Granularity - Node | | | | | |
| Non-injective | $0.715 \pm 0.021$ | $0.711 \pm 0.019$ | $0.762 \pm 0.017$ | $0.415 \pm 0.011$ | $0.571 \pm 0.019$ |
| Injective | $0.764 \pm 0.019$ | $0.748 \pm 0.017$ | $0.796 \pm 0.015$ | $0.426 \pm 0.012$ | $0.583 \pm 0.019$ |
| Fixed axes: Rel. Dist. - Agg-NTN; Stage - Early; Non-linearity - Hinge; Granularity - Node | | | | | |
| Non-injective | $0.765 \pm 0.019$ | $0.766 \pm 0.016$ | $0.8 \pm 0.014$ | $0.409 \pm 0.012$ | $0.572 \pm 0.02$ |
| Injective | $0.768 \pm 0.019$ | $0.755 \pm 0.017$ | $0.798 \pm 0.015$ | $0.422 \pm 0.012$ | $0.571 \pm 0.019$ |
| Fixed axes: Rel. Dist. - Agg-NTN; Stage - Early; Non-linearity - Neural; Granularity - Node | | | | | |
| Non-injective | $0.707 \pm 0.021$ | $0.701 \pm 0.019$ | $0.69 \pm 0.019$ | $0.408 \pm 0.011$ | $0.586 \pm 0.018$ |
| Injective | $0.766 \pm 0.019$ | $0.748 \pm 0.018$ | $0.8 \pm 0.015$ | $0.409 \pm 0.011$ | $0.584 \pm 0.019$ |

Table 16: Comparison of different **interaction structures** for **early interaction** models with **node-level granularity** across **last five datasets**, using mean average precision (MAP). Green cells indicate the best method for the corresponding datasets.

**Analysis of Early Edge Interaction along Interaction Structure Design Choices**    In Tables 17 and 18, we observe: **(1)** Similar to node granularity, several non-injective networks either match or surpass the performance of their injective counterparts, primarily due to the Hinge non-linearity. For instance, with relevance distance Agg-hinge and non-linearity Hinge, this configuration wins in six datasets. **(2)** Additionally, with relevance distance Agg-NTN and non-linearity Hinge, this combination wins in four datasets.

| Interaction Structure ↓ | AIDS | Mutag | FM | NCI | MOLT |
|---|---|---|---|---|---|
| Fixed axes: Rel. Dist. - Agg-hinge; Stage - Early; Non-linearity - Dot Product; Granularity - Edge | | | | | |
| Non-injective | $0.7 \pm 0.022$ | $0.734 \pm 0.025$ | $0.78 \pm 0.018$ | $0.625 \pm 0.02$ | $0.653 \pm 0.018$ |
| Injective | $0.755 \pm 0.019$ | $0.763 \pm 0.025$ | $0.826 \pm 0.014$ | $0.656 \pm 0.02$ | $0.67 \pm 0.019$ |
| Fixed axes: Rel. Dist. - Agg-hinge; Stage - Early; Non-linearity - Hinge; Granularity - Edge | | | | | |
| Non-injective | $0.763 \pm 0.018$ | $0.753 \pm 0.025$ | $0.831 \pm 0.015$ | $0.666 \pm 0.02$ | $0.686 \pm 0.017$ |
| Injective | $0.758 \pm 0.019$ | $0.78 \pm 0.023$ | $0.83 \pm 0.015$ | $0.661 \pm 0.02$ | $0.681 \pm 0.019$ |
| Fixed axes: Rel. Dist. - Agg-hinge; Stage - Early; Non-linearity - Neural; Granularity - Edge | | | | | |
| Non-injective | $0.677 \pm 0.022$ | $0.729 \pm 0.025$ | $0.747 \pm 0.019$ | $0.618 \pm 0.019$ | $0.633 \pm 0.018$ |
| Injective | $0.68 \pm 0.022$ | $0.755 \pm 0.024$ | $0.807 \pm 0.016$ | $0.625 \pm 0.02$ | $0.656 \pm 0.019$ |
| Fixed axes: Rel. Dist. - Agg-MLP; Stage - Early; Non-linearity - Dot Product; Granularity - Edge | | | | | |
| Non-injective | $0.677 \pm 0.022$ | $0.766 \pm 0.023$ | $0.792 \pm 0.016$ | $0.638 \pm 0.019$ | $0.65 \pm 0.018$ |
| Injective | $0.758 \pm 0.019$ | $0.773 \pm 0.023$ | $0.853 \pm 0.014$ | $0.662 \pm 0.02$ | $0.683 \pm 0.017$ |
| Fixed axes: Rel. Dist. - Agg-MLP; Stage - Early; Non-linearity - Hinge; Granularity - Edge | | | | | |
| Non-injective | $0.748 \pm 0.019$ | $0.778 \pm 0.023$ | $0.847 \pm 0.014$ | $0.658 \pm 0.021$ | $0.684 \pm 0.018$ |
| Injective | $0.789 \pm 0.017$ | $0.807 \pm 0.023$ | $0.871 \pm 0.013$ | $0.663 \pm 0.02$ | $0.678 \pm 0.018$ |
| Fixed axes: Rel. Dist. - Agg-MLP; Stage - Early; Non-linearity - Neural; Granularity - Edge | | | | | |
| Non-injective | $0.662 \pm 0.024$ | $0.735 \pm 0.024$ | $0.745 \pm 0.019$ | $0.629 \pm 0.02$ | $0.65 \pm 0.018$ |
| Injective | $0.703 \pm 0.021$ | $0.738 \pm 0.026$ | $0.815 \pm 0.015$ | $0.643 \pm 0.021$ | $0.644 \pm 0.019$ |
| Fixed axes: Rel. Dist. - Set align.; Stage - Early; Non-linearity - Dot Product; Granularity - Edge | | | | | |
| Non-injective | $0.715 \pm 0.021$ | $0.779 \pm 0.023$ | $0.812 \pm 0.017$ | $0.635 \pm 0.019$ | $0.659 \pm 0.017$ |
| Injective | $0.798 \pm 0.019$ | $0.789 \pm 0.023$ | $0.862 \pm 0.015$ | $0.674 \pm 0.019$ | $0.688 \pm 0.018$ |
| Fixed axes: Rel. Dist. - Set align.; Stage - Early; Non-linearity - Hinge; Granularity - Edge | | | | | |
| Non-injective | $0.783 \pm 0.017$ | $0.785 \pm 0.024$ | $0.807 \pm 0.015$ | $0.657 \pm 0.021$ | $0.696 \pm 0.018$ |
| Injective | $0.817 \pm 0.017$ | $0.837 \pm 0.02$ | $0.887 \pm 0.012$ | $0.677 \pm 0.02$ | $0.71 \pm 0.018$ |
| Fixed axes: Rel. Dist. - Set align.; Stage - Early; Non-linearity - Neural; Granularity - Edge | | | | | |
| Non-injective | $0.708 \pm 0.022$ | $0.763 \pm 0.024$ | $0.807 \pm 0.016$ | $0.648 \pm 0.019$ | $0.655 \pm 0.018$ |
| Injective | $0.725 \pm 0.021$ | $0.784 \pm 0.023$ | $0.837 \pm 0.015$ | $0.649 \pm 0.019$ | $0.664 \pm 0.018$ |
| Fixed axes: Rel. Dist. - Agg-NTN; Stage - Early; Non-linearity - Dot Product; Granularity - Edge | | | | | |
| Non-injective | $0.71 \pm 0.022$ | $0.751 \pm 0.024$ | $0.815 \pm 0.016$ | $0.653 \pm 0.02$ | $0.644 \pm 0.017$ |
| Injective | $0.74 \pm 0.02$ | $0.783 \pm 0.023$ | $0.833 \pm 0.015$ | $0.655 \pm 0.02$ | $0.672 \pm 0.018$ |
| Fixed axes: Rel. Dist. - Agg-NTN; Stage - Early; Non-linearity - Hinge; Granularity - Edge | | | | | |
| Non-injective | $0.79 \pm 0.017$ | $0.785 \pm 0.023$ | $0.835 \pm 0.014$ | $0.661 \pm 0.02$ | $0.683 \pm 0.018$ |
| Injective | $0.76 \pm 0.019$ | $0.766 \pm 0.025$ | $0.855 \pm 0.013$ | $0.653 \pm 0.021$ | $0.679 \pm 0.018$ |
| Fixed axes: Rel. Dist. - Agg-NTN; Stage - Early; Non-linearity - Neural; Granularity - Edge | | | | | |
| Non-injective | $0.701 \pm 0.023$ | $0.734 \pm 0.028$ | $0.811 \pm 0.016$ | $0.625 \pm 0.02$ | $0.648 \pm 0.019$ |
| Injective | $0.689 \pm 0.022$ | $0.749 \pm 0.027$ | $0.831 \pm 0.015$ | $0.651 \pm 0.021$ | $0.664 \pm 0.019$ |

Table 17: Comparison of different **interaction structures** for **early interaction** models with **edge-level granularity** across **first five datasets**, using mean average precision (MAP). Green cells indicate the best method for the corresponding datasets.

| Interaction Structure ↓ | FR | MM | MR | MSRC | MCF |
|---|---|---|---|---|---|
| Fixed axes: Rel. Dist. - Agg-hinge; Stage - Early; Non-linearity - Dot Product; Granularity - Edge | | | | | |
| Non-injective | $0.727 \pm 0.019$ | $0.706 \pm 0.02$ | $0.746 \pm 0.019$ | $0.421 \pm 0.012$ | $0.601 \pm 0.018$ |
| Injective | $0.8 \pm 0.018$ | $0.777 \pm 0.018$ | $0.803 \pm 0.015$ | $0.415 \pm 0.011$ | $0.616 \pm 0.018$ |
| Fixed axes: Rel. Dist. - Agg-hinge; Stage - Early; Non-linearity - Hinge; Granularity - Edge | | | | | |
| Non-injective | $0.772 \pm 0.018$ | $0.799 \pm 0.015$ | $0.815 \pm 0.014$ | $0.423 \pm 0.012$ | $0.612 \pm 0.019$ |
| Injective | $0.798 \pm 0.019$ | $0.785 \pm 0.016$ | $0.817 \pm 0.014$ | $0.421 \pm 0.012$ | $0.619 \pm 0.019$ |
| Fixed axes: Rel. Dist. - Agg-hinge; Stage - Early; Non-linearity - Neural; Granularity - Edge | | | | | |
| Non-injective | $0.704 \pm 0.022$ | $0.691 \pm 0.02$ | $0.722 \pm 0.02$ | $0.422 \pm 0.011$ | $0.589 \pm 0.019$ |
| Injective | $0.749 \pm 0.022$ | $0.705 \pm 0.019$ | $0.766 \pm 0.016$ | $0.406 \pm 0.012$ | $0.595 \pm 0.018$ |
| Fixed axes: Rel. Dist. - Agg-MLP; Stage - Early; Non-linearity - Dot Product; Granularity - Edge | | | | | |
| Non-injective | $0.751 \pm 0.019$ | $0.723 \pm 0.019$ | $0.788 \pm 0.016$ | $0.416 \pm 0.011$ | $0.596 \pm 0.02$ |
| Injective | $0.786 \pm 0.017$ | $0.791 \pm 0.016$ | $0.827 \pm 0.014$ | $0.404 \pm 0.011$ | $0.614 \pm 0.018$ |
| Fixed axes: Rel. Dist. - Agg-MLP; Stage - Early; Non-linearity - Hinge; Granularity - Edge | | | | | |
| Non-injective | $0.756 \pm 0.019$ | $0.787 \pm 0.015$ | $0.786 \pm 0.015$ | $0.416 \pm 0.012$ | $0.628 \pm 0.019$ |
| Injective | $0.815 \pm 0.016$ | $0.833 \pm 0.014$ | $0.851 \pm 0.011$ | $0.423 \pm 0.012$ | $0.599 \pm 0.018$ |
| Fixed axes: Rel. Dist. - Agg-MLP; Stage - Early; Non-linearity - Neural; Granularity - Edge | | | | | |
| Non-injective | $0.74 \pm 0.02$ | $0.687 \pm 0.021$ | $0.732 \pm 0.02$ | $0.416 \pm 0.011$ | $0.603 \pm 0.018$ |
| Injective | $0.771 \pm 0.019$ | $0.717 \pm 0.018$ | $0.795 \pm 0.016$ | $0.401 \pm 0.011$ | $0.601 \pm 0.018$ |
| Fixed axes: Rel. Dist. - Set align.; Stage - Early; Non-linearity - Dot Product; Granularity - Edge | | | | | |
| Non-injective | $0.799 \pm 0.017$ | $0.789 \pm 0.016$ | $0.81 \pm 0.016$ | $0.421 \pm 0.012$ | $0.609 \pm 0.02$ |
| Injective | $0.817 \pm 0.016$ | $0.791 \pm 0.016$ | $0.819 \pm 0.014$ | $0.435 \pm 0.012$ | $0.63 \pm 0.018$ |
| Fixed axes: Rel. Dist. - Set align.; Stage - Early; Non-linearity - Hinge; Granularity - Edge | | | | | |
| Non-injective | $0.773 \pm 0.019$ | $0.814 \pm 0.015$ | $0.795 \pm 0.016$ | $0.435 \pm 0.012$ | $0.626 \pm 0.02$ |
| Injective | $0.854 \pm 0.013$ | $0.849 \pm 0.012$ | $0.864 \pm 0.011$ | $0.424 \pm 0.012$ | $0.64 \pm 0.018$ |
| Fixed axes: Rel. Dist. - Set align.; Stage - Early; Non-linearity - Neural; Granularity - Edge | | | | | |
| Non-injective | $0.733 \pm 0.02$ | $0.75 \pm 0.018$ | $0.782 \pm 0.017$ | $0.421 \pm 0.011$ | $0.607 \pm 0.018$ |
| Injective | $0.752 \pm 0.02$ | $0.773 \pm 0.016$ | $0.788 \pm 0.015$ | $0.433 \pm 0.012$ | $0.609 \pm 0.02$ |
| Fixed axes: Rel. Dist. - Agg-NTN; Stage - Early; Non-linearity - Dot Product; Granularity - Edge | | | | | |
| Non-injective | $0.754 \pm 0.019$ | $0.719 \pm 0.021$ | $0.777 \pm 0.016$ | $0.418 \pm 0.012$ | $0.614 \pm 0.018$ |
| Injective | $0.771 \pm 0.018$ | $0.81 \pm 0.014$ | $0.858 \pm 0.012$ | $0.425 \pm 0.012$ | $0.617 \pm 0.018$ |
| Fixed axes: Rel. Dist. - Agg-NTN; Stage - Early; Non-linearity - Hinge; Granularity - Edge | | | | | |
| Non-injective | $0.791 \pm 0.017$ | $0.8 \pm 0.016$ | $0.833 \pm 0.014$ | $0.421 \pm 0.011$ | $0.608 \pm 0.02$ |
| Injective | $0.824 \pm 0.016$ | $0.821 \pm 0.014$ | $0.844 \pm 0.013$ | $0.426 \pm 0.012$ | $0.635 \pm 0.019$ |
| Fixed axes: Rel. Dist. - Agg-NTN; Stage - Early; Non-linearity - Neural; Granularity - Edge | | | | | |
| Non-injective | $0.723 \pm 0.022$ | $0.704 \pm 0.02$ | $0.753 \pm 0.018$ | $0.419 \pm 0.012$ | $0.584 \pm 0.02$ |
| Injective | $0.762 \pm 0.02$ | $0.767 \pm 0.018$ | $0.802 \pm 0.016$ | $0.411 \pm 0.012$ | $0.593 \pm 0.018$ |

Table 18: Comparison of different **interaction structures** for **early interaction** models with **edge-level granularity** across **last five datasets**, using mean average precision (MAP). Green cells indicate the best method for the corresponding datasets.

**Analysis of Late Node Interaction along Interaction Non-Linearity Design Choices**    In Table 19, we observe: **(1)** The Neural non-linearity wins in six datasets for both Injective and Non-injective structures, making it the preferred alternative in this context. **(2)** Both Hinge and Dot Product exhibit competitive performance; however, the preferred option between these two varies depending on the dataset.

| Interaction Non-linearity ↓ | AIDS | Mutag | FM | NCI | MOLT |
|---|---|---|---|---|---|
| Fixed axes: Rel. Dist. - Set align.; Stage - Late; Structure - Injective; Granularity - Node | | | | | |
| Dot Product | $0.631 \pm 0.022$ | $0.66 \pm 0.026$ | $0.722 \pm 0.019$ | $0.608 \pm 0.019$ | $0.612 \pm 0.016$ |
| Hinge | $0.633 \pm 0.021$ | $0.647 \pm 0.028$ | $0.72 \pm 0.019$ | $0.625 \pm 0.019$ | $0.622 \pm 0.017$ |
| Neural | $0.664 \pm 0.021$ | $0.69 \pm 0.026$ | $0.758 \pm 0.017$ | $0.612 \pm 0.019$ | $0.621 \pm 0.017$ |
| Fixed axes: Rel. Dist. - Set align.; Stage - Late; Structure - Non-injective; Granularity - Node | | | | | |
| Dot Product | $0.6 \pm 0.023$ | $0.652 \pm 0.027$ | $0.702 \pm 0.021$ | $0.598 \pm 0.02$ | $0.617 \pm 0.017$ |
| Hinge | $0.636 \pm 0.022$ | $0.647 \pm 0.027$ | $0.694 \pm 0.022$ | $0.617 \pm 0.019$ | $0.616 \pm 0.017$ |
| Neural | $0.635 \pm 0.022$ | $0.681 \pm 0.026$ | $0.736 \pm 0.017$ | $0.605 \pm 0.019$ | $0.609 \pm 0.017$ |

| Interaction Non-linearity ↓ | FR | MM | MR | MSRC | MCF |
|---|---|---|---|---|---|
| Fixed axes: Rel. Dist. - Set align.; Stage - Late; Structure - Injective; Granularity - Node | | | | | |
| Dot Product | $0.62 \pm 0.023$ | $0.664 \pm 0.019$ | $0.643 \pm 0.019$ | $0.415 \pm 0.012$ | $0.57 \pm 0.019$ |
| Hinge | $0.652 \pm 0.024$ | $0.72 \pm 0.017$ | $0.694 \pm 0.018$ | $0.399 \pm 0.012$ | $0.574 \pm 0.019$ |
| Neural | $0.683 \pm 0.022$ | $0.711 \pm 0.017$ | $0.738 \pm 0.017$ | $0.41 \pm 0.012$ | $0.576 \pm 0.019$ |
| Fixed axes: Rel. Dist. - Set align.; Stage - Late; Structure - Non-injective; Granularity - Node | | | | | |
| Dot Product | $0.621 \pm 0.024$ | $0.671 \pm 0.019$ | $0.681 \pm 0.02$ | $0.403 \pm 0.012$ | $0.56 \pm 0.021$ |
| Hinge | $0.645 \pm 0.024$ | $0.634 \pm 0.021$ | $0.701 \pm 0.02$ | $0.392 \pm 0.012$ | $0.577 \pm 0.019$ |
| Neural | $0.689 \pm 0.021$ | $0.705 \pm 0.019$ | $0.741 \pm 0.017$ | $0.399 \pm 0.012$ | $0.58 \pm 0.019$ |

Table 19: Comparison of different **interaction non-linearities** for **late interaction** models with **node-level granularity** across ten datasets, using mean average precision (MAP). Green and yellow cells indicate the best and second best methods respectively for the corresponding dataset.

**Analysis of Late Edge Interaction along Interaction Non-Linearity Design Choices**    In Table 20, we observe: **(1)** Hinge non-linearity clearly outperforms other options for both Injective and Non-injective structures. Notably, the Injective variant stands out as the best-performing late interaction model in our study. **(2)** Neural non-linearity, while not as good as Hinge, outperforms Dot Product for most datasets.

| Interaction Non-linearity ↓ | AIDS | Mutag | FM | NCI | MOLT |
|---|---|---|---|---|---|
| Fixed axes: Rel. Dist. - Set align.; Stage - Late; Structure - Injective; Granularity - Edge | | | | | |
| Dot Product | $0.68 \pm 0.021$ | $0.712 \pm 0.026$ | $0.755 \pm 0.016$ | $0.623 \pm 0.02$ | $0.634 \pm 0.018$ |
| Hinge | $0.712 \pm 0.018$ | $0.721 \pm 0.025$ | $0.793 \pm 0.016$ | $0.643 \pm 0.02$ | $0.662 \pm 0.016$ |
| Neural | $0.704 \pm 0.021$ | $0.733 \pm 0.023$ | $0.782 \pm 0.017$ | $0.615 \pm 0.019$ | $0.649 \pm 0.016$ |
| Fixed axes: Rel. Dist. - Set align.; Stage - Late; Structure - Non-injective; Granularity - Edge | | | | | |
| Dot Product | $0.681 \pm 0.022$ | $0.681 \pm 0.026$ | $0.759 \pm 0.019$ | $0.611 \pm 0.019$ | $0.621 \pm 0.017$ |
| Hinge | $0.702 \pm 0.021$ | $0.687 \pm 0.027$ | $0.774 \pm 0.016$ | $0.631 \pm 0.021$ | $0.638 \pm 0.017$ |
| Neural | $0.7 \pm 0.022$ | $0.713 \pm 0.025$ | $0.77 \pm 0.017$ | $0.609 \pm 0.02$ | $0.618 \pm 0.017$ |

| Interaction Non-linearity ↓ | FR | MM | MR | MSRC | MCF |
|---|---|---|---|---|---|
| Fixed axes: Rel. Dist. - Set align.; Stage - Late; Structure - Injective; Granularity - Edge | | | | | |
| Dot Product | $0.719 \pm 0.02$ | $0.676 \pm 0.018$ | $0.735 \pm 0.018$ | $0.404 \pm 0.012$ | $0.58 \pm 0.019$ |
| Hinge | $0.744 \pm 0.019$ | $0.758 \pm 0.015$ | $0.782 \pm 0.014$ | $0.397 \pm 0.013$ | $0.572 \pm 0.02$ |
| Neural | $0.734 \pm 0.02$ | $0.758 \pm 0.016$ | $0.764 \pm 0.015$ | $0.411 \pm 0.012$ | $0.587 \pm 0.019$ |
| Fixed axes: Rel. Dist. - Set align.; Stage - Late; Structure - Non-injective; Granularity - Edge | | | | | |
| Dot Product | $0.693 \pm 0.022$ | $0.702 \pm 0.019$ | $0.723 \pm 0.019$ | $0.411 \pm 0.012$ | $0.571 \pm 0.02$ |
| Hinge | $0.731 \pm 0.018$ | $0.735 \pm 0.017$ | $0.776 \pm 0.017$ | $0.412 \pm 0.012$ | $0.609 \pm 0.018$ |
| Neural | $0.716 \pm 0.02$ | $0.721 \pm 0.016$ | $0.757 \pm 0.017$ | $0.403 \pm 0.011$ | $0.572 \pm 0.019$ |

Table 20: Comparison of different **interaction non-linearities** for **late interaction** models with **edge-level granularity** across ten datasets, using mean average precision (MAP). Green and yellow cells indicate the best and second best methods respectively for the corresponding dataset.

**Analysis of Early Node Interaction along Interaction Non-Linearity Design Choices** In Tables 21 and 22 , we observe: **(1)** Hinge non-linearity is typically the best choice for non-injective structures. **(2)** For injective networks, no alternative clearly stands out; however, both Neural and Dot Product non-linearities demonstrate comparable performance to Hinge. **(3)** Notably, Dot Product emerges as the best option in six datasets when used with Agg-MLP relevance distance in an injective network.

| Interaction Non-linearity↓ | AIDS | Mutag | FM | NCI | MOLT |
|---|---|---|---|---|---|
| Fixed axes: Rel. Dist. - Agg-hinge; Stage - Early; Structure - Injective; Granularity - Node | | | | | |
| Dot Product | $0.64 \pm 0.019$ | $0.75 \pm 0.023$ | $0.79 \pm 0.016$ | $0.619 \pm 0.019$ | $0.63 \pm 0.018$ |
| Hinge | $0.662 \pm 0.019$ | $0.734 \pm 0.025$ | $0.785 \pm 0.017$ | $0.624 \pm 0.02$ | $0.635 \pm 0.018$ |
| Neural | $0.614 \pm 0.021$ | $0.736 \pm 0.025$ | $0.779 \pm 0.017$ | $0.613 \pm 0.02$ | $0.637 \pm 0.019$ |
| Fixed axes: Rel. Dist. - Agg-MLP; Stage - Early; Structure - Injective; Granularity - Node | | | | | |
| Dot Product | $0.658 \pm 0.019$ | $0.768 \pm 0.023$ | $0.806 \pm 0.016$ | $0.641 \pm 0.019$ | $0.649 \pm 0.017$ |
| Hinge | $0.683 \pm 0.019$ | $0.757 \pm 0.023$ | $0.785 \pm 0.016$ | $0.629 \pm 0.019$ | $0.627 \pm 0.018$ |
| Neural | $0.629 \pm 0.021$ | $0.764 \pm 0.024$ | $0.777 \pm 0.016$ | $0.626 \pm 0.02$ | $0.641 \pm 0.017$ |
| Fixed axes: Rel. Dist. - Set align.; Stage - Early; Structure - Injective; Granularity - Node | | | | | |
| Dot Product | $0.71 \pm 0.019$ | $0.779 \pm 0.022$ | $0.8 \pm 0.017$ | $0.632 \pm 0.02$ | $0.652 \pm 0.017$ |
| Hinge | $0.734 \pm 0.019$ | $0.774 \pm 0.023$ | $0.834 \pm 0.016$ | $0.64 \pm 0.019$ | $0.647 \pm 0.018$ |
| Neural | $0.69 \pm 0.02$ | $0.783 \pm 0.023$ | $0.827 \pm 0.015$ | $0.654 \pm 0.019$ | $0.659 \pm 0.018$ |
| Fixed axes: Rel. Dist. - Agg-NTN; Stage - Early; Structure - Injective; Granularity - Node | | | | | |
| Dot Product | $0.721 \pm 0.019$ | $0.772 \pm 0.022$ | $0.812 \pm 0.016$ | $0.626 \pm 0.019$ | $0.636 \pm 0.018$ |
| Hinge | $0.743 \pm 0.018$ | $0.773 \pm 0.024$ | $0.805 \pm 0.016$ | $0.615 \pm 0.019$ | $0.629 \pm 0.017$ |
| Neural | $0.667 \pm 0.02$ | $0.791 \pm 0.021$ | $0.828 \pm 0.015$ | $0.629 \pm 0.019$ | $0.651 \pm 0.018$ |
| Fixed axes: Rel. Dist. - Agg-hinge; Stage - Early; Structure - Non-injective; Granularity - Node | | | | | |
| Dot Product | $0.609 \pm 0.02$ | $0.693 \pm 0.026$ | $0.686 \pm 0.018$ | $0.588 \pm 0.019$ | $0.603 \pm 0.018$ |
| Hinge | $0.726 \pm 0.02$ | $0.723 \pm 0.026$ | $0.79 \pm 0.016$ | $0.618 \pm 0.019$ | $0.651 \pm 0.018$ |
| Neural | $0.598 \pm 0.021$ | $0.694 \pm 0.025$ | $0.712 \pm 0.019$ | $0.591 \pm 0.019$ | $0.629 \pm 0.018$ |
| Fixed axes: Rel. Dist. - Agg-MLP; Stage - Early; Structure - Non-injective; Granularity - Node | | | | | |
| Dot Product | $0.63 \pm 0.022$ | $0.713 \pm 0.025$ | $0.765 \pm 0.016$ | $0.593 \pm 0.019$ | $0.619 \pm 0.017$ |
| Hinge | $0.637 \pm 0.021$ | $0.725 \pm 0.024$ | $0.808 \pm 0.015$ | $0.63 \pm 0.02$ | $0.636 \pm 0.017$ |
| Neural | $0.629 \pm 0.023$ | $0.715 \pm 0.025$ | $0.769 \pm 0.017$ | $0.623 \pm 0.018$ | $0.627 \pm 0.018$ |
| Fixed axes: Rel. Dist. - Set align.; Stage - Early; Structure - Non-injective; Granularity - Node | | | | | |
| Dot Product | $0.608 \pm 0.022$ | $0.754 \pm 0.024$ | $0.759 \pm 0.017$ | $0.603 \pm 0.019$ | $0.62 \pm 0.017$ |
| Hinge | $0.676 \pm 0.022$ | $0.669 \pm 0.024$ | $0.772 \pm 0.018$ | $0.599 \pm 0.02$ | $0.637 \pm 0.018$ |
| Neural | $0.593 \pm 0.021$ | $0.748 \pm 0.025$ | $0.772 \pm 0.016$ | $0.614 \pm 0.019$ | $0.646 \pm 0.017$ |
| Fixed axes: Rel. Dist. - Agg-NTN; Stage - Early; Structure - Non-injective; Granularity - Node | | | | | |
| Dot Product | $0.669 \pm 0.022$ | $0.742 \pm 0.025$ | $0.75 \pm 0.016$ | $0.602 \pm 0.019$ | $0.628 \pm 0.017$ |
| Hinge | $0.686 \pm 0.019$ | $0.758 \pm 0.022$ | $0.818 \pm 0.015$ | $0.641 \pm 0.02$ | $0.664 \pm 0.017$ |
| Neural | $0.635 \pm 0.025$ | $0.755 \pm 0.025$ | $0.796 \pm 0.016$ | $0.633 \pm 0.019$ | $0.641 \pm 0.017$ |

Table 21: Comparison of different **interaction non-linearities** for **early interaction** models with **node-level granularity** across **first five datasets**, using mean average precision (MAP). Green and yellow cells indicate the best and second best methods respectively for the corresponding dataset.

| Interaction Non-linearity↓ | FR | MM | MR | MSRC | MCF |
|---|---|---|---|---|---|
| *Fixed axes: Rel. Dist. - Agg-hinge; Stage - Early; Structure - Injective; Granularity - Node* | | | | | |
| Dot Product | $0.73 \pm 0.019$ | $0.699 \pm 0.019$ | $0.736 \pm 0.016$ | $0.423 \pm 0.012$ | $0.574 \pm 0.018$ |
| Hinge | $0.733 \pm 0.021$ | $0.713 \pm 0.018$ | $0.752 \pm 0.018$ | $0.403 \pm 0.011$ | $0.575 \pm 0.02$ |
| Neural | $0.73 \pm 0.02$ | $0.712 \pm 0.018$ | $0.733 \pm 0.017$ | $0.417 \pm 0.011$ | $0.579 \pm 0.019$ |
| *Fixed axes: Rel. Dist. - Agg-MLP; Stage - Early; Structure - Injective; Granularity - Node* | | | | | |
| Dot Product | $0.751 \pm 0.018$ | $0.741 \pm 0.017$ | $0.809 \pm 0.014$ | $0.431 \pm 0.012$ | $0.577 \pm 0.019$ |
| Hinge | $0.75 \pm 0.019$ | $0.774 \pm 0.017$ | $0.781 \pm 0.016$ | $0.42 \pm 0.011$ | $0.603 \pm 0.019$ |
| Neural | $0.772 \pm 0.017$ | $0.707 \pm 0.018$ | $0.785 \pm 0.015$ | $0.416 \pm 0.011$ | $0.597 \pm 0.019$ |
| *Fixed axes: Rel. Dist. - Set align.; Stage - Early; Structure - Injective; Granularity - Node* | | | | | |
| Dot Product | $0.774 \pm 0.017$ | $0.736 \pm 0.017$ | $0.803 \pm 0.014$ | $0.436 \pm 0.012$ | $0.593 \pm 0.019$ |
| Hinge | $0.774 \pm 0.017$ | $0.759 \pm 0.016$ | $0.806 \pm 0.013$ | $0.426 \pm 0.012$ | $0.584 \pm 0.019$ |
| Neural | $0.746 \pm 0.019$ | $0.764 \pm 0.016$ | $0.776 \pm 0.015$ | $0.433 \pm 0.012$ | $0.613 \pm 0.019$ |
| *Fixed axes: Rel. Dist. - Agg-NTN; Stage - Early; Structure - Injective; Granularity - Node* | | | | | |
| Dot Product | $0.764 \pm 0.019$ | $0.748 \pm 0.017$ | $0.796 \pm 0.015$ | $0.426 \pm 0.012$ | $0.583 \pm 0.019$ |
| Hinge | $0.768 \pm 0.019$ | $0.755 \pm 0.017$ | $0.798 \pm 0.015$ | $0.422 \pm 0.012$ | $0.571 \pm 0.019$ |
| Neural | $0.766 \pm 0.019$ | $0.748 \pm 0.018$ | $0.8 \pm 0.015$ | $0.409 \pm 0.011$ | $0.584 \pm 0.019$ |
| *Fixed axes: Rel. Dist. - Agg-hinge; Stage - Early; Structure - Non-injective; Granularity - Node* | | | | | |
| Dot Product | $0.667 \pm 0.021$ | $0.627 \pm 0.02$ | $0.683 \pm 0.017$ | $0.416 \pm 0.012$ | $0.549 \pm 0.018$ |
| Hinge | $0.75 \pm 0.018$ | $0.723 \pm 0.018$ | $0.787 \pm 0.015$ | $0.414 \pm 0.012$ | $0.584 \pm 0.02$ |
| Neural | $0.674 \pm 0.022$ | $0.643 \pm 0.02$ | $0.662 \pm 0.02$ | $0.401 \pm 0.011$ | $0.554 \pm 0.019$ |
| *Fixed axes: Rel. Dist. - Agg-MLP; Stage - Early; Structure - Non-injective; Granularity - Node* | | | | | |
| Dot Product | $0.687 \pm 0.02$ | $0.678 \pm 0.018$ | $0.741 \pm 0.017$ | $0.42 \pm 0.011$ | $0.562 \pm 0.019$ |
| Hinge | $0.744 \pm 0.02$ | $0.71 \pm 0.018$ | $0.773 \pm 0.016$ | $0.41 \pm 0.011$ | $0.586 \pm 0.02$ |
| Neural | $0.698 \pm 0.021$ | $0.698 \pm 0.018$ | $0.739 \pm 0.018$ | $0.415 \pm 0.012$ | $0.575 \pm 0.019$ |
| *Fixed axes: Rel. Dist. - Set align.; Stage - Early; Structure - Non-injective; Granularity - Node* | | | | | |
| Dot Product | $0.706 \pm 0.02$ | $0.691 \pm 0.017$ | $0.715 \pm 0.019$ | $0.423 \pm 0.012$ | $0.553 \pm 0.019$ |
| Hinge | $0.678 \pm 0.022$ | $0.711 \pm 0.018$ | $0.768 \pm 0.017$ | $0.419 \pm 0.012$ | $0.601 \pm 0.02$ |
| Neural | $0.719 \pm 0.02$ | $0.703 \pm 0.019$ | $0.742 \pm 0.016$ | $0.416 \pm 0.012$ | $0.579 \pm 0.018$ |
| *Fixed axes: Rel. Dist. - Agg-NTN; Stage - Early; Structure - Non-injective; Granularity - Node* | | | | | |
| Dot Product | $0.715 \pm 0.021$ | $0.711 \pm 0.019$ | $0.762 \pm 0.017$ | $0.415 \pm 0.011$ | $0.571 \pm 0.019$ |
| Hinge | $0.765 \pm 0.019$ | $0.766 \pm 0.016$ | $0.8 \pm 0.014$ | $0.409 \pm 0.012$ | $0.572 \pm 0.02$ |
| Neural | $0.707 \pm 0.021$ | $0.701 \pm 0.019$ | $0.69 \pm 0.019$ | $0.408 \pm 0.011$ | $0.586 \pm 0.018$ |

Table 22: Comparison of different **interaction non-linearities** for **early interaction** models with **node-level granularity** across **last five datasets**, using mean average precision (MAP). Green and yellow cells indicate the best and second best methods respectively for the corresponding dataset.

**Analysis of Early Edge Interaction along Interaction Non-Linearity Design Choices** In Tables 23 and 24, we observe: **(1)**. Hinge non-linearity is the best performer across all datasets, regardless of the interaction structure, **(2)**, Dot Product method ranks second in most cases, although it occasionally outperforms Hinge., **(3)** Neural method is consistently the lowest performer among the three.

| Interaction Non-linearity↓ | AIDS | Mutag | FM | NCI | MOLT |
|---|---|---|---|---|---|
| Fixed axes: Rel. Dist. - Agg-hinge; Stage - Early; Structure - Injective; Granularity - Edge | | | | | |
| Dot Product | $0.755 \pm 0.019$ | $0.763 \pm 0.025$ | $0.826 \pm 0.014$ | $0.656 \pm 0.02$ | $0.67 \pm 0.019$ |
| Hinge | $0.758 \pm 0.019$ | $0.78 \pm 0.023$ | $0.83 \pm 0.015$ | $0.661 \pm 0.02$ | $0.681 \pm 0.019$ |
| Neural | $0.68 \pm 0.022$ | $0.755 \pm 0.024$ | $0.807 \pm 0.016$ | $0.625 \pm 0.02$ | $0.656 \pm 0.019$ |
| Fixed axes: Rel. Dist. - Agg-MLP; Stage - Early; Structure - Injective; Granularity - Edge | | | | | |
| Dot Product | $0.758 \pm 0.019$ | $0.773 \pm 0.023$ | $0.853 \pm 0.014$ | $0.662 \pm 0.02$ | $0.683 \pm 0.017$ |
| Hinge | $0.789 \pm 0.017$ | $0.807 \pm 0.023$ | $0.871 \pm 0.013$ | $0.663 \pm 0.02$ | $0.678 \pm 0.018$ |
| Neural | $0.703 \pm 0.021$ | $0.738 \pm 0.026$ | $0.815 \pm 0.015$ | $0.643 \pm 0.021$ | $0.644 \pm 0.019$ |
| Fixed axes: Rel. Dist. - Set align.; Stage - Early; Structure - Injective; Granularity - Edge | | | | | |
| Dot Product | $0.798 \pm 0.019$ | $0.789 \pm 0.023$ | $0.862 \pm 0.015$ | $0.674 \pm 0.019$ | $0.688 \pm 0.018$ |
| Hinge | $0.817 \pm 0.017$ | $0.837 \pm 0.02$ | $0.887 \pm 0.012$ | $0.677 \pm 0.02$ | $0.71 \pm 0.018$ |
| Neural | $0.725 \pm 0.021$ | $0.784 \pm 0.023$ | $0.837 \pm 0.015$ | $0.649 \pm 0.019$ | $0.664 \pm 0.018$ |
| Fixed axes: Rel. Dist. - Agg-NTN; Stage - Early; Structure - Injective; Granularity - Edge | | | | | |
| Dot Product | $0.74 \pm 0.02$ | $0.783 \pm 0.023$ | $0.833 \pm 0.015$ | $0.655 \pm 0.02$ | $0.672 \pm 0.018$ |
| Hinge | $0.76 \pm 0.019$ | $0.766 \pm 0.025$ | $0.855 \pm 0.013$ | $0.653 \pm 0.021$ | $0.679 \pm 0.018$ |
| Neural | $0.689 \pm 0.022$ | $0.749 \pm 0.027$ | $0.831 \pm 0.015$ | $0.651 \pm 0.021$ | $0.664 \pm 0.019$ |
| Fixed axes: Rel. Dist. - Agg-hinge; Stage - Early; Structure - Non-injective; Granularity - Edge | | | | | |
| Dot Product | $0.7 \pm 0.022$ | $0.734 \pm 0.025$ | $0.78 \pm 0.018$ | $0.625 \pm 0.02$ | $0.653 \pm 0.018$ |
| Hinge | $0.763 \pm 0.018$ | $0.753 \pm 0.025$ | $0.831 \pm 0.015$ | $0.666 \pm 0.02$ | $0.686 \pm 0.017$ |
| Neural | $0.677 \pm 0.022$ | $0.729 \pm 0.025$ | $0.747 \pm 0.019$ | $0.618 \pm 0.019$ | $0.633 \pm 0.018$ |
| Fixed axes: Rel. Dist. - Agg-MLP; Stage - Early; Structure - Non-injective; Granularity - Edge | | | | | |
| Dot Product | $0.677 \pm 0.022$ | $0.766 \pm 0.023$ | $0.792 \pm 0.016$ | $0.638 \pm 0.019$ | $0.65 \pm 0.018$ |
| Hinge | $0.748 \pm 0.019$ | $0.778 \pm 0.023$ | $0.847 \pm 0.014$ | $0.658 \pm 0.021$ | $0.684 \pm 0.018$ |
| Neural | $0.662 \pm 0.024$ | $0.735 \pm 0.024$ | $0.745 \pm 0.019$ | $0.629 \pm 0.02$ | $0.65 \pm 0.018$ |
| Fixed axes: Rel. Dist. - Set align.; Stage - Early; Structure - Non-injective; Granularity - Edge | | | | | |
| Dot Product | $0.715 \pm 0.021$ | $0.779 \pm 0.023$ | $0.812 \pm 0.017$ | $0.635 \pm 0.019$ | $0.659 \pm 0.017$ |
| Hinge | $0.783 \pm 0.017$ | $0.785 \pm 0.024$ | $0.807 \pm 0.015$ | $0.657 \pm 0.021$ | $0.696 \pm 0.018$ |
| Neural | $0.708 \pm 0.022$ | $0.763 \pm 0.024$ | $0.807 \pm 0.016$ | $0.648 \pm 0.019$ | $0.655 \pm 0.018$ |
| Fixed axes: Rel. Dist. - Agg-NTN; Stage - Early; Structure - Non-injective; Granularity - Edge | | | | | |
| Dot Product | $0.71 \pm 0.022$ | $0.751 \pm 0.024$ | $0.815 \pm 0.016$ | $0.653 \pm 0.02$ | $0.644 \pm 0.017$ |
| Hinge | $0.79 \pm 0.017$ | $0.785 \pm 0.023$ | $0.835 \pm 0.014$ | $0.661 \pm 0.02$ | $0.683 \pm 0.018$ |
| Neural | $0.701 \pm 0.023$ | $0.734 \pm 0.028$ | $0.811 \pm 0.016$ | $0.625 \pm 0.02$ | $0.648 \pm 0.019$ |

Table 23: Comparison of different **interaction non-linearities** for **early interaction** models with **edge-level granularity** across **first five datasets**, using mean average precision (MAP). Green and yellow cells indicate the best and second best methods respectively for the corresponding dataset.

| Interaction Non-linearity↓ | FR | MM | MR | MSRC | MCF |
|---|---|---|---|---|---|
| Fixed axes: Rel. Dist. - Agg-hinge; Stage - Early; Structure - Injective; Granularity - Edge | | | | | |
| Dot Product | $0.8 \pm 0.018$ | $0.777 \pm 0.018$ | $0.803 \pm 0.015$ | $0.415 \pm 0.011$ | $0.616 \pm 0.018$ |
| Hinge | $0.798 \pm 0.019$ | $0.785 \pm 0.016$ | $0.817 \pm 0.014$ | $0.421 \pm 0.012$ | $0.619 \pm 0.019$ |
| Neural | $0.749 \pm 0.022$ | $0.705 \pm 0.019$ | $0.766 \pm 0.016$ | $0.406 \pm 0.012$ | $0.595 \pm 0.018$ |
| Fixed axes: Rel. Dist. - Agg-MLP; Stage - Early; Structure - Injective; Granularity - Edge | | | | | |
| Dot Product | $0.786 \pm 0.017$ | $0.791 \pm 0.016$ | $0.827 \pm 0.014$ | $0.404 \pm 0.011$ | $0.614 \pm 0.018$ |
| Hinge | $0.815 \pm 0.016$ | $0.833 \pm 0.014$ | $0.851 \pm 0.011$ | $0.423 \pm 0.012$ | $0.599 \pm 0.018$ |
| Neural | $0.771 \pm 0.019$ | $0.717 \pm 0.018$ | $0.795 \pm 0.016$ | $0.401 \pm 0.011$ | $0.601 \pm 0.018$ |
| Fixed axes: Rel. Dist. - Set align.; Stage - Early; Structure - Injective; Granularity - Edge | | | | | |
| Dot Product | $0.817 \pm 0.016$ | $0.791 \pm 0.016$ | $0.819 \pm 0.014$ | $0.435 \pm 0.012$ | $0.63 \pm 0.018$ |
| Hinge | $0.854 \pm 0.013$ | $0.849 \pm 0.012$ | $0.864 \pm 0.011$ | $0.424 \pm 0.012$ | $0.64 \pm 0.018$ |
| Neural | $0.752 \pm 0.02$ | $0.773 \pm 0.016$ | $0.788 \pm 0.015$ | $0.433 \pm 0.012$ | $0.609 \pm 0.02$ |
| Fixed axes: Rel. Dist. - Agg-NTN; Stage - Early; Structure - Injective; Granularity - Edge | | | | | |
| Dot Product | $0.771 \pm 0.018$ | $0.81 \pm 0.014$ | $0.858 \pm 0.012$ | $0.425 \pm 0.012$ | $0.617 \pm 0.018$ |
| Hinge | $0.824 \pm 0.016$ | $0.821 \pm 0.014$ | $0.844 \pm 0.013$ | $0.426 \pm 0.012$ | $0.635 \pm 0.019$ |
| Neural | $0.762 \pm 0.02$ | $0.767 \pm 0.018$ | $0.802 \pm 0.016$ | $0.411 \pm 0.012$ | $0.593 \pm 0.018$ |
| Fixed axes: Rel. Dist. - Agg-hinge; Stage - Early; Structure - Non-injective; Granularity - Edge | | | | | |
| Dot Product | $0.727 \pm 0.019$ | $0.706 \pm 0.02$ | $0.746 \pm 0.019$ | $0.421 \pm 0.012$ | $0.601 \pm 0.018$ |
| Hinge | $0.772 \pm 0.018$ | $0.799 \pm 0.015$ | $0.815 \pm 0.014$ | $0.423 \pm 0.012$ | $0.612 \pm 0.019$ |
| Neural | $0.704 \pm 0.022$ | $0.691 \pm 0.02$ | $0.722 \pm 0.02$ | $0.422 \pm 0.011$ | $0.589 \pm 0.019$ |
| Fixed axes: Rel. Dist. - Agg-MLP; Stage - Early; Structure - Non-injective; Granularity - Edge | | | | | |
| Dot Product | $0.751 \pm 0.019$ | $0.723 \pm 0.019$ | $0.788 \pm 0.016$ | $0.416 \pm 0.011$ | $0.596 \pm 0.02$ |
| Hinge | $0.756 \pm 0.019$ | $0.787 \pm 0.015$ | $0.786 \pm 0.015$ | $0.416 \pm 0.012$ | $0.628 \pm 0.019$ |
| Neural | $0.74 \pm 0.02$ | $0.687 \pm 0.021$ | $0.732 \pm 0.02$ | $0.416 \pm 0.011$ | $0.603 \pm 0.018$ |
| Fixed axes: Rel. Dist. - Set align.; Stage - Early; Structure - Non-injective; Granularity - Edge | | | | | |
| Dot Product | $0.799 \pm 0.017$ | $0.789 \pm 0.016$ | $0.81 \pm 0.016$ | $0.421 \pm 0.012$ | $0.609 \pm 0.02$ |
| Hinge | $0.773 \pm 0.019$ | $0.814 \pm 0.015$ | $0.795 \pm 0.016$ | $0.435 \pm 0.012$ | $0.626 \pm 0.02$ |
| Neural | $0.733 \pm 0.02$ | $0.75 \pm 0.018$ | $0.782 \pm 0.017$ | $0.421 \pm 0.011$ | $0.607 \pm 0.018$ |
| Fixed axes: Rel. Dist. - Agg-NTN; Stage - Early; Structure - Non-injective; Granularity - Edge | | | | | |
| Dot Product | $0.754 \pm 0.019$ | $0.719 \pm 0.021$ | $0.777 \pm 0.016$ | $0.418 \pm 0.012$ | $0.614 \pm 0.018$ |
| Hinge | $0.791 \pm 0.017$ | $0.8 \pm 0.016$ | $0.833 \pm 0.014$ | $0.421 \pm 0.011$ | $0.608 \pm 0.02$ |
| Neural | $0.723 \pm 0.022$ | $0.704 \pm 0.02$ | $0.753 \pm 0.018$ | $0.419 \pm 0.012$ | $0.584 \pm 0.02$ |

Table 24: Comparison of different **interaction non-linearities** for **early interaction** models with **edge-level granularity** across **last five datasets**, using mean average precision (MAP). Green and yellow cells indicate the best and second best methods respectively for the corresponding dataset.

## G   REAL-WORLD LARGE DATASETS

Recent work such as Greed (Ranjan et al., 2022) proposes a heuristic for adapting small-scale, fast neural graph solvers to the problem of subgraph localization within large-scale graphs. Inspired by this approach, we extended our experiments to include three large-scale graphs drawn from the SNAP repository:

1. com-Amazon: Represents an Amazon product co-purchasing network, with nodes as products and edges denoting co-purchasing relationships. It consists of 334,863 nodes and 925,872 edges.
2. email-Enron: Represents an email communication network, with nodes as individuals and edges as email exchanges. It consists of 36,692 nodes and 183,831 edges.
3. roadnet-CA: Represents the road network of California, with nodes as road intersections and edges as connecting roads. It consists of 1,965,206 nodes and 2,766,607 edges.

To integrate these datasets into our study, we followed the same preprocessing and subgraph extraction methodology outlined in the paper and ran all the designed experiments to further explore the design space of the subgraph matching methods.

| Rel. Dist. | Structure | Non-linearity | Amazon | Email | Roadnet |
|---|---|---|---|---|---|
| Agg-hinge | NA | NA | 0.616 | 0.723 | 0.556 |
| Agg-MLP | NA | NA | 0.544 | 0.631 | 0.616 |
| Agg-NTN | NA | NA | 0.656 | 0.778 | 0.499 |
| Set align. | Non-injective | Dot Product | 0.66 | 0.798 | 0.618 |
| Set align. | Non-injective | Hinge | 0.739 | 0.826 | 0.609 |
| Set align. | Non-injective | Neural | 0.752 | 0.805 | 0.585 |
| Set align. | Injective | Dot Product | 0.739 | 0.852 | 0.637 |
| Set align. | Injective | Hinge | 0.772 | 0.864 | 0.576 |
| Set align. | Injective | Neural | 0.771 | 0.852 | 0.605 |

Table 25: MAP for **late interaction** models with **node-level granularity** across three large and diverse datasets.

| Rel. Dist. | Structure | Non-linearity | Amazon | Email | Roadnet |
|---|---|---|---|---|---|
| Agg-hinge | NA | NA | 0.692 | 0.847 | 0.744 |
| Agg-MLP | NA | NA | 0.652 | 0.783 | 0.682 |
| Agg-NTN | NA | NA | 0.678 | 0.851 | 0.706 |
| Set align. | Non-injective | Dot Product | 0.737 | 0.834 | 0.621 |
| Set align. | Non-injective | Hinge | 0.776 | 0.865 | 0.741 |
| Set align. | Non-injective | Neural | 0.771 | 0.838 | 0.72 |
| Set align. | Injective | Dot Product | 0.749 | 0.852 | 0.668 |
| Set align. | Injective | Hinge | 0.8 | 0.849 | 0.708 |
| Set align. | Injective | Neural | 0.788 | 0.871 | 0.724 |

Table 26: MAP for **late interaction** models with **edge-level granularity** across three large and diverse datasets.

| Rel. Dist. | Structure | Non-linearity | Amazon | Email | Roadnet |
|---|---|---|---|---|---|
| Agg-MLP | Injective | Hinge | 0.805 | 0.902 | 0.68 |
| Agg-MLP | Injective | Dot Product | 0.725 | 0.911 | 0.643 |
| Agg-MLP | Injective | Neural | 0.798 | 0.895 | 0.668 |
| Agg-MLP | Non-Injective | Hinge | 0.77 | 0.869 | 0.646 |
| Agg-MLP | Non-Injective | Dot Product | 0.72 | 0.817 | 0.648 |
| Agg-MLP | Non-Injective | Neural | 0.748 | 0.834 | 0.586 |
| Agg-NTN | Injective | Hinge | 0.798 | 0.906 | 0.699 |
| Agg-NTN | Injective | Dot Product | 0.789 | 0.902 | 0.669 |
| Agg-NTN | Injective | Neural | 0.787 | 0.92 | 0.732 |
| Agg-NTN | Non-Injective | Hinge | 0.768 | 0.855 | 0.668 |
| Agg-NTN | Non-Injective | Dot Product | 0.668 | 0.829 | 0.612 |
| Agg-NTN | Non-Injective | Neural | 0.695 | 0.839 | 0.645 |
| Agg-hinge | Injective | Hinge | 0.756 | 0.866 | 0.659 |
| Agg-hinge | Injective | Dot Product | 0.748 | 0.861 | 0.682 |
| Agg-hinge | Injective | Neural | 0.743 | 0.862 | 0.66 |
| Agg-hinge | Non-Injective | Hinge | 0.799 | 0.878 | 0.662 |
| Agg-hinge | Non-Injective | Dot Product | 0.657 | 0.764 | 0.61 |
| Agg-hinge | Non-Injective | Neural | 0.676 | 0.759 | 0.592 |
| Set align | Injective | Hinge | 0.849 | 0.935 | 0.745 |
| Set align | Injective | Dot Product | 0.816 | 0.905 | 0.706 |
| Set align | Injective | Neural | 0.823 | 0.905 | 0.723 |
| Set align | Non-Injective | Hinge | 0.768 | 0.86 | 0.652 |
| Set align | Non-Injective | Dot Product | 0.729 | 0.827 | 0.648 |
| Set align | Non-Injective | Neural | 0.747 | 0.842 | 0.648 |

Table 27: MAP for **early interaction** models with **node-level granularity** across three large and diverse datasets.

| Rel. Dist. | Structure | Non-linearity | Amazon | Email | Roadnet |
|---|---|---|---|---|---|
| Agg-MLP | Injective | Hinge | 0.84 | 0.926 | 0.773 |
| Agg-MLP | Injective | Dot Product | 0.799 | 0.934 | 0.76 |
| Agg-MLP | Injective | Neural | 0.72 | 0.91 | 0.74 |
| Agg-MLP | Non-Injective | Hinge | 0.801 | 0.886 | 0.671 |
| Agg-MLP | Non-Injective | Dot Product | 0.74 | 0.837 | 0.745 |
| Agg-MLP | Non-Injective | Neural | 0.703 | 0.869 | 0.613 |
| Agg-NTN | Injective | Hinge | 0.791 | 0.922 | 0.731 |
| Agg-NTN | Injective | Dot Product | 0.783 | 0.916 | 0.76 |
| Agg-NTN | Injective | Neural | 0.789 | 0.908 | 0.737 |
| Agg-NTN | Non-Injective | Hinge | 0.795 | 0.868 | 0.816 |
| Agg-NTN | Non-Injective | Dot Product | 0.765 | 0.874 | 0.77 |
| Agg-NTN | Non-Injective | Neural | 0.701 | 0.853 | 0.708 |
| Agg-hinge | Injective | Hinge | 0.827 | 0.939 | 0.801 |
| Agg-hinge | Injective | Dot Product | 0.77 | 0.915 | 0.805 |
| Agg-hinge | Injective | Neural | 0.753 | 0.911 | 0.768 |
| Agg-hinge | Non-Injective | Hinge | 0.797 | 0.874 | 0.73 |
| Agg-hinge | Non-Injective | Dot Product | 0.736 | 0.845 | 0.607 |
| Agg-hinge | Non-Injective | Neural | 0.69 | 0.842 | 0.698 |
| Set align | Injective | Hinge | 0.863 | 0.944 | 0.834 |
| Set align | Injective | Dot Product | 0.827 | 0.921 | 0.828 |
| Set align | Injective | Neural | 0.826 | 0.909 | 0.752 |
| Set align | Non-Injective | Hinge | 0.783 | 0.875 | 0.832 |
| Set align | Non-Injective | Dot Product | 0.802 | 0.872 | 0.802 |
| Set align | Non-Injective | Neural | 0.769 | 0.858 | 0.74 |

Table 28: MAP for **early interaction** models with **edge-level granularity** across three large and diverse datasets.

## H    TRANSFER ABILITY ACROSS DATASETS

In this section, we test our models in an out-of-distribution **transfer** setting by evaluating models on datasets other than the one they were originally trained on. We first fix AIDS as the target dataset and iterate over Mutag, FR and MOLT as source datasets.

**Analysis of transfer abilities of Early Node Interaction models**    In Tables 29 and 30, we observe: **(1)** The strongest transfer abilities are displayed by models trained on the FR dataset, followed by the Mutag dataset, while models trained on the MOLT dataset show severe degradation in performance compared to the baseline model trained on AIDS itself. This pattern can be explained by the extent of relative dissimilarity between the source datasets and AIDS, which is maximum for MOLT (doubly-sized graphs as AIDS), followed by Mutag (mean node/edge counts off by one/two) and finally FR. **(2)** Hinge non-linearity is consistently the best performer even in the transfer setting, similar to the default setting of training and testing on the same dataset.

| Rel. Dist. | Structure | Non-linearity | AIDS → AIDS | Mutag → AIDS | FR → AIDS | MOLT → AIDS |
|---|---|---|---|---|---|---|
| Agg-hinge | Non-injective | Dot Product | 0.609 | 0.330 | 0.520 | 0.253 |
| Agg-hinge | Non-injective | Hinge | **0.726** | **0.360** | **0.626** | **0.369** |
| Agg-hinge | Non-injective | Neural | 0.598 | 0.331 | 0.510 | 0.238 |
| Agg-hinge | Injective | Dot Product | 0.64 | 0.394 | 0.581 | 0.322 |
| Agg-hinge | Injective | Hinge | **0.662** | **0.422** | 0.595 | **0.346** |
| Agg-hinge | Injective | Neural | 0.614 | 0.403 | **0.598** | 0.322 |
| Agg-MLP | Non-injective | Dot Product | 0.63 | 0.323 | 0.569 | 0.363 |
| Agg-MLP | Non-injective | Hinge | **0.637** | **0.378** | **0.624** | **0.437** |
| Agg-MLP | Non-injective | Neural | 0.629 | 0.363 | 0.552 | 0.337 |
| Agg-MLP | Injective | Dot Product | 0.658 | **0.486** | 0.612 | 0.427 |
| Agg-MLP | Injective | Hinge | **0.683** | 0.457 | 0.611 | **0.455** |
| Agg-MLP | Injective | Neural | 0.629 | 0.464 | **0.625** | 0.414 |
| Agg-NTN | Non-injective | Dot Product | 0.669 | **0.450** | 0.557 | **0.407** |
| Agg-NTN | Non-injective | Hinge | **0.686** | 0.442 | **0.623** | 0.405 |
| Agg-NTN | Non-injective | Neural | 0.635 | 0.367 | 0.549 | 0.268 |
| Agg-NTN | Injective | Dot Product | 0.721 | 0.487 | **0.635** | **0.420** |
| Agg-NTN | Injective | Hinge | **0.743** | 0.478 | 0.626 | 0.356 |
| Agg-NTN | Injective | Neural | 0.667 | **0.499** | 0.617 | 0.395 |
| Set align. | Non-injective | Dot Product | 0.608 | 0.377 | 0.574 | 0.419 |
| Set align. | Non-injective | Hinge | **0.676** | 0.394 | **0.600** | **0.451** |
| Set align. | Non-injective | Neural | 0.593 | **0.450** | 0.599 | 0.363 |
| Set align. | Injective | Dot Product | 0.71 | **0.510** | 0.663 | 0.448 |
| Set align. | Injective | Hinge | **0.734** | 0.505 | **0.684** | 0.480 |
| Set align. | Injective | Neural | 0.69 | 0.507 | 0.641 | **0.494** |

Table 29: Comparison of different network configurations for early interaction models with **node-level granularity**, using mean average precision (MAP). Green and yellow cells indicate the best and second best transfer methods respectively for the corresponding network. Cells in **boldface** represent the best non-linearity individually for each transfer combination.

| Rel. Dist. | Structure | Non-linearity | AIDS → AIDS | Mutag → AIDS | FR → AIDS | MOLT → AIDS |
|---|---|---|---|---|---|---|
| Agg-hinge | Non-injective | Dot Product | 0.7 | 0.400 | 0.560 | 0.290 |
| Agg-hinge | Non-injective | Hinge | **0.763** | **0.445** | **0.640** | **0.447** |
| Agg-hinge | Non-injective | Neural | 0.677 | 0.414 | 0.541 | 0.219 |
| Agg-hinge | Injective | Dot Product | 0.755 | 0.454 | 0.657 | **0.305** |
| Agg-hinge | Injective | Hinge | **0.758** | **0.477** | **0.664** | 0.274 |
| Agg-hinge | Injective | Neural | 0.68 | 0.422 | 0.578 | 0.278 |
| Agg-MLP | Non-injective | Dot Product | 0.677 | 0.451 | 0.589 | 0.341 |
| Agg-MLP | Non-injective | Hinge | **0.748** | **0.423** | **0.619** | 0.337 |
| Agg-MLP | Non-injective | Neural | 0.662 | 0.396 | 0.563 | **0.351** |
| Agg-MLP | Injective | Dot Product | 0.758 | 0.454 | 0.662 | 0.373 |
| Agg-MLP | Injective | Hinge | **0.789** | **0.491** | **0.718** | **0.425** |
| Agg-MLP | Injective | Neural | 0.703 | 0.419 | 0.620 | 0.312 |
| Agg-NTN | Non-injective | Dot Product | 0.71 | 0.444 | 0.603 | **0.399** |
| Agg-NTN | Non-injective | Hinge | **0.79** | **0.489** | **0.658** | 0.241 |
| Agg-NTN | Non-injective | Neural | 0.701 | 0.419 | 0.572 | 0.361 |
| Agg-NTN | Injective | Dot Product | 0.74 | 0.466 | 0.623 | 0.367 |
| Agg-NTN | Injective | Hinge | **0.76** | **0.518** | **0.714** | **0.431** |
| Agg-NTN | Injective | Neural | 0.689 | 0.409 | 0.599 | 0.320 |
| Set align. | Non-injective | Dot Product | 0.715 | 0.474 | 0.668 | 0.374 |
| Set align. | Non-injective | Hinge | **0.783** | **0.483** | **0.687** | **0.497** |
| Set align. | Non-injective | Neural | 0.708 | 0.429 | 0.594 | 0.412 |
| Set align. | Injective | Dot Product | 0.798 | 0.505 | 0.695 | 0.447 |
| Set align. | Injective | Hinge | **0.817** | **0.599** | **0.773** | **0.538** |
| Set align. | Injective | Neural | 0.725 | 0.468 | 0.599 | 0.428 |

Table 30: Comparison of different network configurations for early interaction models with **edge-level granularity**, using mean average precision (MAP). Green and yellow cells indicate the best and second best transfer methods respectively for the corresponding network. Cells in **boldface** represent the best non-linearity individually for each transfer combination.

## I   VARIATION IN PERFORMANCE WITH INTRINSIC DATASET CHARACTERISTICS

In this section, we study how different characteristics of the dataset affect the performance of our models. In particular, we choose a metric (like edge count in a graph), split the corpus set into $4$ equal-sized buckets based on this metric, and compute the MAP score over all test query graphs on each of these splits independently, using the model trained on the entire corpus. Three metrics are considered - **(1)** Node count (representative of graph size) **(2)** Edge count (representative of graph size) **(3)** Standard deviation of degrees of all nodes in the graph (representative of the regularity of graphs). We tackle each of these metrics individually in the subsections below. Subset 0 represents the split of the dataset with the lowest value for that metric, while subset 3 is the other extreme, with the highest values for the metric. For instance, subset 0 for the *Standard deviation of node degrees* metric represents the set of graphs with the least variation in node degrees i.e. the most regularity.

## I.1 Edge Count

In Tables 31 and 32, we observe: **(1)** With non-injective interaction structure, we observe that graphs with more edges see better performance on average. **(2)** Under the injective interaction structure, the difference in scores across subsets is more subtle, which indicates that injective mapping might not be affected significantly by edge count.

| Rel. Dist. | Structure | Non-linearity | AIDS | Subset 0 | Subset 1 | Subset 2 | Subset 3 |
|---|---|---|---|---|---|---|---|
| Agg-hinge | Non-injective | Dot Product | 0.609 | 0.555 | 0.574 | 0.617 | 0.665 |
| Agg-hinge | Non-injective | Hinge | 0.726 | 0.763 | 0.703 | 0.738 | 0.739 |
| Agg-hinge | Non-injective | Neural | 0.598 | 0.519 | 0.547 | 0.606 | 0.659 |
| Agg-hinge | Injective | Dot Product | 0.64 | 0.655 | 0.627 | 0.658 | 0.672 |
| Agg-hinge | Injective | Hinge | 0.662 | 0.663 | 0.643 | 0.664 | 0.693 |
| Agg-hinge | Injective | Neural | 0.614 | 0.535 | 0.566 | 0.623 | 0.673 |
| Agg-MLP | Non-injective | Dot Product | 0.63 | 0.562 | 0.580 | 0.641 | 0.686 |
| Agg-MLP | Non-injective | Hinge | 0.637 | 0.624 | 0.613 | 0.651 | 0.671 |
| Agg-MLP | Non-injective | Neural | 0.629 | 0.564 | 0.584 | 0.646 | 0.674 |
| Agg-MLP | Injective | Dot Product | 0.658 | 0.687 | 0.650 | 0.670 | 0.677 |
| Agg-MLP | Injective | Hinge | 0.683 | 0.722 | 0.685 | 0.688 | 0.698 |
| Agg-MLP | Injective | Neural | 0.629 | 0.612 | 0.599 | 0.642 | 0.667 |
| Agg-NTN | Non-injective | Dot Product | 0.669 | 0.613 | 0.630 | 0.672 | 0.715 |
| Agg-NTN | Non-injective | Hinge | 0.686 | 0.681 | 0.641 | 0.690 | 0.726 |
| Agg-NTN | Non-injective | Neural | 0.635 | 0.572 | 0.579 | 0.628 | 0.698 |
| Agg-NTN | Injective | Dot Product | 0.721 | 0.736 | 0.697 | 0.728 | 0.745 |
| Agg-NTN | Injective | Hinge | 0.743 | 0.773 | 0.736 | 0.756 | 0.750 |
| Agg-NTN | Injective | Neural | 0.667 | 0.643 | 0.619 | 0.675 | 0.713 |
| Set align. | Non-injective | Dot Product | 0.608 | 0.574 | 0.592 | 0.614 | 0.650 |
| Set align. | Non-injective | Hinge | 0.676 | 0.706 | 0.671 | 0.693 | 0.703 |
| Set align. | Non-injective | Neural | 0.593 | 0.566 | 0.566 | 0.616 | 0.637 |
| Set align. | Injective | Dot Product | 0.71 | 0.767 | 0.698 | 0.713 | 0.721 |
| Set align. | Injective | Hinge | 0.734 | 0.821 | 0.735 | 0.740 | 0.724 |
| Set align. | Injective | Neural | 0.69 | 0.735 | 0.680 | 0.705 | 0.704 |

Table 31: Comparison of different network configurations for early interaction models with **node-level granularity** across splits of the AIDS dataset with **increasing edge count**, using mean average precision (MAP). Green and yellow cells indicate the subsets where the model trained on the full dataset performs best and second best respectively.

| Rel. Dist. | Structure | Non-linearity | AIDS | Subset 0 | Subset 1 | Subset 2 | Subset 3 |
|---|---|---|---|---|---|---|---|
| Agg-hinge | Non-injective | Dot Product | 0.7 | 0.667 | 0.662 | 0.708 | 0.742 |
| Agg-hinge | Non-injective | Hinge | 0.763 | 0.784 | 0.754 | 0.773 | 0.774 |
| Agg-hinge | Non-injective | Neural | 0.677 | 0.642 | 0.617 | 0.693 | 0.723 |
| Agg-hinge | Injective | Dot Product | 0.755 | 0.763 | 0.742 | 0.770 | 0.765 |
| Agg-hinge | Injective | Hinge | 0.758 | 0.781 | 0.742 | 0.769 | 0.775 |
| Agg-hinge | Injective | Neural | 0.68 | 0.628 | 0.645 | 0.689 | 0.724 |
| Agg-MLP | Non-injective | Dot Product | 0.677 | 0.619 | 0.617 | 0.670 | 0.721 |
| Agg-MLP | Non-injective | Hinge | 0.748 | 0.767 | 0.731 | 0.751 | 0.770 |
| Agg-MLP | Non-injective | Neural | 0.662 | 0.592 | 0.619 | 0.665 | 0.715 |
| Agg-MLP | Injective | Dot Product | 0.758 | 0.778 | 0.737 | 0.762 | 0.773 |
| Agg-MLP | Injective | Hinge | 0.789 | 0.837 | 0.786 | 0.797 | 0.784 |
| Agg-MLP | Injective | Neural | 0.703 | 0.685 | 0.676 | 0.710 | 0.736 |
| Agg-NTN | Non-injective | Dot Product | 0.71 | 0.713 | 0.678 | 0.714 | 0.741 |
| Agg-NTN | Non-injective | Hinge | 0.79 | 0.842 | 0.786 | 0.794 | 0.794 |
| Agg-NTN | Non-injective | Neural | 0.701 | 0.663 | 0.671 | 0.711 | 0.744 |
| Agg-NTN | Injective | Dot Product | 0.74 | 0.718 | 0.725 | 0.749 | 0.762 |
| Agg-NTN | Injective | Hinge | 0.76 | 0.785 | 0.747 | 0.759 | 0.781 |
| Agg-NTN | Injective | Neural | 0.689 | 0.669 | 0.642 | 0.695 | 0.731 |
| Set align. | Non-injective | Dot Product | 0.715 | 0.656 | 0.664 | 0.695 | 0.755 |
| Set align. | Non-injective | Hinge | 0.783 | 0.805 | 0.750 | 0.790 | 0.786 |
| Set align. | Non-injective | Neural | 0.708 | 0.661 | 0.670 | 0.709 | 0.749 |
| Set align. | Injective | Dot Product | 0.798 | 0.829 | 0.782 | 0.796 | 0.805 |
| Set align. | Injective | Hinge | 0.817 | 0.879 | 0.812 | 0.827 | 0.801 |
| Set align. | Injective | Neural | 0.725 | 0.716 | 0.677 | 0.716 | 0.749 |

Table 32: Comparison of different network configurations for early interaction models with **edge-level granularity** across splits of the AIDS dataset with **increasing edge count**, using mean average precision (MAP). Green and yellow cells indicate the subsets where the model trained on the full dataset performs best and second best respectively.

## I.2 NODE COUNT

In Tables 33 and 34, we observe: **(1)** Across splits with increasing node count, performance consistently improves for all network configurations. This can be explained by the presence of padding nodes in the injective / non-injective maps

| Rel. Dist. | Structure | Non-linearity | AIDS | Subset 0 | Subset 1 | Subset 2 | Subset 3 |
|---|---|---|---|---|---|---|---|
| Agg-hinge | Non-injective | Dot Product | 0.609 | 0.531 | 0.586 | 0.623 | 0.670 |
| Agg-hinge | Non-injective | Hinge | 0.726 | 0.687 | 0.714 | 0.724 | 0.758 |
| Agg-hinge | Non-injective | Neural | 0.598 | 0.511 | 0.577 | 0.603 | 0.660 |
| Agg-hinge | Injective | Dot Product | 0.64 | 0.594 | 0.629 | 0.652 | 0.687 |
| Agg-hinge | Injective | Hinge | 0.662 | 0.596 | 0.660 | 0.671 | 0.701 |
| Agg-hinge | Injective | Neural | 0.614 | 0.526 | 0.580 | 0.638 | 0.674 |
| Agg-MLP | Non-injective | Dot Product | 0.63 | 0.543 | 0.616 | 0.637 | 0.691 |
| Agg-MLP | Non-injective | Hinge | 0.637 | 0.567 | 0.620 | 0.653 | 0.685 |
| Agg-MLP | Non-injective | Neural | 0.629 | 0.540 | 0.603 | 0.634 | 0.689 |
| Agg-MLP | Injective | Dot Product | 0.658 | 0.614 | 0.658 | 0.674 | 0.693 |
| Agg-MLP | Injective | Hinge | 0.683 | 0.660 | 0.688 | 0.685 | 0.711 |
| Agg-MLP | Injective | Neural | 0.629 | 0.566 | 0.607 | 0.634 | 0.683 |
| Agg-NTN | Non-injective | Dot Product | 0.669 | 0.579 | 0.643 | 0.683 | 0.721 |
| Agg-NTN | Non-injective | Hinge | 0.686 | 0.614 | 0.643 | 0.692 | 0.745 |
| Agg-NTN | Non-injective | Neural | 0.635 | 0.543 | 0.603 | 0.642 | 0.707 |
| Agg-NTN | Injective | Dot Product | 0.721 | 0.665 | 0.714 | 0.726 | 0.757 |
| Agg-NTN | Injective | Hinge | 0.743 | 0.684 | 0.749 | 0.757 | 0.767 |
| Agg-NTN | Injective | Neural | 0.667 | 0.593 | 0.639 | 0.675 | 0.721 |
| Set align. | Non-injective | Dot Product | 0.608 | 0.533 | 0.594 | 0.623 | 0.662 |
| Set align. | Non-injective | Hinge | 0.676 | 0.633 | 0.672 | 0.697 | 0.727 |
| Set align. | Non-injective | Neural | 0.593 | 0.536 | 0.576 | 0.608 | 0.644 |
| Set align. | Injective | Dot Product | 0.71 | 0.678 | 0.704 | 0.709 | 0.744 |
| Set align. | Injective | Hinge | 0.734 | 0.723 | 0.746 | 0.732 | 0.747 |
| Set align. | Injective | Neural | 0.69 | 0.652 | 0.676 | 0.703 | 0.723 |

Table 33: Comparison of different network configurations for early interaction models with **node-level granularity** across splits of the AIDS dataset with **increasing node count**, using mean average precision (MAP). Green and yellow cells indicate the subsets where the model trained on the full dataset performs best and second best respectively.

| Rel. Dist. | Structure | Non-linearity | AIDS | Subset 0 | Subset 1 | Subset 2 | Subset 3 |
|---|---|---|---|---|---|---|---|
| Agg-hinge | Non-injective | Dot Product | 0.7 | 0.633 | 0.666 | 0.706 | 0.751 |
| Agg-hinge | Non-injective | Hinge | 0.763 | 0.736 | 0.759 | 0.771 | 0.790 |
| Agg-hinge | Non-injective | Neural | 0.677 | 0.604 | 0.643 | 0.686 | 0.730 |
| Agg-hinge | Injective | Dot Product | 0.755 | 0.706 | 0.751 | 0.758 | 0.784 |
| Agg-hinge | Injective | Hinge | 0.758 | 0.716 | 0.753 | 0.765 | 0.791 |
| Agg-hinge | Injective | Neural | 0.68 | 0.602 | 0.660 | 0.688 | 0.731 |
| Agg-MLP | Non-injective | Dot Product | 0.677 | 0.584 | 0.649 | 0.683 | 0.732 |
| Agg-MLP | Non-injective | Hinge | 0.748 | 0.699 | 0.736 | 0.759 | 0.786 |
| Agg-MLP | Non-injective | Neural | 0.662 | 0.565 | 0.636 | 0.676 | 0.721 |
| Agg-MLP | Injective | Dot Product | 0.758 | 0.695 | 0.742 | 0.765 | 0.793 |
| Agg-MLP | Injective | Hinge | 0.789 | 0.754 | 0.791 | 0.790 | 0.813 |
| Agg-MLP | Injective | Neural | 0.703 | 0.635 | 0.687 | 0.714 | 0.742 |
| Agg-NTN | Non-injective | Dot Product | 0.71 | 0.660 | 0.696 | 0.704 | 0.752 |
| Agg-NTN | Non-injective | Hinge | 0.79 | 0.765 | 0.783 | 0.796 | 0.814 |
| Agg-NTN | Non-injective | Neural | 0.701 | 0.630 | 0.677 | 0.704 | 0.755 |
| Agg-NTN | Injective | Dot Product | 0.74 | 0.676 | 0.726 | 0.744 | 0.777 |
| Agg-NTN | Injective | Hinge | 0.76 | 0.715 | 0.750 | 0.762 | 0.794 |
| Agg-NTN | Injective | Neural | 0.689 | 0.615 | 0.666 | 0.694 | 0.741 |
| Set align. | Non-injective | Dot Product | 0.715 | 0.612 | 0.698 | 0.710 | 0.767 |
| Set align. | Non-injective | Hinge | 0.783 | 0.725 | 0.764 | 0.786 | 0.816 |
| Set align. | Non-injective | Neural | 0.708 | 0.615 | 0.680 | 0.722 | 0.755 |
| Set align. | Injective | Dot Product | 0.798 | 0.760 | 0.781 | 0.807 | 0.820 |
| Set align. | Injective | Hinge | 0.817 | 0.798 | 0.813 | 0.822 | 0.830 |
| Set align. | Injective | Neural | 0.725 | 0.653 | 0.696 | 0.731 | 0.756 |

Table 34: Comparison of different network configurations for early interaction models with **edge-level granularity** across splits of the AIDS dataset with **increasing node count**, using mean average precision (MAP). Green and yellow cells indicate the subsets where the model trained on the full dataset performs best and second best respectively.

