# OpenReview forum: "Charting the Design Space of Neural Graph Representations for Subgraph Matching"
_ICLR.cc/2025/Conference — ICLR 2025 Poster_

### Official Review · Reviewer_VJh8 · 2024-10-28

**Soundness:** 2
**Presentation:** 4
**Contribution:** 3
**Rating:** 6
**Confidence:** 4

**Summary:**

This paper studies the problem of neural network based subgraph matching, focusing on several model/system design choices, e.g. early vs late interaction, trainable vs fixed non-linearity, etc. A set of guidelines and best practices are outlined which are the key contributions of this work.

**Strengths:**

1. The overall problem formulation is interesting and timely considering the various design choices in subgraph matching literature.
2. Ample experimental study is performed adding credibility and rigorousity to the findings and guidelines.
3. Error bar is shown for the results, e.g. in Figure 2.

**Weaknesses:**

1. It seems all the 10 datasets are relatively small, e.g. up to 50 nodes. I wonder if there is a reason for not choosing much larger graphs, e.g. graphs up to 1M nodes as the target graph to be searched for (while using a small query graph of ~50 nodes).
2. Some of the paper writing can be made more clearer. E.g. “Challenging the widely
held expectation that early interaction is more powerful, IsoNet’s late interaction approach outperforms GMN, even when GMN’s final score computation is made asymmetric” – It is a bit unclear what it means by “asymmetric”, especially due to the fact this is written at the beginning of intro and the audience may not be familiar with this area of research.
3. It may be useful to introduce/outline the summarized findings at the end of the introduction section.

**Questions:**

1. How is this work related to neural architecture search (NAS)? It might be worth comparing/surveting works in NAS, e.g. https://arxiv.org/pdf/2403.05064v1, https://openreview.net/pdf?id=GcM7qfl5zY, etc. Referencing such works can position this work better within the broader literature and bring out further discussion.

---

> ### Author Response · Authors · 2024-11-25
> **Response to Reviewer VJh8 (Part 1)**
>
> We thank the reviewer for their insightful review. The weaknesses and questions raised in the review are addressed below.
>
> > It seems all the 10 datasets are relatively small, e.g. up to 50 nodes. I wonder if there is a reason for not choosing much larger graphs, e.g. graphs up to 1M nodes as the target graph to be searched for (while using a small query graph of ~50 nodes).
>
> There are two key reasons why we could not work with large graphs in the original version. But now, we have made strong evaluations on large graphs as well.
>
> Similar to other ML tasks, our work also requires multiple graphs for training; here, one entire graph represents one instance of data. Data sets with a single large graph are more common than data sets containing multiple large graphs, which prevents a neural method to test its performance on large graphs. Moreover, the need for our and existing works are driven by practical applications like molecular retrieval. For instance, works such as GMN [Li et al., 2019], Neuromatch [Lou et al., 2020] and IsoNet [Roy et al., 2021] focus on small- to medium-sized graphs in their benchmarks. This scale is largely motivated by application domains such as molecular graph retrieval and object detection in images via scene graphs, where the focus is more on retrieving multiple smaller graphs rather than processing single extremely large graphs.
>
> Still, we consider the following datasets drawn from the SNAP repository (https://snap.stanford.edu/data/index.html):
>
> 1. com-Amazon: Represents an Amazon product co-purchasing network, with nodes as products and edges denoting co-purchasing relationships. It consists of 334,863 nodes and 925,872 edges.
> 2. email-Enron: Represents an email communication network, with nodes as individuals and edges as email exchanges. It consists of 36,692 nodes and 183,831 edges.
> 3. roadnet-CA: Represents the road network of California, with nodes as road intersections and edges as connecting roads. It consists of 1,965,206 nodes and 2,766,607 edges.
>
> Recent work such as Greed [Ranjan et al., 2022] proposes a heuristic for adapting small-scale, fast neural graph solvers to the problem of subgraph localization within large-scale graphs. Inspired by this approach, we extended our experiments on the above datasets.
>
>
> Tables 1–4 present the results in terms of MAP for all proposed variations discussed in our study, corresponding to Node-Late, Edge-Late, Node-Early, and Edge-Early approaches, respectively.
>
> **Table 1:** MAP  for **Node-Late** interaction models for three diverse datasets.
> |Rel. Dist.|Structure|Non-linearity|Amazon|Email|Roadnet|
> |-|-|-|-|-|-|
> |Agg-hinge|NA|NA|0.616|0.723|0.556|
> |Agg-MLP|NA|NA|0.544|0.631|0.616|
> |Agg-NTN|NA|NA|0.656|0.778|0.499|
> |Set align|Non-injective|Dot Product|0.66|0.798|0.618|
> |Set align|Non-injective|Hinge|0.739|0.826|0.609|
> |Set align|Non-injective|Neural|0.752|0.805|0.585|
> |Set align|Injective|Dot Product|0.739|0.852|**0.637**|
> |Set align|Injective|Hinge|**0.772**|**0.864**|0.576|
> |Set align|Injective|Neural|0.771|0.852|0.605|
>
> **Table 2:** MAP  for **Edge-Late** interaction models for three diverse datasets.
>
> |Rel. Dist.|Structure|Non-linearity|Amazon|Email|Roadnet|
> |-|-|-|-|-|-|
> |Agg-hinge|NA|NA|0.692|0.847|**0.744**|
> |Agg-MLP|NA|NA|0.652|0.783|0.682|
> |Agg-NTN|NA|NA|0.678|0.851|0.706|
> |Set align.|Non-injective|Dot Product|0.737|0.834|0.621|
> |Set align.|Non-injective|Hinge|0.776|0.865|0.741|
> |Set align.|Non-injective|Neural|0.771|0.838|0.72|
> |Set align.|Injective|Dot Product|0.749|0.852|0.668|
> |Set align.|Injective|Hinge|**0.800**|0.849|0.708|
> |Set align.|Injective|Neural|0.788|**0.871**|0.724|

---

> ### Author Response · Authors · 2024-11-25
> **Response to Reviewer VJh8 (Part 2)**
>
> **Table 3:** MAP  for **Node-Early** interaction models for three diverse datasets.
>
> |Rel. Dist.  |Structure   |Non-linearity|Amazon|Email|Roadnet|
> |-----------------|-----------------|---------------|--------|-------|---------|
> |Agg-MLP     |Injective   |Hinge     |0.805|0.902|0.680|
> |Agg-MLP     |Injective   |Dot Product|0.725|0.911|0.643|
> |Agg-MLP     |Injective   |Neural    |0.798|0.895|0.668|
> |Agg-MLP     |Non-Injective|Hinge     |0.770|0.869|0.646|
> |Agg-MLP     |Non-Injective|Dot Product|0.720|0.817|0.648|
> |Agg-MLP     |Non-Injective|Neural    |0.748|0.834|0.586|
> |Agg-NTN     |Injective   |Hinge     |0.798|0.906|0.699|
> |Agg-NTN     |Injective   |Dot Product|0.789|0.902|0.669|
> |Agg-NTN     |Injective   |Neural    |0.787|0.920|0.732|
> |Agg-NTN     |Non-Injective|Hinge     |0.768|0.855|0.668|
> |Agg-NTN     |Non-Injective|Dot Product|0.668|0.829|0.612|
> |Agg-NTN     |Non-Injective|Neural    |0.695|0.839|0.645|
> |Agg-hinge   |Injective   |Hinge     |0.756|0.866|0.659|
> |Agg-hinge   |Injective   |Dot Product|0.748|0.861|0.682|
> |Agg-hinge   |Injective   |Neural    |0.743|0.862|0.660|
> |Agg-hinge   |Non-Injective|Hinge     |0.799|0.878|0.662|
> |Agg-hinge   |Non-Injective|Dot Product|0.657|0.764|0.610|
> |Agg-hinge   |Non-Injective|Neural    |0.676|0.759|0.592|
> |Set align   |Injective   |Hinge     |**0.849**|**0.935**|**0.745**|
> |Set align   |Injective   |Dot Product|0.816|0.905|0.706|
> |Set align   |Injective   |Neural    |0.823|0.905|0.723|
> |Set align   |Non-Injective|Hinge     |0.768|0.860|0.652|
> |Set align   |Non-Injective|Dot Product|0.729|0.827|0.648|
> |Set align   |Non-Injective|Neural    |0.747|0.842|0.648|
>
> **Table 4:** MAP  for **Edge-Early** interaction models for three diverse datasets.
> |Rel. Dist.|Structure  |Non-linearity|Amazon|Email|Roadnet|
> |---------------|----------------|---------------|--------|--------|---------|
> |Agg-MLP   |Injective  |Hinge     |0.84|0.926|0.773|
> |Agg-MLP   |Injective  |Dot Product|0.799|0.934|0.76|
> |Agg-MLP   |Injective  |Neural    |0.72|0.91|0.74|
> |Agg-MLP   |Non-Injective|Hinge     |0.801|0.886|0.671|
> |Agg-MLP   |Non-Injective|Dot Product|0.74|0.837|0.745|
> |Agg-MLP   |Non-Injective|Neural    |0.703|0.869|0.613|
> |Agg-NTN   |Injective  |Hinge     |0.791|0.922|0.731|
> |Agg-NTN   |Injective  |Dot Product|0.783|0.916|0.76|
> |Agg-NTN   |Injective  |Neural    |0.789|0.908|0.737|
> |Agg-NTN   |Non-Injective|Hinge     |0.795|0.868|0.816|
> |Agg-NTN   |Non-Injective|Dot Product|0.765|0.874|0.77|
> |Agg-NTN   |Non-Injective|Neural    |0.701|0.853|0.708|
> |Agg-hinge |Injective  |Hinge     |0.827|0.939|0.805|
> |Agg-hinge |Injective  |Dot Product|0.77|0.915|0.805|
> |Agg-hinge |Injective  |Neural    |0.753|0.911|0.768|
> |Agg-hinge |Non-Injective|Hinge     |0.797|0.874|0.73|
> |Agg-hinge |Non-Injective|Dot Product|0.736|0.845|0.607|
> |Agg-hinge |Non-Injective|Neural    |0.69|0.842|0.699|
> |Set align |Injective  |Hinge     |**0.863**|**0.944**|**0.834**|
> |Set align |Injective  |Dot Product|0.827|0.921|0.828|
> |Set align |Injective  |Neural    |0.826|0.909|0.752|
> |Set align |Non-Injective|Hinge     |0.783|0.875|0.832|
> |Set align |Non-Injective|Dot Product|0.802|0.872|0.802|
> |Set align |Non-Injective|Neural    |0.769|0.858|0.74|
>
>
>
>
> We note that the performance trends observed on the small molecule datasets are consistent with those on these diverse real-world datasets.
> In particular, we note that:
> 1. Edge-level granularity outperforms node-level granularity, as evidenced by comparisons between corresponding cells in Table 1 (Node-Late) and Table 2 (Edge-Late), as well as Table 3 (Node-Early) and Table 4 (Edge-Early).
> 2. Early interaction variants (Tables 3 and 4) demonstrate significantly higher MAP values compared to late-interaction models (Tables 1 and 2).
> 3. Across all four tables, Set Alignment relevance distance consistently achieves the highest MAP values in most cases.
> 4. Injective interaction structure variants generally provide higher MAP values compared to their non-injective counterparts across all tables.
> 5. The combination of Set Alignment with an injective interaction structure and Hinge interaction non-linearity is particularly effective, yielding the highest MAP values for these datasets
>
>
> The performance on subgraph localization tasks on these larger real-world datasets, aligning with the performance trends on the smaller and medium-sized datasets, reinforces the generality of our findings and conclusions.
>
> These results further suggest that our observations can be extended to heuristics operating on much larger graphs. While this is not the primary focus of our study, these experiments highlight the broader applicability of our approach to large graph scenarios.

---

> ### Author Response · Authors · 2024-11-25
> **Response to Reviewer VJh8 (Part 3)**
>
> > “Challenging the widely held expectation that early interaction is more powerful, IsoNet’s late interaction approach outperforms GMN, even when GMN’s final score computation is made asymmetric” – It is a bit unclear what it means by “asymmetric”, especially due to the fact this is written at the beginning of intro and the audience may not be familiar with this area of research.
>
> Thanks for the suggestion ― indeed the reference to "asymmetric" was too early and cryptic; we have removed it.
>
> > Some of the paper writing can be made more clearer. It may be useful to introduce/outline the summarized findings at the end of the introduction section.
>
> We have added the following para at the end of the intro. In addition, we have clearly summarized the effect of each design axis at the end of the respective sections. These are highlighted in the PDF for convenience.
>
> **💡 Key takeaways and design tips** Our systematic navigation of the design space resolves hitherto unexplained observations and provides reliable guidelines for future methods.
> 1. We conclusively explain (late-interaction) IsoNet’s earlier-observed superiority over (early-interaction) GMN. If GMN’s early interaction is supplemented with any of set alignment, injective structure, hinge nonlinearity, or edge-based interaction, it can readily outperform IsoNet.
> 2. These five design principles are vital, and their combination unveils a novel graph retrieval model that surpasses all existing methods.
> 3. Shifting from late to early interaction may increase computational cost, but compensates for the limitations of relevance distance defined using aggregated single-vector embeddings.
>
> > How is this work related to neural architecture search (NAS)? It might be worth comparing/surveting works in NAS, e.g. https://arxiv.org/pdf/2403.05064v1, https://openreview.net/pdf?id=GcM7qfl5zY, etc. Referencing such works can position this work better within the broader literature and bring out further discussion.
>
> We thank the reviewer for referring us to the relevant domain of Neural Architecture Search. Upon further study, we have decided to add the following discussion to our Related Works section and cite the two papers as references.
>
> Within specific families of graph encoder networks, such as GNNs/GATs [NAS 1] or graph transformers [NAS 2], researchers have proposed super-networks to explore the parametric space of encoder networks, using a bi-level optimization framework. We have discussed and referenced the papers our revised version. Thanks for bringing forward these related work. It would be of future interest to investigate if NAS methods can be extended to subgraph search and other combinatorial graph problems, to automatically explore network design spaces.
>
> [NAS 1, Zhang et. al.] [Unsupervised Graph Neural Architecture Search with Disentangled Self-supervision](https://arxiv.org/pdf/2403.05064v1). *Neural Information Processing Systems, 2023*
>
> [NAS 2, Zhang et. al.] [AutoGT: Automated Graph Transformer Architecture Search](https://openreview.net/pdf?id=GcM7qfl5zY). *International Conference on Learning Representations, 2023*

---

### Official Review · Reviewer_SvpM · 2024-10-31

**Soundness:** 3
**Presentation:** 2
**Contribution:** 2
**Rating:** 5
**Confidence:** 4

**Summary:**

The paper presents an exploration of the design space for neural subgraph matching, where subcomponents have been presented in existing research. The authors observe that prior methods for neural graph matching have been narrowly focused, with most methods falling into limited design categories. To address this, the authors organize a comprehensive framework including key architectural choices: relevance distance, interaction stages, interactions structures, interaction non-linearity, and interaction granularity. Extensive experiments demonstrate that unexplored combinations within this design space can lead to performance improvement.

**Strengths:**

- The authors show a detailed and thorough literature survey on neural subgraph matching and introduce general subcomponents for subgraph matching models.
- The authors conduct extensive experiments on all possible combinations within design spaces.
- The authors discover a new combination of subcomponents that outperforms the existing state-of-the-art model.

**Weaknesses:**

-  While the paper does a thorough job of exploring the design space or structured ablation studies, it does not provide a principled explanation of why certain configurations (or subcomponents) yield performance improvements. Adding a more in-depth empirical and theoretical analysis can address this issue. This can not only strengthen the contributions but also offer a more principled foundation for future research in this area.
- Although the best combination shows performance benefits, its novelty is limited. The only differences between the best model and the existing state-of-the-art (IsoNet) are the interaction stage and non-linearity (which have been proposed in existing studies). Without providing principled rationales (as the first bullet), this could make the contributions appear incremental.
- The dense presentation makes it challenging for readers to immediately see the paper’s structure and key subcomponents. I think the authors should present a clearer organization and structured outline. In particular, Figure 1 is hard to interpret. I think that the authors can provide more human-readable visualization. Plus, please put some margin to the bottom of Figure 1.

**Questions:**

.

---

> ### Author Response · Authors · 2024-11-25
> **Response to Reviewer SvpM (Part 1)**
>
> >While the paper does a thorough job of exploring the design space or structured ablation studies, it does not provide a principled explanation of why certain configurations (or subcomponents) yield performance improvements. Adding a more in-depth empirical and theoretical analysis can address this issue. This can not only strengthen the contributions but also offer a more principled foundation for future research in this area.
>
> We appreciate this insightful feedback and acknowledge the importance of providing a more principled explanation of the performance improvements observed in the various configurations. In response, we have made an effort to distill the key empirical observations and present them succinctly. Additionally, we highlight the key takeaways from these analyses. A detailed discussion on the design choice axes is presented separately below.
>
> ### *Relevance Distance: Choice of  Agg-\* v/s  Set Aligned*
> We present a comparison between set alignment-based relevance distance and aggregated embedding-based distance. Table 1 shows the MAP for Node-Late interaction models, while Table 2 presents the MAP for Edge-Late interaction models. The Agg-* variants rely on aggregated graph-level embeddings, which are subsequently processed through either an NTN module, a hinge scoring layer, or a general MLP. The interaction structure and non-linearities employed by the set alignment relevance distance are specified in brackets. In most cases, the set alignment relevance distance outperforms the aggregated distance variants.
>
> **Table 1:** MAP  for **Node-Late** interaction models for the first three datasets.
> | Rel. Dist. (Structure, Non-linearity)    | AIDS  | Mutag | FM   |
> |------------------------------------------|-------|-------|------|
> | Agg-hinge                                | 0.557 | 0.594 | 0.636|
> | Agg-MLP                                  | 0.548 | 0.64  | 0.674|
> | Agg-NTN                                  | 0.576 | **0.708** | 0.744|
> | Set align. (Injective, Hinge)            | 0.633 | 0.647 | 0.72 |
> | Set align. (Injective, Neural)           | **0.664** | 0.69  | **0.758**|
>
>
> **Table 2:** MAP  for **Edge-Late** interaction models for the first three datasets.
> | Rel. Dist. (Structure, Non-linearity)     | AIDS  | Mutag | FM   |
> |-------------------------------------------|-------|-------|------|
> | Agg-hinge                                | 0.635 | 0.694 | 0.712|
> | Agg-MLP                                  | 0.607 | 0.63  | 0.727|
> | Agg-NTN                                  | 0.66  | 0.718 | 0.759|
> | Set align. (Injective, Hinge)            | **0.712** | 0.721 | 0.793|
> | Set align. (Injective, Neural)           | 0.704 | **0.733** | **0.782**|
>
> **Takeaways and Design Tips**
>
> Compressing the entire graph into a low-dimensional vector can result in information loss. Therefore, comparing the node embeddings at the set-level granularity yields better performance than their single-vector representations. Similar observations have been reported in other domains.
>
> In knowledge graph alignment, encoding the neighborhood of an entity as a set and then aligning two such sets has been found to perform better than comparing compressed single-vector representations of each entity [BERT-INT].
>
> In textual entailment [Lai et al., 2017; Chen et al., 2020; Bevan et al., 2023], allowing cross-interaction between all tokens of the two sentences is generally more effective than compressing each sentence into a single vector and comparing them.
>
> Our work reconfirms this intuition for subgraph retrieval as well.
>
>
> [BERT-INT] X. Tang, J. Zhang, B. Chen, Y. Yang, H. Chen, and C. Li. Bert-int: A bert-based interaction model for knowledge graph alignment. interactions, 100:e1, 2020.
>
> [Lai et al., 2017] A. Lai and J. Hockenmaier. Learning to predict denotational probabilities for modeling entailment.
>
> [Chen et al., 2020] T. Chen, Z. Jiang, A. Poliak, K. Sakaguchi, and B. Van Durme. Uncertain natural language inference.
>
> [Bevan et al., 2023] R. Bevan, O. Turbitt, and M. Aboshokor. Mdc at semeval-2023 task 7: Fine-tuning transformers for textual entailment prediction and evidence retrieval in clinical trials

---

> ### Author Response · Authors · 2024-11-26
> **Response to Reviewer SvpM (Part 2)**
>
> ### *Interaction Stage: Choice of  Late v/s Early*
> We compare late and early interaction variants for both the top-performing aggregated embedding-based method (Agg-NTN) and the best-performing set alignment-based method. Table 3 presents the results for Node interaction granularity, while Table 4 shows the results for Edge interaction granularity. Early interaction is seen to afford significant performance improvements over its late interaction counterparts.
>
> **Table 3:** MAP  for **Node** interaction models for the first three datasets.
> |Interaction Stage (Rel. Dist, Structure, Non-linearity) | AIDS  | Mutag | FM   |
> |-------------------------------------------|-------|-------|------|
> | Late (Agg-NTN)                           | 0.576 | 0.708 | 0.744|
> | Early (Agg-NTN)                           |  **0.743** | **0.791** | **0.828**|
> |-------------------------------------------|-------|-------|------|
> | Late (Set align.)                         | 0.664 | 0.69  | 0.758|
> | Early (Set align.)                        | **0.734** | **0.783**  | **0.834**|
>
> **Table 4:** MAP  for **Edge** interaction models for the first three datasets.
> |Interaction Stage (Rel. Dist) | AIDS  | Mutag | FM   |
> |-------------------------------------------|-------|-------|------|
> | Late (Agg-NTN)              | 0.66  | 0.718 | 0.759|
> | Early (Agg-NTN)              |**0.768**  | **0.755** | **0.800**|
> |-------------------------------------------|-------|-------|------|
> | Late (Set align.)            | 0.704 | 0.733 | 0.782|
> | Early (Set align.)           |**0.774**  | **0.764** | **0.806**|
>
> **Takeaways and Design Tips**
> Although late interaction potentially enables fast nearest neighbor search, early interaction is generally known to be superior in text retrieval [ColBERT, Figure~1]. The comparison between IsoNet [ISONET] vs GMN [GMN] apparently contradicts this general trend. Therefore, it is important to resolve this issue using carefully controlled experiments.
>
> [ColBERT] O. Khattab and M. Zaharia. Colbert: Efficient and effective passage search via contextualized late interaction over bert.
>
> [ISONET] I. Roy, V. S. Velugoti, S. Chakrabarti, and A. De. Interpretable neural subgraph matching for graph retrieval.
>
> [GMN] Y. Li, C. Gu, T. Dullien, O. Vinyals, and P. Kohli. Graph matching networks for learning the similarity
> of graph structured objects
>
> ### *Interaction Structure: Injective v/s non-Injective*
> We compare injective and non-injective interaction variants for both the top-performing aggregated embedding-based method (Agg-NTN) and the best-performing set alignment-based method. Table 5 presents the results for **Node-Early** variants, while Table 6 shows the results for **Edge-Early** variants. Injective interaction structure is seen to afford significant performance improvements over non-injective interaction in a majority of cases. Furthermore, we observe that the performance improvement from switching to injective interaction is more pronounced in set alignment relevance distance-based methods compared to aggregated NTN-based methods.
>
>
> **Table 5:** MAP  for **Node-Early** interaction models for the first three datasets.
> |Interaction Structure (Rel. Dist) | AIDS  | Mutag | FM   |
> |--------------------------------------------|------|-------|-------|
> | Non-Injective (Agg-NTN)                   |0.686 | 0.758 | 0.818 |
> | Injective (Agg-NTN)                        |**0.743** | **0.791** | **0.828** |
> |-------------------------------------------|-------|-------|------|
> | Non-Injective (Set align.)                 |0.676 | 0.754 | 0.772 |
> | Injective (Set align.)                     |**0.734** | **0.783** | **0.834** |
>
>
> **Table 6:** MAP  for **Edge-Early** interaction models for the first three datasets.
> |Interaction Structure (Rel. Dist) | AIDS  | Mutag | FM   |
> |--------------------------------------------|------|-------|-------|
> | Non-Injective (Agg-NTN)                   | **0.790**  | **0.785**  | 0.835  |
> | Injective (Agg-NTN)                        | 0.760 | 0.783  | **0.855**  |
> |-------------------------------------------|-------|-------|------|
> | Non-Injective (Set align.)                | 0.783 | 0.785  | 0.812  |
> | Injective (Set align.)                     | **0.817** | **0.837**  |  **0.887** |
>
> **Takeaways and Design Tips**
> The combinatorial definition of graph matching includes finding an injective mapping between pairs of nodes from the two graphs.  The mapping is also an interpretable artifact.
> Attention from one node to all nodes in the other graph, even if maintained from each graph separately, cannot achieve a consistent 1-1 mapping.  Our experiments suggest that an injective mapping (or its continuous relaxation --- doubly stochastic matrices) performs better.

---

> > ### Author Response · Authors · 2024-11-26
> > **Response to Reviewer SvpM (Part 3)**
> >
> > ### *Interaction non-linearity: Neural vs. Dot Product vs. Hinge*
> >
> > We compare the three types of interaction non-linearities -- Dot Product, Hinge and Neural -- for both the top-performing aggregated embedding-based method (Agg-NTN) and the best-performing set alignment-based method. Table 7 presents the results for **Node-Early-Injective** variants, while Table 8 shows the results for **Edge-Early-Injective** variants. Hinge non-linearity is seen to be the best performer for Set alignment driven variants, while Neural is seen to be generally the best for Aggregated NTN based variants.
> >
> > **Table 7:** MAP  for **Node-Early-Injective** interaction models for the first three datasets.
> > |  Non-linearity (Rel. Dist.)     | AIDS   | Mutag  | FM    |
> > |-------------------------------------------|--------|--------|-------|
> > | Dot Product (Agg-NTN)        | 0.721  | 0.772  | 0.812 |
> > | Hinge (Agg-NTN)              | **0.743**  | 0.773  | 0.805 |
> > | Neural (Agg-NTN)             | 0.667  | **0.791**  | **0.828** |
> > |-------------------------------------------|-------|-------|------|
> > | Dot Product (Set align)     | 0.71   | 0.779  | 0.8   |
> > | Hinge (Set align)             | **0.734**  | 0.774  | **0.834** |
> > | Neural (Set align.)          | 0.69   | **0.783**  | 0.827 |
> >
> > **Table 8:** MAP  for **Edge-Early-Injective** interaction models for the first three datasets.
> > | Non-linearity (Rel. Dist.)         | AIDS   | Mutag  | FM    |
> > |-------------------------------------------|--------|--------|-------|
> > | Dot Product (Agg-NTN)         | 0.74   | **0.783**  | 0.812 |
> > | Hinge (Agg-NTN)               | **0.76**   | 0.766  | **0.855** |
> > | Neural (Agg-NTN)              | 0.689  | 0.749  | 0.831 |
> > |-------------------------------------------|-------|-------|------|
> > | Dot Product (Set align)       | 0.798  | 0.789  | 0.862 |
> > | Hinge (Set align)             | **0.817**  | **0.837**  | **0.887** |
> > | Neural (Set align.)           | 0.725  | 0.784  | 0.837 |
> >
> > **Takeaways and Design Tips**
> > Set alignment provides a more interpretable approach for measuring coverage in subgraph matching, where explicit coverage modeling using hinge non-linearity emerges as the top performer by directly encoding the necessary inductive bias. In contrast, methods that rely on "black-box" techniques, such as NTN, which use aggregated graph embeddings, appear to align better with neural-based models.
> >
> >
> >
> > ### *Interaction Granularuty : Node vs Edge*
> > We compare two interaction granularities—nodes v/s edges --  focusing on the most promising design choices over the previous axes: Relevance Distance (Set Align.), Interaction Structure (Injective), and Interaction Stage (Early). Table 9 presents the best-performing configurations in terms of MAP for both node and edge granularities across all interaction non-linearities. Notably, edge granularity-based variants demonstrate a significant performance improvement.
> >
> > **Table 9:** MAP  for **Early-Injective-SetAlign** interaction models for the first three datasets.
> > | Granularity  | AIDS   | Mutag  | FM    |
> > |-------------------------------------------|--------|--------|-------|
> > | Node            | 0.734 | 0.783 | 0.834 |
> > | Edge             | **0.817** | **0.837** | **0.887** |
> >
> >
> > **Takeaways and Design Tips**
> > The significant performance boost observed with edge granularity highlights the effectiveness of utilizing higher-order structures. Consider prioritizing edge granularity in designs to enhance performance, and exploring other higher-order alignments to enhance model expressiveness and accuracy in graph-based applications.
> >
> >
> >
> > We have also included clearly defined sections throughout the main paper, providing a more principled outline and explanation of the key takeaways and design tips.

---

> ### Author Response · Authors · 2024-11-26
> **Response to Reviewer SvpM (Part 4)**
>
> > Although the best combination shows performance benefits, its novelty is limited. The only differences between the best model and the existing state-of-the-art (IsoNet) are the interaction stage and non-linearity (which have been proposed in existing studies). Without providing principled rationales (as the first bullet), this could make the contributions appear incremental.
>
>
> Our key contribution is not providing another new model. We rather identify the key design choices and while doing that, we demystify many existing works, as reviewer 6vg7 also pointed out. While there are so many excellent papers, there is a lack of study why some methods work and some methods don't.  We find out key design choices involving the representation of and interaction between graph representations, and then systematically explore subtle interplay between these design choices.
>
> Our systematic navigation of the design space resolves hitherto- unexplained observations and provides reliable guidelines for future methods. (1) We conclusively explain (late-interaction) IsoNet’s earlier-observed superiority over (early-interaction) GMN. If GMN’s early interaction is supplemented with any of set alignment, injective structure, hinge nonlinearity, or edge-based interaction, it can readily outperform IsoNet. (2) These five design principles are vital, and their combination unveils a novel graph retrieval model that surpasses all existing methods. (3) Shifting from late to early interaction may increase computational cost, but compensates for the limitations of relevance distance defined using aggregated single-vector embeddings.
>
>
>
>
> >The dense presentation makes it challenging for readers to immediately see the paper’s structure and key subcomponents. I think the authors should present a clearer organization and structured outline. In particular, Figure 1 is hard to interpret. I think that the authors can provide more human-readable visualization. Plus, please put some margin to the bottom of Figure 1.
>
> We appreciate the feedback and have made significant improvements to enhance clarity. Figure 1 has been revised to provide a more readable and intuitive visualization, with an updated caption for better interpretation. We have also restructured the treatment of each design axis and included a dedicated section summarizing the key takeaways.

---

> ### Author Response · Authors · 2024-11-28
> **Incorporating suggestions of Reviewer SvpM**
>
> Many thanks for your feedback. We now better understand your concerns, which we address below.
>
> > *Novelty and surprise in results:*
>
> Fig 3 suggests that (1) components interact; (2) components that are suboptimal within the individual axes may be combined to give high accuracy in fast inference time. This figure clearly guides a practitioner to choose the model combination based the inference time cost permissible in the underlying usecase, rather than trying all possible combinations.
> We provide some novel and surprising observations as follows:
>
> (1) Components do interact. Suppose, we just switch one axis (relevance distance) from the optimal combination:  **SA**+Early+Injective+Hinge+Edge  to **Agg-NTN**+Early+Injective+Hinge+Edge, performance degrades drastically (by 7%). We expected that if we further change choice across another axis, MAP would decrease more. Contrary to this expectation, when we change injective to non-injective (**Agg-NTN**+Early+**Non-Injective**+Hinge+Edge), the performance actually boosts significantly. Moreover, inference becomes much faster (2X faster than the best MAP setup). This was indeed surprising to us, since Injective mapping is a key criteria which is advocated by IsoNet. But it turns out Injectivity only shows its effect if set alignment is the final score.
>
> However, injective maps may not always be the preferred solution, because the Sinkhorn network consumes large compute resources.  The designer may choose non-injective mapping which may result in considerable speed up. In this case, Agg-NTN+Early+Non-Injective+Hinge+Edge may be the best design.
>
> (2) In principle, an MLP with cascaded layers of Linear-ReLU-Linear networks should be able to learn any underlying nonlinearity, certainly simple nonlinearities like hinge. Hence, if we change  SA+Early+Injective+Hinge+Edge  to SA+Early+Injective+Neural+Edge, we expect comparable  performance . However, the performance drops drastically (by 11.3%). This is surprising since SA+Early+Injective+Neural+Edge is the superclass of the hypotheses modeled using  SA+Early+Injective+Hinge+Edge.
>
> (3) Most surprisingly, in IsoNet, injective vs Non-injective does not make any difference. See the bottom left cluster in scatter plot Fig 3: (Late interaction, set alignment, edge (also node)), where injective and non-injective points are overlapping.
>
> Moreover, IsoNet claimed that edge alignment model, set alignment and injective mapping are the key components for subgraph matching model. They claimed that due to these factors, GMN are outperformed by them even if we use asymmetric scores in GMN. However, as the table here (also page 22, Table4) suggests, if we just change GMN’s nonlinearity to hinge, then it performs better than Isonet, despite use of **node alignment, aggregated scoring (no set alignment) and non-injective mapping.**
>
> |Dataset|IsoNet|GMN(DP → Hinge)|
> |-|-|-|
> |AIDS|0.704|**0.726**|
> |Mutag|**0.733**|0.723|
> |FM|0.782|**0.79**|
> |NCI|0.615|**0.618**|
> |MOLT|0.649|**0.651**|
> |FR|0.734|**0.75**|
>
> > *Leveraging prior studies' ablation findings could have narrowed the search space*
>
> As we have seen in the above table Node alignment, non-injective mapping, aggregated-hinge with hinge nonlinearity work better than IsoNet’s edge alignment. Hence, IsoNet’s claim that node alignment is always worse than edge alignment is not true and therefore, narrowing down the search from related work may not be helpful.
>
> > *differences between your best and IsoNet are two*
>
> The best model reported in the paper corresponds to early interaction with injective mapping. This is extremely costly and it is not preferred by the practitioner where the usecase involves time constraint. Apart from this expensive combination, Fig 3 in our paper allows for a wide variety of combinations, catering to various inference time costs. Under such time constraint, one has to perform changes across several axes of IsoNet, which still outperforms IsoNet.
> Following table shows that changes across multiple axes are necessary to beat Isonet in comparable run-time.
> **Our-early-1 is close to our-early-best but more than 2x faster,  and it is different from IsoNet in four out of five axes.**
> It provides excellent trade off between accuracy and time.
>
> | |Rel. Dist.|Stage|Structure|Non-linearity|Granularity|Aids|Mutag|FM|NCI|Inf Time (ms)|
> |-|-|-|-|-|-|-|-|-|-|-|
> |IsoNet|SA|Late|Injective|Neural|Edge|0.704|0.733|0.782|0.615|23 ms|
> |Our-early-1|Agg-NTN|Early|Non-Injective|Hinge|Edge|0.79|0.785|0.835|0.661|28 ms|
> |Our-early-2|Agg-Hinge|Early|Non-Injective|Hinge|Node|0.726|0.723|0.79|0.618|26 ms|
> |Our-early-best|SA|Early|Injective|Hinge|Edge|0.817|0.837|0.887|0.677|58 ms|
>
> We are keen to hear from you on whether your concerns have been addressed.

---

> ### Author Response · Authors · 2024-12-01
> **Further discussion based on  suggestions (Part 1)**
>
> > *Beyond educated guesses, what are the fundamental reasons why a certain combination performs so well*
>
> We answer this question directly from the following theoretical underpinning. Specifically, we describe the combinatorial formulation of subgraph matching and then show why certain combination works so well and others don't.
>
> First consider the combinatorial cost:
> $$
> \begin{align}
> \text{dist}(G _q , G _c )  = \min _{P}
> \sum _{u,v}  [\big(A _q-P A _c P^{\top}\big) _+] [u,v]---(a) \\\\
> P \text{ is a permutation matrix}---(b)
> \end{align}
> $$
> The standard way to minimize it is a Projected gradient descent approach, where P is updated as
> $$    P _{k} \leftarrow  \text{argmin}  _{P} \textrm{Trace}\left(P^T\nabla _{P}\ \text{dist}(G  _q , G  _c )  \big| _{P = P _{k-1}}\right)
> --- (c)$$
>
> ---
>
> ### Relaxation of (a--c)
>
> A neural approx should ideally relax all three steps (a--c), keeping each relaxation close to the original combinatorial equation, to get the highest benefit from perspective of inductive bias, which will allow more interpretable and accurate neural model. We will now show relaxation of *all* of them naturally leads to the best combination of our framework-- however, almost all other deviations do not follow these relaxation principles meticulously which is why, they show suboptimal performance.
>
> **R1: Relaxing $\text{dist}(G _q , G _c )$ in Eq (a):** As mentioned in L204-207, Eq (a) aims to solve a quadratic assignment problem (QAP) which is NP-Hard. Hence, we convert them into a more tractable linear assignment problem as
>
> $$\min _{P} \sum _{u,i}  [\big(H _q-P H _c \big) _+] [u,i]  \ \  (\text{\textcolor{blue}{  This naturally gives set alignment based distance}})---- (a1).$$
>
> **R2: Relaxing $\text{dist}(G _q , G _c )$ in Eq (b):**  Sinkhorn iterations give natural extensions to the Permutation matrices, $\text{\textcolor{blue}{which is injective mapping}}$.
>
> Now, how we can solve Eq (a1) using R1 and R2? As we mentioned in Appendix D.1 (repeated as follows), **proves that hinge is correct nonlinearity**.
> First note that:
> $$\min _{P} \sum _{u,i}  [\big(H _q-P H _c \big) _+] [u,i] =  \min _{P} \sum _{u,i}  \big(H _q[u,i]- H _c [v,i]\big)  _+ P[u,v] \quad \textbf{ (since P is permutation matrix)}
> $$ Given B is the set of of doubly stochastic matrices and $\epsilon \to 0$, Eq. a1 directly reduces to:
> $$ \min _{P\in B} \sum _{u} \underbrace{\sum _{i}  \big(H _q[u,i]- H _c [v,i]\big) _+]} _{C} P[u,v] + \epsilon \sum _{u,v} P[u,v]\log P[u,v]---(a2)$$
> Hence, P is $Z_T$ where $Z _0 = \exp(C / \tau)$ and $Z_t$ is computed using sinkhorn iterations:
> $$
> Z _{t+1}[u, u'] = \frac{Z _t[u, u']}{\sum _{v' \in [N]} Z_t[u, v']}, \quad
> \text{where} \quad Z _t'[u, u'] = \frac{Z _t[u, u']}{\sum _{v \in [N]} Z _t[v, u']}, \quad \text{for all } (u, u')
> $$
> Eq (a2) shows the $\text{\textcolor{blue}{nonlinearity in C must be hinge}}$. This is one of simple yet novel quick and dirty trick, which significantly boosts performance of any model. Keeping everything suboptimal, if we use it in GMN, it outperforms IsoNet. This provides a principled explanation on why hinge is better than dot product and neural method.
>
> **R3: relaxing Eq c:**  One can always argue that update of $P$ is already being performed by the above updates on Z. But, if you look carefully, Eq. c involves $\nabla_{P}\text{dist}(G  _q , G  _c )  \big| _{P = P _{k-1}}$, which indicates that in principle, C should depend on $P$. However, since the approximation (a1) is linear, C becomes independent of P.  $\text{\textcolor{blue}{To get past this crude approximation, one should make C dependent on P, which implies that $H _q$ and $H _c$ should be }}$   $\text{\textcolor{blue}{dependent on $P$, which again implies that $H_q$ and $H_c$ should become dependent on each other, giving to an early }}$   $\text{\textcolor{blue}{interaction model. }}$
>
> This clearly shows why early interaction model, hinge nonlinearity, injective mapping, set alignment work well.

---

> ### Author Response · Authors · 2024-12-01
> **Further discussion based on suggestions (Part 2)**
>
> > Academic values derived from underperforming combinations
>
>
> The key underperforming combinations are as follows:
>
> 1. Poor design choices for $\nabla _P \text{dist}(G _q, G _c)$ significantly impact performance. The widely used Dot Product interaction (commonly found in attention-based models like GMN and H2MN) and the neural interaction non-linearity introduced by IsoNet are consistently outperformed by the Hinge non-linearity. Hinge offers a clear advantage across all metrics and is the preferred choice for interaction modeling.
>
> 2. Using dot product or neural methods, pivoting to set alignment relevance distance improves MAP values. However, this comes at the cost of a significant increase in inference time, making it suboptimal for achieving the best MAP-inference time Pareto trade-off.
>
> 3. In the above, while using dot product or neural methods, transitioning to non-injective set alignment in an effort to improve inference times results in a notable performance drop. This approach fails to provide a satisfactory trade-off between accuracy and efficiency.
>
> 4. When using Early Interaction with Hinge non-linearity and non-injective interaction, the choice of relevance distance becomes less critical. In such cases, opting for faster aggregated relevance variants over set alignment achieves similar performance while reducing computational overhead.
>
> We elaborate on both underperforming and well-performing combinations in the **Takeaways and Design Tips** (pages 5–7) and the **Design Guidelines** (pages 9–10). Please refer to the text highlighted in teal for detailed insights.
>
>
> > On another note, I recently came across the following study: https://openreview.net/forum?id=udTwwF7tks Could you let me know which combination in your submission this model aligns with?
>
>
> If cast into our specified design axes, this paper employs Early Interaction, with an injective interaction structure, neural interaction non-linearity, and set alignment relevance distance. However, their early interaction model is more sophisticated than prior works, factoring in node pairs and updating alignments lazily at fixed intervals. While this approach appears to improve MAP, it is computationally extremely expensive, requiring further investigation to address: (1) How this novel interaction model interacts with other design choices, such as whether hinge non-linearity could further enhance performance or interact adversely; (2) Its position in the MAP-inference time Pareto trade-off compared to other design choices.
>
> > Of course, this is a neurips 2024 paper, so it does not necessarily have to be included in this submission, but it would be nice to have it mentioned later. And I am also planning to compare the results in the paper and your submission.
>
> We agree this study is relevant to our submission. However, as it became publicly available only recently (just three weeks before today and more than one month later than submission deadline). Comparing the results of this paper against ours is beyond the scope of our current work. We will certainly mention clearly later. Notably, this paper underscores the importance of our current submission in the rapidly evolving landscape of subgraph matching methods.
>
> As with many recent proposals, this paper appears to treat its specific design choices as achieving a robust (local) maximum in performance. However, without systematically incorporating superior design options—such as injective interaction, neural non-linearity, and edge granularity—for baselines like GMN, it may be challenging to attribute performance gains solely to novel aspects like node-pair attention and lazy updates. This highlights the need for a principled framework to evaluate and benchmark design choices systematically.
>
> Our work addresses this need by proposing clear guidelines for evaluating new models and understanding the combinatorial interplay of design choices. We believe this will inspire more confidence and measurable progress in subgraph matching research.
>
>
>
> We reiterate that our proposal is not in competition with this new paper and this new paper does not diminish the significance of our work. Instead, this paper aligns with our specified design axes and can further advance the field when analyzed using our proposed guideline. In fact, our analysis can be used to enhance the quality of this paper too.
>
>
>
>
> ### Appeal to Reviewer SvpM
>
> We greatly appreciate your continued engagement in improving this work. We has earlier highlighted the key takeaways and design tips throughout the paper and elaborated on the theoretical underpinnings of hinge non-linearity in Section D.1 of the Appendix. Your feedback has helped us refine these insights more coherently. While we are unable to update the paper PDF at this stage, we will ensure these improvements are incorporated in final revision, if the paper gets accepted. Kindly let us know if this further discussion has addressed your concerns.

---

### Official Review · Reviewer_6vg7 · 2024-11-02

**Soundness:** 3
**Presentation:** 4
**Contribution:** 3
**Rating:** 8
**Confidence:** 3

**Summary:**

This paper presents a unified design space for existing neural subgraph matching methods over various dimensions. By carefully controlling the design option for each dimension, it effectively reveals the optimal design option combination for neural subgraph matching with substantial improvement, providing new insights on the previous methods and reported results.

**Strengths:**

**S1.** The charting of the design space is convincing and well-articulated.

**S2.** Empirical studies demonstrate the effectiveness of the design space in characterizing existing methods and devising superior new methods for the experiments considered.

**S3.** The presentation is quite neat and clear overall.

**Weaknesses:**

**W1.** Most datasets considered are for small molecules despite the various applications of subgraph matching mentioned in the introduction.

**W2.** There is a lack of understanding in subgraph matching performance with respect to the intrinsic challenge of the setting, e.g., highly regular graphs, different graph sizes, different feature distributions, out-of-distribution settings. Such study may be best achieved with synthetic datasets.

**minor**
- The use of notations $\omega$ and $\eta$ are not consistent between L124 and Figure 1.

**Questions:**

**Q1.** Does the use of hinge distance make sense for set alignment and aggregated-hinge (L216-236)? In particular, the authors justify the use of hinge distance for adjacency matrices, but the adjacency matrices are binary while node/aggregated node embeddings are just real-valued.

---

> ### Author Response · Authors · 2024-11-25
> **Response to Reviewer 6vg7 (Part 1)**
>
> We thank the reviewer for their detailed review and valuable feedback that helps improve our work. We address the concerns posed by the reviewer below:
> > W1. Most datasets considered are for small molecules despite the various applications of subgraph matching mentioned in the introduction.
>
> While the datasets currently included in the paper are widely recognized benchmarks, the reviewer correctly highlights that most of these are sourced from small molecule datasets (e.g., AIDS, MUTAG) and computer vision tasks (e.g., MSRC). We acknowledge this limitation and have extended our dataset suite to include three large-scale, real-world datasets that reflect diverse and practical applications of subgraph matching. These datasets were obtained from the SNAP repository (https://snap.stanford.edu/data/index.html) and were processed to align with the experimental setup described in our study. The details of the newly added datasets are as follows:
>
> 1. com-Amazon (Amazon): This dataset represents an Amazon product co-purchasing network, where nodes correspond to products and edges denote frequent co-purchasing relationships. Subgraph matching in this dataset is relevant for tasks such as identifying frequent co-purchase patterns or improving recommendation systems.
>
> 1. email-Enron (Email): This dataset comprises an email communication network from the Enron corporation, where nodes represent individuals and edges correspond to email exchanges. Subgraph matching in this context can provide insights into communication patterns and aid in identifying specific organizational structures.
>
> 1. roadnet-CA (RoadNet): This dataset represents the highway network of California, where nodes correspond to road intersections, and edges denote roads connecting them. Subgraph matching in this setting can assist in applications such as traffic management and subnetwork optimization.
>
> To integrate these datasets into our study, we followed the same preprocessing and subgraph extraction methodology outlined in the paper and ran all the designed experiments to further  explore the design space of the subgraph matching methods.
>
> Tables 1–4 present the results in terms of MAP for all proposed variations discussed in our study, corresponding to Node-Late, Edge-Late, Node-Early, and Edge-Early approaches, respectively.
>
> **Table 1:** MAP  for **Node-Late** interaction models for three diverse datasets.
> | Rel. Dist.      | Structure       | Non-linearity | Amazon | Email | Roadnet |
> |------------------|-----------------|---------------|--------|-------|---------|
> | Agg-hinge       | NA              | NA            | 0.616  | 0.723 | 0.556   |
> | Agg-MLP         | NA              | NA            | 0.544  | 0.631 | 0.616   |
> | Agg-NTN         | NA              | NA            | 0.656  | 0.778 | 0.499   |
> | Set align       | Non-injective   | Dot Product   | 0.66   | 0.798 | 0.618   |
> | Set align       | Non-injective   | Hinge         | 0.739  | 0.826 | 0.609   |
> | Set align       | Non-injective   | Neural        | 0.752  | 0.805 | 0.585   |
> | Set align       | Injective       | Dot Product   | 0.739  | 0.852 | **0.637**   |
> | Set align       | Injective       | Hinge         | **0.772**  | **0.864** | 0.576   |
> | Set align       | Injective       | Neural        | 0.771  | 0.852 | 0.605   |
>
> **Table 2:** MAP  for **Edge-Late** interaction models for three diverse datasets.
>
> | Rel. Dist.    | Structure      | Non-linearity | Amazon | Email  | Roadnet |
> |---------------|----------------|---------------|--------|--------|---------|
> | Agg-hinge     | NA             | NA        | 0.692  | 0.847  | **0.744**   |
> | Agg-MLP       | NA             | NA            | 0.652  | 0.783  | 0.682   |
> | Agg-NTN       | NA             | NA            | 0.678  | 0.851  | 0.706   |
> | Set align.    | Non-injective  | Dot Product   | 0.737  | 0.834  | 0.621   |
> | Set align.    | Non-injective  | Hinge         | 0.776  | 0.865  | 0.741   |
> | Set align.    | Non-injective  | Neural        | 0.771  | 0.838  | 0.72    |
> | Set align.    | Injective      | Dot Product   | 0.749  | 0.852  | 0.668   |
> | Set align.    | Injective      | Hinge         | **0.800**    | 0.849  | 0.708   |
> | Set align.    | Injective      | Neural        | 0.788  | **0.871**  | 0.724   |

---

> ### Author Response · Authors · 2024-11-25
> **Response to Reviewer 6vg7 (Part 2)**
>
> **Table 3:** MAP  for **Node-Early** interaction models for three diverse datasets.
>
> | Rel. Dist.      | Structure       | Non-linearity | Amazon | Email | Roadnet |
> |------|---------|-------|--------|-------|------|
> | Agg-MLP  | Injective| Hinge  | 0.805  | 0.902 | 0.680   |
> | Agg-MLP  | Injective| Dot Product   | 0.725  | 0.911 | 0.643   |
> | Agg-MLP | Injective | Neural | 0.798  | 0.895 | 0.668   |
> | Agg-MLP | Non-Injective   | Hinge| 0.770  | 0.869 | 0.646   |
> | Agg-MLP | Non-Injective   | Dot Product   | 0.720  | 0.817 | 0.648   |
> | Agg-MLP | Non-Injective   | Neural| 0.748  | 0.834 | 0.586   |
> | Agg-NTN | Injective| Hinge| 0.798  | 0.906 | 0.699   |
> | Agg-NTN | Injective| Dot Product   | 0.789  | 0.902 | 0.669   |
> | Agg-NTN | Injective| Neural| 0.787  | 0.920 | 0.732   |
> | Agg-NTN | Non-Injective   | Hinge| 0.768  | 0.855 | 0.668   |
> | Agg-NTN | Non-Injective   | Dot Product   | 0.668  | 0.829 | 0.612   |
> | Agg-NTN | Non-Injective   | Neural| 0.695  | 0.839 | 0.645   |
> | Agg-hinge| Injective| Hinge| 0.756  | 0.866 | 0.659   |
> | Agg-hinge| Injective| Dot Product   | 0.748  | 0.861 | 0.682   |
> | Agg-hinge| Injective| Neural| 0.743  | 0.862 | 0.660   |
> | Agg-hinge| Non-Injective| Hinge| 0.799  | 0.878 | 0.662   |
> | Agg-hinge| Non-Injective| Dot Product   | 0.657  | 0.764 | 0.610   |
> | Agg-hinge| Non-Injective| Neural| 0.676  | 0.759 | 0.592   |
> | Set align| Injective| Hinge|**0.849**  |**0.935** | **0.745**   |
> | Set align| Injective| Dot Product   | 0.816  | 0.905 | 0.706   |
> | Set align| Injective| Neural| 0.823  | 0.905 | 0.723   |
> | Set align| Non-Injective| Hinge| 0.768  | 0.860 | 0.652   |
> | Set align| Non-Injective| Dot Product   | 0.729  | 0.827 | 0.648   |
> | Set align| Non-Injective| Neural| 0.747  | 0.842 | 0.648   |
>
> **Table 4:** MAP  for **Edge-Early** interaction models for three diverse datasets.
> | Rel. Dist.    | Structure      | Non-linearity | Amazon | Email  | Roadnet |
> |---------------|----------------|---------------|--------|--------|---------|
> | Agg-MLP| Injective| Hinge| 0.84   | 0.926  | 0.773   |
> | Agg-MLP| Injective| Dot Product   | 0.799  | 0.934  | 0.76    |
> | Agg-MLP| Injective| Neural| 0.72   | 0.91   | 0.74    |
> | Agg-MLP| Non-Injective  | Hinge| 0.801  | 0.886  | 0.671   |
> | Agg-MLP| Non-Injective  | Dot Product   | 0.74   | 0.837  | 0.745   |
> | Agg-MLP| Non-Injective  | Neural| 0.703  | 0.869  | 0.613   |
> | Agg-NTN| Injective| Hinge| 0.791  | 0.922  | 0.731   |
> | Agg-NTN| Injective| Dot Product   | 0.783  | 0.916  | 0.76    |
> | Agg-NTN| Injective| Neural| 0.789  | 0.908  | 0.737   |
> | Agg-NTN| Non-Injective  | Hinge | 0.795  | 0.868  | 0.816   |
> | Agg-NTN| Non-Injective  | Dot Product   | 0.765  | 0.874  | 0.77    |
> | Agg-NTN| Non-Injective  | Neural| 0.701  | 0.853  | 0.708   |
> | Agg-hinge| Injective| Hinge | 0.827  | 0.939  | 0.805   |
> | Agg-hinge| Injective| Dot Product   | 0.77   | 0.915  | 0.805   |
> | Agg-hinge| Injective| Neural| 0.753  | 0.911  | 0.768   |
> | Agg-hinge| Non-Injective  | Hinge| 0.797  | 0.874  | 0.73    |
> | Agg-hinge| Non-Injective  | Dot Product   | 0.736  | 0.845  | 0.607   |
> | Agg-hinge| Non-Injective  | Neural| 0.69   | 0.842  | 0.699   |
> | Set align| Injective| Hinge | **0.863**  | **0.944**  | **0.834**   |
> | Set align| Injective| Dot Product   | 0.827  | 0.921  | 0.828   |
> | Set align| Injective| Neural | 0.826  | 0.909  | 0.752   |
> | Set align| Non-Injective  | Hinge| 0.783  | 0.875  | 0.832   |
> | Set align| Non-Injective  | Dot Product   | 0.802  | 0.872  | 0.802   |
> | Set align| Non-Injective  | Neural| 0.769  | 0.858  | 0.74    |
>
> We note that the performance trends observed on the small molecule datasets are consistent with those on these diverse real-world datasets.
> In particular, we note that:
> 1. Edge-level granularity outperforms node-level granularity, as evidenced by comparisons between corresponding cells in Table 1 (Node-Late) and Table 2 (Edge-Late), as well as Table 3 (Node-Early) and Table 4 (Edge-Early).
> 2. Early interaction variants (Tables 3 and 4) demonstrate significantly higher MAP values compared to late-interaction models (Tables 1 and 2).
> 3. Across all four tables, Set Alignment relevance distance consistently achieves the highest MAP values in most cases.
> 4. Injective interaction structure variants generally provide higher MAP values compared to their non-injective counterparts across all tables.
> 5. The combination of Set Alignment with an injective interaction structure and Hinge interaction non-linearity is particularly effective, yielding the highest MAP values for these datasets
>
> This consistency reinforces the generality of the observations and conclusions outlined in the paper.
>
> By incorporating these diverse datasets, we address the reviewer’s concerns regarding the representativeness of our experimental suite. We believe this extension strengthens the study and aligns it more closely with the practical applications of subgraph matching discussed in the introduction.

---

> ### Author Response · Authors · 2024-11-25
> **Response to Reviewer 6vg7 (Part 3)**
>
> > W2. There is a lack of understanding in subgraph matching performance with respect to the intrinsic challenge of the setting, e.g., highly regular graphs, different graph sizes, different feature distributions, out-of-distribution settings. Such study may be best achieved with synthetic datasets.
>
> We thank the reviewer for pointing us in the very interesting direction of studying different graph sizes, feature distributions and out-of-distribution settings. Towards this goal, we formula two new experiments:
>
> **Transfer ability (out-of-distribution setting)**
>
> We choose the model trained on dataset X and perform inference on dataset Y, where X ≠ Y. In particular, we fix Y to be the AIDS dataset and vary X in the set {Mutag, PTC-FR, MOLT-4H}. We compare the performance of these OOD models with the model trained on the AIDS dataset itself. For clarity of understanding, we show results on early interaction models with node-level granularity (Table 5) and edge-level granularity (Table 6). **Boldface** represents the dataset that provides the best transfer ability. The following observations can be made -
>
> 1. The strongest transfer abilities are displayed by models trained on PTC-FR, followed by Mutag, and finally MOLT-4H, which shows severe degradation in performance compared to the baseline AIDS-based model. This pattern can be explained by the extent of relative dissimilarity between the source datasets and AIDS, which is maximum for MOLT-4H (doubly-sized graphs as AIDS), followed by Mutag (mean node/edge counts off by one/two) and finally PTC-FR (similar graph size distribution).
> 2. Despite the strong transfer ability, the model trained originally on AIDS is the best performer with almost every network configuration.
>
> **Table 5:** MAP for transfer ability of different **Node-Early** interaction models on the AIDS dataset.
>
> |Rel. Dist. |Structure |Non-linearity |AIDS |Mutag → AIDS |PTC-FR → AIDS |MOLT → AIDS |
> |-|-|--|-|-|-|-|
> |Agg-Hinge|Non-injective|Dot Product|0.609|0.330|**0.520**|0.253|
> |Agg-Hinge|Non-injective|Hinge|0.726|0.360|**0.626**|0.369|
> |Agg-Hinge|Non-injective|Neural|0.598|0.331|**0.510**|0.238|
> |Agg-Hinge|Injective|Dot Product|0.64|0.394|**0.581**|0.322|
> |Agg-Hinge|Injective|Hinge|0.662|0.422|**0.595**|0.346|
> |Agg-Hinge|Injective|Neural|0.614|0.403|**0.598**|0.322|
> |Agg-MLP|Non-injective|Dot Product|0.63|0.323|**0.569**|0.363|
> |Agg-MLP|Non-injective|Hinge|0.637|0.378|**0.624**|0.437|
> |Agg-MLP|Non-injective|Neural|0.629|0.363|**0.552**|0.337|
> |Agg-MLP|Injective|Dot Product|0.658|0.486|**0.612**|0.427|
> |Agg-MLP|Injective|Hinge|0.683|0.457|**0.611**|0.455|
> |Agg-MLP|Injective|Neural|0.629|0.464|**0.625**|0.414|
> |Agg-NTN|Non-injective|Dot Product|0.669|0.450|**0.557**|0.407|
> |Agg-NTN|Non-injective|Hinge|0.686|0.442|**0.623**|0.405|
> |Agg-NTN|Non-injective|Neural|0.635|0.367|**0.549**|0.268|
> |Agg-NTN|Injective|Dot Product|0.721|0.487|**0.635**|0.420|
> |Agg-NTN|Injective|Hinge|0.743|0.478|**0.626**|0.356|
> |Agg-NTN|Injective|Neural|0.667|0.499|**0.617**|0.395|
> |Set align|Non-injective|Dot Product|0.608|0.377|**0.574**|0.419|
> |Set align|Non-injective|Hinge|0.676|0.394|**0.600**|0.451|
> |Set align|Non-injective|Neural|0.593|0.450|**0.599**|0.363|
> |Set align|Injective|Dot Product|0.71|0.510|**0.663**|0.448|
> |Set align|Injective|Hinge|0.734|0.505|**0.684**|0.480|
> |Set align|Injective|Neural|0.69|0.507|**0.641**|0.494|
>
> **Table 6:** MAP for transfer ability of different **Edge-Early** interaction models on the AIDS dataset.
> |Rel. Dist.|Structure|Non-linearity|AIDS|Mutag → AIDS|PTC-FR → AIDS|MOLT → AIDS|
> |-|-|-|-|-|-|-|
> |Agg-Hinge|Non-injective|Dot Product|0.7|0.400|**0.560**|0.290|
> |Agg-Hinge|Non-injective|Hinge|0.763|0.445|**0.640**|0.447|
> |Agg-Hinge|Non-injective|Neural|0.677|0.414|**0.541**|0.219|
> |Agg-Hinge|Injective|Dot Product|0.755|0.454|**0.657**|0.305|
> |Agg-Hinge|Injective|Hinge|0.758|0.477|**0.664**|0.274|
> |Agg-Hinge|Injective|Neural|0.68|0.422|**0.578**|0.278|
> |Agg-MLP|Non-injective|Dot Product|0.677|0.451|**0.589**|0.341|
> |Agg-MLP|Non-injective|Hinge|0.748|0.423|**0.619**|0.337|
> |Agg-MLP|Non-injective|Neural|0.662|0.396|**0.563**|0.351|
> |Agg-MLP|Injective|Dot Product|0.758|0.454|**0.662**|0.373|
> |Agg-MLP|Injective|Hinge|0.789|0.491|**0.718**|0.425|
> |Agg-MLP|Injective|Neural|0.703|0.419|**0.620**|0.312|
> |Agg-NTN|Non-injective|Dot Product|0.71|0.444|**0.603**|0.399|
> |Agg-NTN|Non-injective|Hinge|0.79|0.489|**0.658**|0.241|
> |Agg-NTN|Non-injective|Neural|0.701|0.419|**0.572**|0.361|
> |Agg-NTN|Injective|Dot Product|0.74|0.466|**0.623**|0.367|
> |Agg-NTN|Injective|Hinge|0.76|0.518|**0.714**|0.431|
> |Agg-NTN|Injective|Neural|0.689|0.409|**0.599**|0.320|
> |Set align|Non-injective|Dot Product|0.715|0.474|**0.668**|0.374|
> |Set align|Non-injective|Hinge|0.783|0.483|**0.687**|0.497|
> |Set align|Non-injective|Neural|0.708|0.429|**0.594**|0.412|
> |Set align|Injective|Dot Product|0.798|0.505|**0.695**|0.447|
> |Set align|Injective|Hinge|0.817|0.599|**0.773**|0.538|
> |Set align|Injective|Neural|0.725|0.468|**0.599**|0.428|

---

> ### Author Response · Authors · 2024-11-25
> **Response to Reviewer 6vg7 (Part 4)**
>
> **Variation in performance with latent dataset characteristics**
>
> To study how dataset characteristics affect our proposed networks, we create splits of our datasets and perform inference on them. In particular, we select a metric and divide the corpus set of the AIDS dataset sorted by this metric into 4 contiguous splits. Inference is performed with all query graphs using the model trained on AIDS, on each of the corpus splits and the MAP scores are reported. Subset 0 represents the split with graphs pertaining to the lowest values for the corresponding metric while Subset 3 represents that with the highest values.
>
> In the tables below, **boldface** represents the subset with the highest MAP score for the corresponding row.
>
> (1) **Metric = Node Count**
>
> Node count is one measure of graph size and is extremely relevant to the problem at hand, since a smaller node count requires a larger number of padding nodes to allow batched processing, which can be detrimental towards performance. In Table 7, we display dataset statistics for each split. The statistics are shown for the corpus graphs only, since the query graphs used for inference are identical across all splits.
>
> **Table 7:** Dataset statistics for splits of the AIDS dataset based on the Node Count metric.
> |Datset|Avg. Node Count (Corpus)|
> |-|-|
> |AIDS (original)|18.50|
> |Subset 0|17.00|
> |Subset 1|17.96|
> |Subset 2|19.02|
> |Subset 3|20.00|
>
> **Observation:** The effect of padding nodes is clearly noticeable for both node-early (Table 8) and edge-early (Table 9) models. Subset 0 (minimum number of nodes) corresponds to the smallest MAP score, which consistently increases as we increase node size and peaks for Subset 3.
>
> **Table 8:** MAP for **Node-Early** interaction models for splits of the dataset with increasing **node count**
> |Rel. Dist.|Structure|Non-linearity|AIDS|Subset 0|Subset 1|Subset 2|Subset 3|
> |-|-|-|-|-|-|-|-|
> |Agg-Hinge|Non-inj.|Dot Product|0.609|0.531|0.586|0.623|**0.670**|
> |Agg-Hinge|Non-inj.|Hinge|0.726|0.687|0.714|0.724|**0.758**|
> |Agg-Hinge|Non-inj.|Neural|0.598|0.511|0.577|0.603|**0.660**|
> |Agg-Hinge|Inj.|Dot Product|0.64|0.594|0.629|0.652|**0.687**|
> |Agg-Hinge|Inj.|Hinge|0.662|0.596|0.660|0.671|**0.701**|
> |Agg-Hinge|Inj.|Neural|0.614|0.526|0.580|0.638|**0.674**|
> |Agg-MLP|Non-inj.|Dot Product|0.63|0.543|0.616|0.637|**0.691**|
> |Agg-MLP|Non-inj.|Hinge|0.637|0.567|0.620|0.653|**0.685**|
> |Agg-MLP|Non-inj.|Neural|0.629|0.540|0.603|0.634|**0.689**|
> |Agg-MLP|Inj.|Dot Product|0.658|0.614|0.658|0.674|**0.693**|
> |Agg-MLP|Inj.|Hinge|0.683|0.660|0.688|0.685|**0.711**|
> |Agg-MLP|Inj.|Neural|0.629|0.566|0.607|0.634|**0.683**|
> |Agg-NTN|Non-inj.|Dot Product|0.669|0.579|0.643|0.683|**0.721**|
> |Agg-NTN|Non-inj.|Hinge|0.686|0.614|0.643|0.692|**0.745**|
> |Agg-NTN|Non-inj.|Neural|0.635|0.543|0.603|0.642|**0.707**|
> |Agg-NTN|Inj.|Dot Product|0.721|0.665|0.714|0.726|**0.757**|
> |Agg-NTN|Inj.|Hinge|0.743|0.684|0.749|0.757|**0.767**|
> |Agg-NTN|Inj.|Neural|0.667|0.593|0.639|0.675|**0.721**|
> |Set align|Non-inj.|Dot Product|0.608|0.533|0.594|0.623|**0.662**|
> |Set align|Non-inj.|Hinge|0.676|0.633|0.672|0.697|**0.727**|
> |Set align|Non-inj.|Neural|0.593|0.536|0.576|0.608|**0.644**|
> |Set align|Inj.|Dot Product|0.71|0.678|0.704|0.709|**0.744**|
> |Set align|Inj.|Hinge|0.734|0.723|0.746|0.732|**0.747**|
> |Set align|Inj.|Neural|0.69|0.652|0.676|0.703|**0.723**|
>
> **Table 9:** MAP for **Edge-Early** interaction models for splits of the dataset with increasing **node count**
> |Rel. Dist.|Structure|Non-linearity|AIDS|Subset 0|Subset 1|Subset 2|Subset 3|
> |-|-|-|-|-|-|-|-|
> |Agg-Hinge|Non-inj.|Dot Product|0.7|0.633|0.666|0.706|**0.751**|
> |Agg-Hinge|Non-inj.|Hinge|0.763|0.736|0.759|0.771|**0.790**|
> |Agg-Hinge|Non-inj.|Neural|0.677|0.604|0.643|0.686|**0.730**|
> |Agg-Hinge|Inj.|Dot Product|0.755|0.706|0.751|0.758|**0.784**|
> |Agg-Hinge|Inj.|Hinge|0.758|0.716|0.753|0.765|**0.791**|
> |Agg-Hinge|Inj.|Neural|0.68|0.602|0.660|0.688|**0.731**|
> |Agg-MLP|Non-inj.|Dot Product|0.677|0.584|0.649|0.683|**0.732**|
> |Agg-MLP|Non-inj.|Hinge|0.748|0.699|0.736|0.759|**0.786**|
> |Agg-MLP|Non-inj.|Neural|0.662|0.565|0.636|0.676|**0.721**|
> |Agg-MLP|Inj.|Dot Product|0.758|0.695|0.742|0.765|**0.793**|
> |Agg-MLP|Inj.|Hinge|0.789|0.754|0.791|0.790|**0.813**|
> |Agg-MLP|Inj.|Neural|0.703|0.635|0.687|0.714|**0.742**|
> |Agg-NTN|Non-inj.|Dot Product|0.71|0.660|0.696|0.704|**0.752**|
> |Agg-NTN|Non-inj.|Hinge|0.79|0.765|0.783|0.796|**0.814**|
> |Agg-NTN|Non-inj.|Neural|0.701|0.630|0.677|0.704|**0.755**|
> |Agg-NTN|Inj.|Dot Product|0.74|0.676|0.726|0.744|**0.777**|
> |Agg-NTN|Inj.|Hinge|0.76|0.715|0.750|0.762|**0.794**|
> |Agg-NTN|Inj.|Neural|0.689|0.615|0.666|0.694|**0.741**|
> |Set align|Non-inj.|Dot Product|0.715|0.612|0.698|0.710|**0.767**|
> |Set align|Non-inj.|Hinge|0.783|0.725|0.764|0.786|**0.816**|
> |Set align|Non-inj.|Neural|0.708|0.615|0.680|0.722|**0.755**|
> |Set align|Inj.|Dot Product|0.798|0.760|0.781|0.807|**0.820**|
> |Set align|Inj.|Hinge|0.817|0.798|0.813|0.822|**0.830**|
> |Set align|Inj.|Neural|0.725|0.653|0.696|0.731|**0.756**|

---

> ### Author Response · Authors · 2024-11-25
> **Response to Reviewer 6vg7 (Part 5)**
>
> (2) **Metric = Edge Count**
>
> Edge count is another measure of graph size. Since the logical unit of computation in edge-granularity models is an edge, we may expect to see similar trends as shown by node-granularity models with the Node Count metric (Table 8).
>
> In Table 10, we display dataset statistics for each split.
>
> **Table 10:** Dataset statistics for splits of the AIDS dataset based on the Edge Count metric.
> |Datset|Avg. Edge Count (Corpus)|
> |-|-|
> |AIDS (original)|18.87|
> |Subset 0|16.93|
> |Subset 1|18.32|
> |Subset 2|19.43|
> |Subset 3|20.78|
>
> **Observation:** The effect of increasing Edge Count, while quite prominent, is not as monotonic as that seen for Node Count. In particular, for models with **Injective** interaction structure or a **Set alignment** based relevance distance, Subset 0 often shows the best performance, which indicates that usage of injective mapping and set-alignment is more immune towards the presence of padding nodes/edges.
>
> **Table 11:** MAP for **Node-Early** interaction models for splits of the dataset with increasing **edge count**.
> |Rel. Dist.|Structure|Non-linearity|AIDS|Subset 0|Subset 1|Subset 2|Subset 3|
> |-|-|-|-|-|-|-|-|
> |Agg-Hinge|Non-injective|Dot Product|0.609|0.555|0.574|0.617|**0.665**|
> |Agg-Hinge|Non-injective|Hinge|0.726|**0.763**|0.703|0.738|0.739|
> |Agg-Hinge|Non-injective|Neural|0.598|0.519|0.547|0.606|**0.659**|
> |Agg-Hinge|Injective|Dot Product|0.64|0.655|0.627|0.658|**0.672**|
> |Agg-Hinge|Injective|Hinge|0.662|0.663|0.643|0.664|**0.693**|
> |Agg-Hinge|Injective|Neural|0.614|0.535|0.566|0.623|**0.673**|
> |Agg-MLP|Non-injective|Dot Product|0.63|0.562|0.580|0.641|**0.686**|
> |Agg-MLP|Non-injective|Hinge|0.637|0.624|0.613|0.651|**0.671**|
> |Agg-MLP|Non-injective|Neural|0.629|0.564|0.584|0.646|**0.674**|
> |Agg-MLP|Injective|Dot Product|0.658|**0.687**|0.650|0.670|0.677|
> |Agg-MLP|Injective|Hinge|0.683|**0.722**|0.685|0.688|0.698|
> |Agg-MLP|Injective|Neural|0.629|0.612|0.599|0.642|**0.667**|
> |Agg-NTN|Non-injective|Dot Product|0.669|0.613|0.630|0.672|**0.715**|
> |Agg-NTN|Non-injective|Hinge|0.686|0.681|0.641|0.690|**0.726**|
> |Agg-NTN|Non-injective|Neural|0.635|0.572|0.579|0.628|**0.698**|
> |Agg-NTN|Injective|Dot Product|0.721|0.736|0.697|0.728|**0.745**|
> |Agg-NTN|Injective|Hinge|0.743|**0.773**|0.736|0.756|0.750|
> |Agg-NTN|Injective|Neural|0.667|0.643|0.619|0.675|**0.713**|
> |Set align|Non-injective|Dot Product|0.608|0.574|0.592|0.614|**0.650**|
> |Set align|Non-injective|Hinge|0.676|**0.706**|0.671|0.693|0.703|
> |Set align|Non-injective|Neural|0.593|0.566|0.566|0.616|**0.637**|
> |Set align|Injective|Dot Product|0.71|**0.767**|0.698|0.713|0.721|
> |Set align|Injective|Hinge|0.734|**0.821**|0.735|0.740|0.724|
> |Set align|Injective|Neural|0.69|**0.735**|0.680|0.705|0.704|
>
>
> **Table 12:** MAP for **Edge-Early** interaction models for splits of the dataset with increasing **edge count**.
> |Rel. Dist.|Structure|Non-linearity|AIDS|Subset 0|Subset 1|Subset 2|Subset 3|
> |-|-|-|-|-|-|-|-|
> |Agg-Hinge|Non-injective|Dot Product|0.7|0.667|0.662|0.708|**0.742**|
> |Agg-Hinge|Non-injective|Hinge|0.763|**0.784**|0.754|0.773|0.774|
> |Agg-Hinge|Non-injective|Neural|0.677|0.642|0.617|0.693|**0.723**|
> |Agg-Hinge|Injective|Dot Product|0.755|0.763|0.742|**0.770**|0.765|
> |Agg-Hinge|Injective|Hinge|0.758|**0.781**|0.742|0.769|0.775|
> |Agg-Hinge|Injective|Neural|0.68|0.628|0.645|0.689|**0.724**|
> |Agg-MLP|Non-injective|Dot Product|0.677|0.619|0.617|0.670|**0.721**|
> |Agg-MLP|Non-injective|Hinge|0.748|0.767|0.731|0.751|**0.770**|
> |Agg-MLP|Non-injective|Neural|0.662|0.592|0.619|0.665|**0.715**|
> |Agg-MLP|Injective|Dot Product|0.758|**0.778**|0.737|0.762|0.773|
> |Agg-MLP|Injective|Hinge|0.789|**0.837**|0.786|0.797|0.784|
> |Agg-MLP|Injective|Neural|0.703|0.685|0.676|0.710|**0.736**|
> |Agg-NTN|Non-injective|Dot Product|0.71|0.713|0.678|0.714|**0.741**|
> |Agg-NTN|Non-injective|Hinge|0.79|**0.842**|0.786|0.794|0.794|
> |Agg-NTN|Non-injective|Neural|0.701|0.663|0.671|0.711|**0.744**|
> |Agg-NTN|Injective|Dot Product|0.74|0.718|0.725|0.749|**0.762**|
> |Agg-NTN|Injective|Hinge|0.76|**0.785**|0.747|0.759|0.781|
> |Agg-NTN|Injective|Neural|0.689|0.669|0.642|0.695|**0.731**|
> |Set align|Non-injective|Dot Product|0.715|0.656|0.664|0.695|**0.755**|
> |Set align|Non-injective|Hinge|0.783|**0.805**|0.750|0.790|0.786|
> |Set align|Non-injective|Neural|0.708|0.661|0.670|0.709|**0.749**|
> |Set align|Injective|Dot Product|0.798|**0.829**|0.782|0.796|0.805|
> |Set align|Injective|Hinge|0.817|**0.879**|0.812|0.827|0.801|
> |Set align|Injective|Neural|0.725|0.716|0.677|0.716|**0.749**|

---

> ### Author Response · Authors · 2024-11-25
> **Response to Reviewer 6vg7 (Part 6)**
>
> > The use of notations and  are not consistent between L124 and Figure 1.
>
> Thank you for your careful reading and for pointing out this inconsistency. We have addressed this issue and corrected the notation. Additionally, based on this feedback and comments from other reviewers, we have significantly overhauled the figure and revised the caption to provide clearer explanations regarding the sequence of the design choices and the complexity of the implied design space though various combinations of the design choices.
>
>
> > Q1. Does the use of hinge distance make sense for set alignment and aggregated-hinge (L216-236)? In particular, the authors justify the use of hinge distance for adjacency matrices, but the adjacency matrices are binary while node/aggregated node embeddings are just real-valued.
>
> This is a very important question and we are glad to include further intuition, which comes from [Bloom filters](https://en.wikipedia.org/wiki/Bloom_filter). Given a universe of elements $U$ and $X, Y \subseteq U$, we can use bit vectors in $\\{ 0,1 \\}^{|U|}$ to represent $X,Y$, let's call them $\vec{X}, \vec{Y}$. The direct test for $X\subseteq Y$ is easily seen as the test $\vec{X} \le \vec{Y}$ (elementwise). A Bloom filter may be used to compress these bit-vectors into much shorter ones in $\\{0,1\\}^M$ where $M\ll|U|$, however, the test for $X\subseteq Y$ remains the same. Bloom filters have long been replaced by Learnt Bloom Filters (LBFs) and various forms of set transformers. Our work makes the natural progression from set encoders to graph encoders.
>
> --------
> **References**
>
> [Davitkova et. al., 2024] [Learning over Sets for Databases](https://openproceedings.org/2024/conf/edbt/paper-29.pdf). *Extending Database Technology (EDBT), 2024*

---

> ### Comment · Reviewer_6vg7 · 2024-11-25
>
> Thank you for your detailed and well-thought responses, which have addressed most of my questions and concerns. I've increased my scores in accordance. The paper's primary strength lies in its elegant simplification and demystification of existing approaches - a contribution I consider to be of significant value to the field. Such work that brings clarity and accessibility is essential for advancing future research. Regardless of the final acceptance decision, the review process has been intellectually enriching to me, and I want to thank the authors for preparing a high-quality submission, from which I've also learnt a lot.

---

> > ### Author Response · Authors · 2024-11-30
> > **Thanks!**
> >
> > Thank you for your encouraging comments and increasing the score.

---

### Author Response · Authors · 2024-12-04
**Closing Remarks from Authors**

Dear Reviewers and AC:

We would like to thank all reviewers for the helpful discussions and their evident overall favorable impression of the paper.

Gestalt performance comparisons obscure vital design choices whose combinations remain largely unexplored and poorly understood in the community of work on neural graph matching and (sub)graph isomorphism detection.  Conventional papers in this domain typically introduce a new model, which — like all other works — will then claim to harness certain signals. However, as we show, the key reason behind the superiority of a certain model often remains obscured. It may be because of a very simple reason, as opposed to a substantial modeling innovation. This is because the remaining design axes are not standardized across methods. Also, different components across different axes do interact.


As this rapidly evolving field continues to introduce newer models, it becomes increasingly important to focus on  bringing clarity into the space, by asking pertinent questions on why certain models perform better. We distil models into five key design axes, and instrument the existing models far better than the models themselves have done.  As reviewer 6vg7 and SvpM  pointed out, this work will bring clarity, accessibility and clear guidance to the practitioners, ensuring that the field progresses with a deeper understanding and methodological guidance, rather than mere chaotic exploration.


During the course of this exploration, we uncovered how different design components interact. This understanding provided a clear framework for navigating design choices under various time constraints, enabling practitioners to optimize performance using existing techniques. Additionally, our demystification of the design space revealed a theoretically justified, hitherto unexplored, hinge interaction non-linearity, which, through its intuitive inductive bias, delivered significant accuracy gains overall.


Certain specific choices on the design axes incidentally ended up with outperforming SOTA methods. But we did not start out with the goal to design yet another model to improve the SOTA. To embody this message, we put the comparative analysis at the end of the paper. We are not even claiming that we have exhausted all important design axes (many may remain to be explored) — any method that claims to go significantly beyond this recipe must necessarily add very distinctive design axes.  However, our hope is that the precedent set by this paper may pave the way for continued guidance to network design and evaluation.

Regards,

Authors

---

### Meta-Review · Area_Chair_3Toz · 2024-12-17

**Metareview:**

The paper explores the design space for neural subgraph matching, noting that existing methods are often limited to a narrow set of design choices. To address this, the authors propose a comprehensive framework that includes key architectural decisions: relevance distance, interaction stages, interaction structures, interaction non-linearity, and interaction granularity. This work will bring clarity, accessibility and clear guidance to the practitioners, ensuring that the field progresses with a deeper understanding and methodological guidance, rather than mere chaotic exploration.

**Additional Comments On Reviewer Discussion:**

Authors have provided sufficient evidence to clarify questions from reviewers.

---

### Decision · Program_Chairs · 2025-01-22

Accept (Poster)